
# Direct or indirect recharge on groundwater in the
# middle-latitude desert of Otindag, China?
Bing-Qi Zhu[1*], Xiao-Zong Ren[2]
[1]KLWCRESP, IGSNRR, CAS, Beijing, China
[2]SGS, TYNU, Jinzhong, China
*Correspondence to*: Bing-Qi Zhu (zhubingqi@sina.com)
**Abstract.** Although rainfall is scarce in desert lands of the world, the Otindag Desert in the
middle-latitude desert zone of northern China in Northern Hemisphere (NH) is abundant of water
resources, mainly groundwater. To gain an insight into the water origin in this desert, stable and
radioactive isotopes and major ion hydrochemistry of groundwater, as well as other natural waters
including river water, spring water, lake water and precipitation water, were investigated in the eastern
part of the Otindag. The results showed that the groundwaters in the Otindag were freshwater (TDS <
700 mg/L) and were depleted in $\delta^2H$ and $\delta^{18}O$, when compared with the modern precipitation. The
major water types were the Ca–HCO$_3$ and Ca/Mg–SO$_4$ waters. No Cl-type and Na-type waters
occurred in the study area. The ionic and depleted stable isotopic signals in groundwater, as well as the
high values of tritium contents (5-25 TU), indicated that the groundwaters studied were young but not
of meteoric origin, i.e., out of control by the modern and palaeo- direct recharge. Clear difference in the
isotopic signals werer observed between the groundwaters in the north (NPCSX) and south (SPCSX)
parts of the study area, but the signals were silimar between the groundwaters in the NPCSX and its
neighbouring catchment, the Dali Basin. The topographical elevation is decreasing from the SPCSX
(1396 m a.s.l.) to the NPCSX (1317 m a.s.l.) and the Dali (1226 m a.s.l.). Groundwaters in the NPCSX
were characterized by the lower chloride and TDS concentrations, higher tritium contents, higher
deuterium excess, and more depleted values of $\delta^2H$ and $\delta^{18}O$ than those in the SPCSX. The spatial
distribution pattern of these environmental parameters indicated a disunity bwteen the hydraulic
gradient of groundwater and the isotopic and hydrochemical gradients of groundwater in the eastern
Otindag, suggesting that the groundwaters have different recharge sources between the two parts in the
study area. However, the groundwaters in the two areas shared a common evaporation line (EL2) in the
Craig diagram of $\delta^2H$ and $\delta^{18}O$, indicating a genetic relationship in their recharge sources. Combined
analysis was further performed using the isotopic and physiochemical data of natural waters collected
from the Dali Basin and the surrounding mountains. It indicated that the major recharge sources of the
groundwaters in the NPCSX, as well as the river waters and groundwaters in the Dali Basin, were
mainly derived from the Daxin'Anling Mountains, by leaking the Xilamulan River water through thick
aquifer in the eastern margins of the Otindag. While the groundwaters in the SPCSX were mainly
recharged from two sources. One was the flash floods derived from the Yinshan Mountains and the
other was the Xilamulun River waters derived from the Daxin'Anlin Mountains. It indicates that the
modern indirect recharge mechanism, instead of the direct recharge and the palaeowater recharge, is
significant for groudnwater recharge in the eastern Otindag. This suggests that the tectoic settings at a
regional scale, but not the climate, was responsible for the groundwater origin in the Otindag. This
study provided a new sight into the origin and evolution of groundwater resources in the



middle-latitude desert zone of NH.
**Keywords:** groundwater origin; middle-latitude desert; direct and indirect recharge; stable and
radioactive isotope; ion hydrochemistry; climate control; tectonic control; Otindag Desert.

**1. Introduction**

As rainfall events are infrequent in arid and semi-arid regions of the world, surface runoff and

related water resources are globally scarce and ephemeral. These areas thus rely heavily on
groundwater as the primary water resource to support local ecosystems (Herczeg and Leaney, 2011;
Scanlon et al., 2006). It has been widely proved that the origin, quality and quantity of groundwater in
arid lands can be deeply influenced by environmental factors/processes, which controlling the
groundwater recharge and evolution, such as in the arid lands of northwestern China and Central Asia
(Zhu et al., 2015, 2016, 2017). For this reason these factors/processes become an essential component
in the understanding of regional hydrological systems and the management of water resources
(Dogramaci et al., 2012). For example, groundwater recharged by modern precipitation can refill
quickly but is vulnerable to contamination by the surface wastes, inversely, groundwater containing
mostly ancient water may not recharge to a useful extent over human timescales and cannot be affected
by surface waters (Bethke and Johnson, 2008). Therefore, different strategies on groundwater resources
management should be adopted when the different recharge mechanisms of groundwater occurring.

In general, groundwater recharge can be broadly classified into two ways, the direct recharge,

namely diffuse recharge by native water resources, and the indirect recharge, namely focus recharge by
external water resources. The direct recharge is replenished by precipitation infiltration through the
unsaturated zone and the indirect recharge is defined as recharge from mappable features such as rivers,
canals, and lakes originated from remote areas (Healy, 2010). It is well known that groundwater
recharge can be influenced by environmental factors, including climate change, underlying soil and
geology, land cover and population growth, over withdrawal and economic development (Zhu et al.,
2015, 2017), thus the amount of groundwater in arid and semi-arid regions decrease rapidly while
human demands on the limited water resources increase rather than decrease (Ma et al., 2013).
Between environment and groundwater recharge, climate and land cover largely determine
precipitation and evapotranspiration, whereas the underlying soil and geology dictate whether a water
surplus (precipitation minus evapotranspiration) can be transmitted and stored in the subsurface
(Giordano, 2009; Doll, 2009). Modelled estimates of diffuse recharge globally (Doll and Fiedler, 2008;
Wada et al., 2010) range from 13,000 to 15,000 $km^3$/yr, equivalent to ~30% of the world's renewable
freshwater resources (Doll, 2009) or a mean per capita groundwater recharge of 2100 to 2500 $m^3$/yr.
These estimates represent potential recharge fluxes as they are based on a water surplus rather than
measured contributions to aquifers. Furthermore, these modelled global recharge fluxes do not include
focused recharge, which, in semi-arid and arid environments, can be substantial (Scanlon et al., 2006;
Favreau et al., 2009). For keeping sustainable management of water resources, it requires urgently to
understand both diffuse and focused recharge and meet both human and ecosystem needs in arid areas
of the world, particularly in Central Asia and Northern China.





In the middle-latitude desert zone of northern China, many areas of these lands are unexpectedly rich in incommensurate groundwater resources, such as the Badanjilin Desert, the Mu Us Sandy Land and the Hobq Desert (Chen et al., 2012a; Chen et al., 2012b), although they have been under arid or hyper-arid climate for a long time (Sun et al., 2010). How the groudnwaters are originated and recharged in these deserts are thus becaming a key question. Until now, however, it has long been altercated in the acadamic circle. For some of the earth scientists, the direct recharge is thought to be very important for groundwaters in the wide desert lands of northwestern China due to lack of surface runoffs (Yang et al., 2010; Yang and Williams, 2003; Zhao et al., 2017). They argued that although the amount of atmospheric precipitation is small, the vast catchment area in the desert region could concentrate the rainfall into large inland basins, creating an aquifer with large storage capacity and great thickness. However, some of hydrologists suggested that the estimate of direct recharge used by the chloride mass balance method was 1.4 mm/year, approximately only 1.7% of the mean annual precipitation in a cold large desert (Badanjilin) in northern China (Gates et al., 2008). A similar estimation was only 1 mm/year for Gobi deserts from the Hexi Corridor to the Inner Mongolia Plateau in northwestern China (Ma et al., 2008). Consequently, they thought that heavy potential evaporation and little precipitation make it difficult for direct recharge to meet the supply of groundwater in these desert areas. Thus, the indirect recharge is considered to be an important mechanism for groundwater recharge in these desert areas. For example, based on isotopic compositions of natural waters, Zhao et al. (2012) suggested that little precipitation had recharged into groundwaters in the Badain Jaran Desert. Chen et al. (2004) argued that the groundwaters in the Badanjilin Desert were recharged by palaeo-glacial melt water through faults and deep carbonate layers far away from the local desert. Many studies also suggested that palaeowaters stored in aquifer during wetter climate periods could recharge to groundwater under certain conditions in arid lands (Edmunds et al., 2006; Ma and Edmunds, 2006). Other kinds of indirect recharge, such as mountain front recharge from adjacent mountain blocks, are also proposed to offer an important inflow to aquifers within arid to semiarid catchments (Blasch and Bryson, 2007).

The Otindag Desert is one of the largest desert lands in northern China and is the geographical centre of the northeastern Asian Continent, which can be regarded as a significant repository of information relating to the groundwater recharge in the arid Inner Asia. At present, the eastern Otindag is also a typical case for its incommensurate groundwater resources. There is abundant of groundwater in this desert land and even rivers originate here due to the spillover of spring water, such as the tributaries of Xilamulun River in its north and the Shandian River in its south (Fig. 1). Until now, however, little data and documents about the groundwater origin in Otindag can be obtained in literature. Whether the direct or indirect recharge is the major mechanism for groundwater recharge in Otindag, as the abovementioned hot question for other deserts in China, is also unknown.

It should be kept in mind that virgin aquatic conditions may significantly differ from managed conditions in arid environment, because groundwater recharge is not a fixed number, but may vary with the boundary conditions of the recharge system (Seiler and Gat, 2007). Conventional methods such as water balance and hydraulic methods sometimes fail in determining groundwater recharge in extreme environments (arid, semi-arid, or cold) (Drever, 1997), because of missing knowledge and the lack of

reliable data on various characteristics such as the catchment extent, input/output, the hysteretic
hydraulic functions, the transient hydraulic conditions, in-homogeneities, and on transfer functions to
overcome scale problems (Seiler and Gat, 2007). Under such conditions, tracer methods offer a
valuable support for natural water studies.
Geochemical elements and environmental isotopes have been widely used as effective tracers to
determine the sources of groundwater recharge, which could be attributed to infiltration by rainfall,
surface waters or both of them (Zhu et al., 2007, 2008; Zhu et al., 2017). For example, by comparing
the composition of stable isotopics of hydrogen and oxygen in local meteoric waters with these in
groundwaters, many studies successfully applied in identifying whether the rainfall play a vital role in
recharging groundwater or not (Zhu et al., 2007; Petrides et al., 2006; Jobbágy et al., 2011; Zhai et al.,
2013). Also, investigating the spatial distribution of groundwater age represented by the concentration
of tritium or radioactive carbon ($^{14}$C) can provide a way to understand the recharge relationship
between the modern rainfall and the groundwater (Sultan et al., 2000; Zhu et al., 2008). For the indirect
recharge, the groundwater flow regimes or its movement pathway deduced from hydrochemical and
isotopical tracers can indicate its origin and recharge processes. For example, the groundwater
mineralisation will increase as a result of dissolution of evaporite minerals along flow lines that begin
with the recharge area (Guendouz et al., 2003). While, the geochemical and isotopic composition of
groundwaters will be much complex at interface zones between groundwaters with different
hydrochemistry or ages, they will show distinct physiochemical characteristics indicating how they
mixed (Lawrence et al., 1976; Eissa et al., 2014).
The objectives of this study are (1) to examine the distribution patterns of environmental signals in
the stable and radioactive isotopes and the major ionic hydrochemistry of groundwater in the eastern
Otindag drainage system, and (2) to recognize the major sources of groundwater in the area, (3) to
identify the key mechanism of groundwater recharge in the land, particularly to discriminate whether
the direct recharge or the indirect recharge being the major control on groundwater rechage in the
desert land.

**2. Regional setting**
The Otindag Desert (~ 21,400 km$^2$), a middle-latitude desert located in the east of the Inner
Mongolia Plateau, is the fourth largest sandy land in China (Yang et al., 2012), bordered by a flat
steppe terrain to the north, the Yinshan Mountains Range and mountainous loess landscape to the south,
and the the Greater Khingan (Daxing'Anling) Mountains Range to the east (Fig. 1). The Otindag's
elevation is variable, ranging from ca. 1300 m in the southeast to ca. 1000 m in the northwest. The
desert belongs to a temperate arid and semi-arid zone of northern China, with a mean annual
temperature of 2 °C in the north and 4 °C in the south (Liu and Yang, 2013). The climate is typically
controlled by the East-Asian monsoon system whose influence is changing from southeast to northwest,
leading to the mean annual rainfall decreasing from ~450 mm in the southeast to ~150 mm in the
northwest (Yang et al., 2013). Fixed and semi-fixed sandy dunes are dominated in the desert land, with
a few mobile dunes in area of little vegetation. Dune types are various from parabolic to barchans,
linear and grid-formed types, ranging from a few meters to over 40 m in height (Yang et al., 2008; Zhu



et al., 1980).
Two rivers in the Otindag, i.e. the Xilamulun River in the north and the Shandian River with two
tributaries of the Shepi River and the Tuligen River in the south, both stem from the eastern and
southeastern part of the Otindag (Fig. 1). The Xilamulun River flows to the east and finally goes into
the Xiliao River, with a catchment area of $32.54 \times 10^3 \, km^2$ and an annual mean runoff of $6.58 \times 10^8 \, m^3$
(Wu et al., 2014). The Shandian River is the upper reach of the Luan River, with a length of 254 km
and a catchment area of $4.11 \times 10^3 \, km^2$  (Yao et al., 2013).

**3. Methods**
Fieldworks took place during the summer season of 2011 and the spring season of 2012. The
water samples selected in this study were all natural water, including the groundwater, river water, lake
water, spring water and precipitation water in types. Total of twenty-five water samples were collected
for ion chemical, stable and radioactive isotopic analysis in this study. Groundwater is the major type
among these waters, which were mainly taken from shallow and deep wells widely located in dune
fields of the study area. The surface waters were mainly sampled from rivers and lakes in the Otindag,
and the spring waters were collected from the riverhead of the Xilamulun River, the Shepi River and
the Tuligen River. One rainfall sample of the local atmospheric precipitation (p1) was also collected at
the southeastern margin of the Otindag in the 2011 summer season. Water samples were filtered using
0.45μm membrane filters for cation and anion analysis, and were acidified with 1% $HNO_3$ for cation
analysis. Water samples for stable and radioactive isotope analysis were collected in field with a
polyethylene bottle of 0.5L in volume, respectively. Some kinds of analysis were measured on site with
a    portable    instrument    (Eijkelkamp).    These    determinations    included    temperature,    pH,
oxidation-reduction potential (Eh), electrical conductivity (EC), and total dissolved solid (TDS). The
error bars were < ±0.1 °C for temperature, < ±1% for pH, < ±5% for Eh, < ±5% for EC, and < ±0.5%
for TDS, respectively.
The concentrations of major anions ($F^-$, $Cl^-$, $NO_2^-$, $NO_3^-$, $SO_4^{2-}$ and $H_2PO_4^-$) and cations ($Li^+$, $Na^+$,
$NH_4^+$, $K^+$, $Mg^{2+}$ and $Ca^{2+}$) were determined by electrochemical detectors of an ion chromatography
(Dionex 600) in the Institute of Geology and Geophysics, Chinese Academy Sciences, with error bars <
±3% for anions and <±2% for cations. The concentrations of carbonate (alkaline) ions of $HCO_3^-$ and
$CO_3^{2-}$ were measured by titration with HCl (0.1 M) following a Gran Method (Gran, 1952), with an
error bar <±5%. The hardness (HD, German standards) of these water samples was calculated based on
the equation HD = ($[Mg^{2+}] \times 100 / 24.305 + [Ca^{2+}] \times 100 / 40.08$) / 17.847, $[Mg^{2+}]$ and $[Ca^{2+}]$ referring to
the concentration of $Mg^{2+}$ and $Ca^{2+}$ with unit of mg/L.
Two stable isotopes of $^2H$ and $^{18}O$, as being expressed in δ-notation ($\delta^2H = {^2H}/{^1H}$, $\delta^{18}O = {^{18}O}/{^{16}O}$)
relative to Vienna standard mean water (VSMOW), were measured for all of the water samples
collected in this study, by MAT-252 in the Laboratory for Stable Isotope Geochemistry, Institute of
Geology and Geophysics, Chinese Academy Sciences, with σ< ±0.374‰ for $\delta^2H$ and < ±0.062‰ for
$\delta^{18}O$.
Several groundwater samples (500 ml each), collected from wells (6-60 m deep) in the study area,
were prepared for the radioactive isotope (tritium) analysis. 300 ml of water sample, added with 1 g



KMnO4, were distilled to remove any impurities. In order to increase the tritium concentration to an
easily measurable level, electrolytic enrichment was applied (Kaufman, 1954; Baeza et al., 1999). 250
ml previously distilled sample with 2.5 g NaOH was then put to the electrolysis apparatus containing
electrolytic cells with co-axial stainless steel electrodes. Electrolysis was carried out until the volume
of electrolyte was reduced to 8 ml and all runs were performed at a temperature of 2–5 °C to prevent
the loss of tritiated water molecules by evaporation. After electrolysis $CO_2$ was bubbled through the
cell to neutralize the water because the medium in which the electrolysis took place earlier is alkaline.
The water sample was separated from the electrolyte by distilling. The pretreated samples were
measured by a low-level background liquid scintillation counter (Quantulus 1220-003) according to the
manufacturer's guidelines. The error bar of the measurement should be < ±3%. The tritium data of
several groundwater samples collected in this study had been partially mentioned by Yang et al. (2015)
as one of the supplementary meterials. It was systematically discussed in this study.
**4. Results**
The analytical data of the physiochemical parameters and the stable and radioactive isotopes of
the water samples collected in this study were listed in Tables 1, 2 and 3, respectively. The study area
and the sampling sites location for each sample analyzed were showed in Figs.1 and 2, respectively.
**4. 1. Hydrochemistry of the ground and surface waters in the Otindag**
The pH values of the water samples studied varied from 6.26 to 9.44 (except sample p1,
precipitation, 4.61) (Table 1) with a median value of 7.27, indicating that the waters are generally
neutral to slightly alkaline. The TDS ranged between 67 mg/L and 660 mg/L (average 211 mg/L)
(Table 1), all belonging to fresh water (TDS < 1000 mg/L) in the salination classification of natural
water (Meybeck, 2004).
The variations in ion concentrations of the major cations and anions in the studied water samples
were displayed in a Schoeller diagram (Schoeller, 1955), a fingerprint diagram with a semi-logarithm
of y-axis (Fig. 3). In general, the groundwater samples had the highest concentrations of cations and
anions while the precipitation sample (p1) had the lowest concentrations, and the lake, river and spring
waters had the medium values. The calcium concentration was the highest in cations in almost all of the
water samples, and the $HCO_3 + CO_3$ concentration (bicarbonate + carbonate, alkalinity) was the highest
in anions in most of the water samples, except for several groundwater samples (g3, g4, g5, g6 and g11)
and one of the spring sample (s1) and the precipitation sample (p1), which had the higher $SO_4$
concentrations than the alkalinity (Fig. 3).
The relative differences in abundance of ion concentrations between different waters can be
detectable in a Piper diagram (Piper, 1944). The water samples studied can be classified into two water
types in the Piper diagram (Fig. 4), type I, the Ca–$HCO_3$ water, which generally represents the typical
bicarbonate water experienced near-surface mineral weathering, and type II, the Ca/Mg–$SO_4$ water,
which indicates saline water dominated by alkaline earth metals (Zhu et al., 2011, 2012; Clark, 2015).
For water type I, the weak acids exceeded the strong acids; the carbonate hardness (secondary
alkalinity) exceeded 50% and was dominated by the alkaline earths. While for water Type II, the strong





acids exceeded the weak acids and no carbonate hardness exceeded 50%. The alkaline earths (Ca+Mg)
exceeded the alkalis (Na+K) in all the water samples studied. There were no any Cl-type and Na-type
waters occurring in the study area (Fig. 4), indicating a primary stage of water evolution for natural
waters in the Otindag, in terms of the hydrogeochemical perspective.

The hydrochemical facies of the studied water samples can be further illustrated by an Durov

diagram (Durov, 1948) and its expanded models (Lioyd and Heathcote, 1985; Al-Bassam et al., 1997;
Chadha, 1999; Al-Bassam and Khalil, 2012). All the groundwater and spring water samples in this
study fell into the Durov fields 1, 4 and 5 of the expanded Durov diagram (Fig. 5). The water samples
in the Durov field 1 were actually same to those classified into the Piper water type 1 (Fig. 4), while
samples in the Durov fields 4 and 5 were same to those of the Piper water type II (Fig. 4). Based on the
graphic decipherment of Lioyd and Heathcote (1985), water samples in field 1 represent the presence
of $HCO_3^-$ and $Ca^{2+}$ dominant water type, while samples in field 4 indicate the $SO_4^{2-}$ dominant (or anions
indiscriminate) and $Ca^{2+}$ dominant water type, and samples in field 5 represent the water type without
any dominant anion or cation. All the groundwater and spring water samples in this study were
distributed close to the line of simple dissolution or mixing process. However, almost all the river and
lake water samples were located in the Durov field 2 and were close to the line of ion-exchange process
(Fig. 5). These distribution patterns indicated that the ground waters and the surface waters had
experienced different geochemical processes in the formation and evolution of natural waters in the
Otindag.

**4. 2. The stable and radioactive isotopes of natural waters in the Otindag**

The stable isotopes of $\delta^2H$ and $\delta^{18}O$ were analyzed for all the water samples collected in this study,

as shown in Table 3 and Fig. 6. The radioactive isotope of tritium ($^3H$) was analyzed for a part of the
groundwater samples.

The $\delta^2H$ values of the groundwater samples collected in this study varied from -63.42‰ to -75.92‰

(Table 3), with an average -69.53‰. The $\delta^{18}O$ values ranged between -8.64‰ and -11.26‰ (Table 3),
with an average -10.17‰.

The spring water samples, which directly drain into rivers, were relatively concentrated in values

of $\delta^2H$ and $\delta^{18}O$ and were greatly similar to those of the groundwater sample (Fig. 6). The $\delta^2H$ and $\delta^{18}O$
values in the spring samples varied from -70.83‰ to -72.60‰ (mean value -71.72‰) and from -10.34‰
to -10.47‰ (mean value -10.40‰), respectively (Table 3).

The $\delta^2H$ and $\delta^{18}O$ values in the river water samples were slightly varied and were also similar to

those of the groundwater (Fig. 6), with a range of between -65.00‰ and -85.16‰ (mean value
-73.02‰) in $\delta^2H$ values and a range of between -9.55‰ and -11.78‰ (mean value -10.51‰) in $\delta^{18}O$
(Table 3).

The lake water samples in this study were enriched in $\delta^2H$ and $\delta^{18}O$ comparing to the groundwater

samples (Fig. 6), with a variable range of between -34.16‰ and -53.13‰ (mean value -46.47‰) in
$\delta^2H$ values and a range of between 0.38‰ and -6.55‰ (mean value -4.65‰) in $\delta^{18}O$ (Table 3).

The precipitation sample p1 showed the $\delta^2H$ value of -47.4‰ and the $\delta^{18}O$ value of -7.14‰,

respectively (Table 3).



The isotopic regression equation of the Otindag evaporation line (EL1) (Fig. 6), which was calculated based on the $\delta^2H$ and $\delta^{18}O$ data of the groundwater, lake, river and spring water samples in this study, was $\delta^2H = 4.09 \; \delta^{18}O - 28.31$ ($R^2=0.93$, n=24).

The content of radioactive isotope of tritium ($^3H$) was measured in seven well groundwater samples with 6-60 m deepth in this study. The tritium concentrations ranged from 1.86 to 24.35 TU (Table 3), with an average 14.95 TU, higher than the mean tritium concentration (9.8 TU) of groundwater in the Vienna Basin, Austria (Stolp et al., 2010), the seat of the International Atomic Energy Agency (IAEA).

**5. Discussion**

**5.1. Comparison of the isotopic signals between the modern regional precipitation and natural waters in the Otindag**

At present, the extensive record of stable isotope measurements in atmospheric precipitation is still absent in the Otindag, thus the decadal isotope data of atmospheric precipitation around the Otindag were collected in this study to determine the isotopic relationship between the local groundwater and the regional precipitation. A global database, the IAEA Global Network of Isotopes in Precipitation (GNIP), is available to use in this study. Taking into account the boundary between the northern hemispheric westerly and the Asian summer monsoon (Chen et al., 2010), which are the two major climate systems controlling the Otindag (Yang et al., 2013), we chose two GNIP meteorological stations as the representations of the atmospheric precipitation derived from the northern hemispheric westerly and the Asian summer monsoon, respectively. One is the Baotou station located to the southwest of the Otindag (the westerly system), and another is the Tianjin station located to the southeast of the Otindag (the Asian summer monsoon system) (Fig. 1a). The historical isotopic data ($^3H$, $\delta^2H$ and $\delta^{18}O$, ‰VSMOW) over the last four decades from the two stations , as well as other data including the daily precipitation amount (mm) and air temperature (°C) in the same period, were taken as the references of the stable isotopic signals in precipitation in the Otindag.

The annual weighted mean values of $\delta^2H$ and $\delta^{18}O$ at the Baotou station were variable from -64.32‰ to -48.44‰ and from -9.40‰ to -6.50‰ during the period of 1986 to 1992, respectively. The annual weighted mean values of $\delta^2H$ and $\delta^{18}O$ at the Tianjin station varied from -56.30‰ to -43.72‰ and from -8.35‰ to -6.86‰ during the period of 1988 to 1992 and of 2000 to 2001, respectively. The long-term weighted mean values of $\delta^2H$ and $\delta^{18}O$ at the Baotou station (LWMB) were 55.27‰ and -7.78‰, respectively, and were -49.97‰ and -7.70‰ at the Tianjin station (LWMT), respectively. The radioactive isotope of $^3H$ (TU) in precipitation was not stable at the GNIP Baotou station. The annual weighted mean values were higher than 30 TU in this station and tended to be decreased from 1986 to 1991 (72.06, 57.81, 59.97, 52.79, 55.89, 34.35 TU, respectively). The annual weighted mean values of $^3H$ at the GNIP Tianjin station were lower than those of the Baotou station. The mean values were 21.99, 21.65, 18.55, 25.72, 18.80 TU from 1988 to 1992, and 7.01 and 15.48 TU from 2000 to 2001.

As the only one precipitation sample collected in this study during the 2011 summer rainfall event of the Otindag, the sample p1 fell onto the Global Meteoric Water Line (GMWL: $\delta^2H = 8\delta^{18}O + 10$)



estimated by Craig (1961). It showed similar $\delta^2$H and $\delta^{18}$O values to those of the precipitation in the
GNIP stations of Baotou and Tianjin (Fig. 6).
Compared to the precipitation data from the GNIP Baotou and Tianjin stations and from the local
precipitation (p1) in the Otindag, the groundwater samples were evidently depleted in heavy stable
isotopes in the HSKDSL (Fig. 6).
In contrast to the precipitation data, the water samples from springs and rivers in the study area
also showed a depletion characteristics in the stable isotopes of $\delta^2$H and $\delta^{18}$O (Fig. 6).
The regional meteoric water line, i.e. the regional Craig line, can be statistically described as the
isotopic regression equation of $\delta^2$H = 6.36$\delta^{18}$O - 5.21 (line LMWL-B), based on the isotopic data at the
Baotou station, and can be destribed as $\delta^2$H = 6.57$\delta^{18}$O + 0.31 (line LWML-T), based on the data at the
Tianjin station (Fig. 6). Excapt for the lake water samples, most of the groundwater, river water and
spring water samples in the Otindag fell on or lay between the LMWL-B and the LMWL-T lines, and
were located at the lower left area of the precipitation points (Fig. 6). This indicated that no deep
evaporation process was experienced by these ground and surface waters (except for lake waters) than
the precipitation.
For the Otindag evaporation line (EL1), its equation slope and intercept were significantly lower
than that of the GMWL, LMWL-B and LMWL-T (Fig. 6). The point of intersection between the EL1
and LMWL-B was -69.93‰ for $\delta^2$H and -10.18‰ for $\delta^{18}$O, respectively, while the intersection point
between the EL1 and LMWL-T was -75.51‰ for $\delta^2$H and -11.54‰ for $\delta^{18}$O, respectively.

**5. 2. The direct recharge of groundwater in the eastern Otindag**
Water infiltration of atmospheric precipitation through the unsaturated zone to groundwater is
hydrologically definied as the direct recahrge. The deuterium and oxygen isotopes are the composition
of water molecules and are sensitive to physical processes such as mixing and evaporation, hence they
are ideal tracers of the origin of groundwater (Coplen, 1993; Scanlon et al., 2006). We used them to
identify the contribution of preicipitation recharge on groundwater in this study.
Because the annual mean precipitation amount in the semi-arid regions of northern China is
between 200~400 mm, it seems that the direct recharge on groundwater cannot be neglected in the
eastern Otindag under a semi-arid climate. However, when we checked the stable isotopic data from the
GNIP stations both at the Baotou and Tianjin, we observed that almost all the annual weighted mean
values of the stable isotope contents in precipitation were enriched in $\delta^2$H and $\delta^{18}$O than those values
measured for the groundwater, spring water and river water samples in this study (Fig. 6). Because the
isotopic evolution of $\delta^2$H and $\delta^{18}$O in water illustrated in the Craig line represents a one-way and
irreversible process, thus the water bodies distributed at the upper right area of the Craig line can not be
recharge sources for the water bodies distributed at the lower left area of the line. Such results indicated
that the groundwater, river water and spring water in the Otindag were not recharged by the regional
precipitation, namely no significant modern direct recharge has taken place for groundwater in the
Otindag.
Dogramaci et al. (2012) documented that only the intense rainfall events of >20 mm could
remarkably recharge groundwater in the semi-arid Hamersley Basin of northwest Australia, while the



rainfall events <20 mm had limited influences on groundwater recharge. Chen et al. (2014) described
that rainfall events ≤5 mm in the arid and semi-arid region of northern China would be evaporated into
atmosphere rapidly before it is infiltrated into the groundwater system. Based on the analysis on the
data records from two meteorological stations around the Otindag, i.e., the Duolun station and the
Xilinhaote station (see Fig. 1a), we observed that the average times of rainfall events being >20 mm in
amount were only 2.5-3.4 times per year (Table 4). Even none of the rainfall events of >20 mm
occurred during the year from 2005 to 2007 at the Xilinhaote Station. It further confirmed that the
small amounts of intensive rainfall events had limited the contribution of regional precipitation on
groundwater recharge in the Otindag.
In addition to groundwater, the river water and spring water samples in the Otindag had the
similar isotopic signals with those groudnwaters, and were also deviated from the modern regional
precipitation in the Craig diagram (Fig. 6). These water samples came from the Xilamulun, Shepi and
Tuligen rivers. They shared the same evaporation line (EL1) with the groundwater and lake water
samples (Fig. 6). Generally speaking, natural waters that have a same recharge source can be
distributed on a same line of evaporation in the $\delta^2$ and $\delta^{18}O$ diagram (Chen et al., 2012b). This
indicated that the recharge sources of groundwater, river water, spring water and lake water in the
Otindag were genetically associated and were differential to the regional precipitation. During the field
investigation, we observed that the elevation of spring outflow was lower than that of the groundwater
table in some areas. This implyed that the sping water can be originated from the local phreatic water
(groundwater). The same isotopic signals between the two kinds of water confirmed their close
relationship in origin.

**5.3. Potential sources of groundwater other than summer precipitation in the Otindag: three**

**hypotheses**

Since the groundwater samples in the Otindag were depleted in the $\delta^2$ and $\delta^{18}O$ values even than
those of the modern rainfall (Fig. 6), they must be sourced from other waters with same or more
depleted signals in the stabe isotopes compositions.
Because the Otindag is under the control of the East Asian Summer Monsoon climate (Yang et al.,
2013), the modern rainfall in the desert is mainly sourced from the summer season's precipitation, with
rain and heat over the same period. These climatic characteristics were illustrated by the seasonal
distributions of the annual mean precipitation amount (Fig. 7a), the annual mean air temperature (Fig.
7b) and the annual mean water vapor pressure (Fig. 7c) over the last forty years at the two surrounding
GNIP weather stations in the Baotou and Tianjin. These records indicates that the summer rainfall is
warmer and relatively positive in the signals of $\delta^2H$ and $\delta^{18}O$ than those of the waters originated in a
colder environment, due to the evaporation effect on isotopic fractionation. It thus can be speculated
that the potential water sources of groundwater in the Otindag must be derived from waters originated
in a colder environment, sucha as (1) the modern precipitation in winter, (2) the palaeowater formed in
the past glacial period, or (3) the mountains waters with colder and wetter conditions.
Given the hypothsis (1) "the modern winter precipitation", we can get clues from the isotopic
records of winter precipitation in the Baotou and Tianjing stations. It is shown that the annual mean



values of $\delta^2H$ and $\delta^{18}O$ over the last forty years were more depleted in the winter perecipitation than in
the summer precipitation at the Baotou and Tianjin stations (Fig. 8a-b). This suggested that the regional
winter precipitation was qualified to be a potential souce of groundwaters in the Otindag. However, the
limited water amount of the winter precipitation in these regions seemed to be a question towards its
importance as an efficient source of groundwater, because the precipitation amounts and the water
vapor pressures (effective moisture) in the winter months were much lower than those in the summer
months at both the Baotou and Tianjin stations (Fig. 7a and 7d). It indicated that the winter seasons in
these regions were relativly colder and drier but not colder and wetter. A colder-wetter pattern of winter
precipitation is necessary as a water source for the formation of groundwater under a summer monsoon
climate, because the bigger amounts of summer precipitation will easily remove or weaken the deplted
isotopic signals of winter prepicitation in groundwater. In view of this consideration, the modern winter
precipitation might not be an important souce of groundwater in the Otindag. The hypothesis (1) can be
neglected.
As to the hypothesis (2) "the palaeowaters" formed in colder and wetter periods such as the last
glacial", it has been proposed to be a potential water source for groundwaters in the wide arid lands of
the world. In fact, the depleted signals of stable isotopes ($\delta^2H$ and $\delta^{18}O$) in groundwater have been
recognized in global arid and semi-arid regions, such as the Sinai Desert in Egypt (Gat and Issar, 1974),
Israel (Gat, 1983), South Australia (Love et al., 1994, 2000), northern China (Ma et al., 2010), Saudi
Arabia and North Africa (Moser et al., 1983; Guendouz et al., 2003). The signals are very often
explained as palaeo-groundwater that recharged by precipitation during past wetter and colder periods
( Love et al., 1994, 2000; Herczeg and Leaney, 2011). Gat and Issar (1974) reported that palaeowaters
played a central role in the deep aquifers of the Sinai Desert, with the evidents that groundwater stable
isotope compositions ($\delta^{18}O$ and $\delta^2H$) were more negative than those of weighted mean contemporary
rainfall. Ma et al. (2010) presented data from groundwater in the aquifer of Jinchang city and the
adjacent Gobi desert areas in northern China, which showed that palaeowaters were depleted in $^{18}O$ and
$^2H$ relative to modern precipitation in the same region.
In order to identify the role of palaeowater recharge on groundwater in the Otindag, we used the
tritium data as a environmental tracer to estimate the groundwater age in the Otindag. The half life of
tritium is 12.43 yr. Based on this decay time and the tritium concentrations in groundwater, the
exponential decay equation can be used to provide a qualitative age indication to interpretate the
regional groundwater flow system (Ma et al., 2010). Due to the lack of tritium data of local
precipitation in the Otindag, we still used the tritium data at the GNIP stations of the Baotou and
Tianjin as the background values in precipitation of recent years.
A "piston model (flow)" was used to evaluate the residence time of groundwater in aquifer and the
residual tritium of a water body can be calculated by $N = N_0 e^{-\lambda t}$ (Yang and Williams, 2003). Where N =
content of residual tritium in water sample, $\lambda = 0.0565$, the radioactive decay constant, $N_0$ = content of
tritium at the time of rainfall and t = years after precipitation. Based on this equation, the residual
tritium was theoretically calculated and the standard for tritium dating was established. In this study,
the content of tritium was measured for seven groundwater samples (Table 3), all of which were taken
from the wells in the Otindag dune field. To the extent that the input function and piston model are





reasonable approximations, age of 0-60 years were obtained for these groundwater samples (Table 5),
which indicated that recent recharge after the global nuclear tests had been several decade years
undereway. Based on the relatively high tritium contents and the calculated datings of the groundwater
samples in this study (Table 5), we concluded that groundwater is generally not older than 70 years in
the study area. The hypothesis (2) that the groundwater were palaeowater rechared during glacial
period in the Otindag is not valid.

Both the hypotheses (1) and (2) were proved to be valid, indicating that the direct recharge is not a

major mechanism controlling the groundwater recharge in the Otindag.

### 5. 4. The indirect recharge of groundwater in the eastern Otindag?

Through the above analysis, it seemed that the modern winter meteoric water was not a

volumetrically important source of groundwater in the Otindag, and the groundwater was not recharged
by palaeowaters. Thus, the third hypothesis, "the mountains waters with colder and wetter conditions",
should be considered as a key souce of groundwater in the Otindag. In essence, it is an indirect
recharge machnism, as the indirect recharge is defined as water originated from remote areas (Healy,
2010) and it generally occurs through rivers, canals, lakes and flash floodings (Herczeg and Leaney,

2011).

It was worth noting that the values of deuterium and oxygen-18 in the groundwater samples of the

eastern Otindag were variable. These values for groundwater in the north part of the study area were
more depleted in $\delta^2$H and $\delta^{18}$O than those in the south part (Table 3). It suggests that the groundwater in
the study area might be potentially recharged by water resouces coming from the northern neighboring
catchment of the eastern Otindag, such as the Dali Basin.

In order to estimate the potential linkage between the eastern Otindag and the Dali Basin, recently

published data of deuterium and oxygen-18 in groundwaters, lake waters, river waters and spring
waters sampled from the Dali Basin (e.g., Chen et al., 2008; Zhen et al., 2014) were collected in this
study and were co-analyzed with the data from the Otindag.

There were totally about 70 natural water samples from the Dali and Otindag with $\delta^2$H and $\delta^{18}$O

values being shown in a Craig diagram (Fig. 9). As a result, all of these samples fell on or lied near the
evaporation line EL2 in the Craig diragram (Fig. 9), with a regression equation $\delta^2$H = 4.81 $\delta^{18}$O - 21.55
and a higher correlation coefficient ($R^2$=0.98, n=70) than that of EL1 ($R^2$=0.93, n=24) for the Otindag
samples.

Compared to the groundwater samples in the Otindag, water samples from the groundwaters,

rivers and springs in the Dali Basin were more depleted in $\delta^{18}$O and $\delta^2$H (Fig. 9). Such results further
indicated that, in terms of the isotopic perspective, the groundwater in the eastern Otindag has a close
relationship with the natural waters in the Dali Basin, except for the lake water in Dali. It seems that the
Dali water is a potential source for groundwater in the Otindag, or both of them are recharged by a
common souce derived from surrounding mountains.

### 5. 4. 1. Linkage of the river water in the Dali and the groundwater in the Otindag

The similar signals of deuterium and oxygen-18 between the groundwater in the Otindag and the



river water in the Dali (Fig. 9) gave us a possible idea that the groundwate in the Otindag might be
sourced from the river water in the Dali Basin, since the Dali has more depleted isotopic signals in
water than the Otindag (Fig. 9).

Regarding to the topographical gradient of the elevations between the two regions, however, river

water in the Dali Basin can't flow into the eastern Otindag, because the terrain elevation of the Dali
Basin is lower than that of the Otindag (Fig. 1). This is also the reason why the huge Dali Lake is
formed in the Dali Basin but not in the Otindag (Fig. 1). If there is a hydraulic linkage between the two
regions, water should flow from the Otindag into the the Dali, but not conversely.

A hypothesis that water flows from the Otindag into the Dali Lake has also been proposed by

Yang et al. (2015). They argued that a mega-palaeolake in Dali, who was almost twice the size of the
present Dali Lake in area, was recharged by river systems to its south in the Otindag ca. 4,200 years
ago. After that, due to the monsoonal regions being experienced catastrophic precipitation decreasing
and the groundwater in Otindag being sapped and captured by the Xilamulun River flowing eastward,
the Otindag's water was no longer recharging the megalake Dali and left a palaeo-channel between the
two regions (Fig. 2). Since then the connection between surface waters in the two regions was broken.

In view of the hydraulic gradient, river water in the Dali Basin could not be a recharge source for

groundwater in the Otindag. However, in view of the isotopic gradients, groundwater in the Otindag
could not conversely be the source of river water in the Dali at present, due to the more depleted values
of deuterium and oxygen-18 in Dali than in Otindag (Fig. 9). Thus, the similar isotopic signals between
the river water in Dali and the groundwater in Otindag indicated that these waters might be recharged
from a common souce.

**5. 4. 2. Linkage of groundwaters between the Otindag and the Dali**

Similar isotopic signals also occurred in the groundwaters between the Otindag and the Dali Basin

(Fig. 9). The linkage of groundwaters between the two regions is still unknown at present. In order to
answer this question, we need to know the potential movement of groundwater in the transition zone of
the two regions.

Due to the difficult to directly observe groundwater movement along its hydraulic gradient under

ground, inert isotopic and hydrochemical tracers are often used to identify groundwater movement
(Nakaya et al., 2007), such as chloride, TDS and H-O isotopes, which were used as environmental
fingerprints to indicate groundwater movement in arid land (Yang and Williams, 2003). In a theoretical
line of groundwater evolution, the chloride in water is readily removed from matrix materials rather
than being precipitated due to its high solubility, thus chloride concentrations tend to be increased with
the increasing of the flow path's length and residence time of groundwater (Lioyd and Heathcote,
1985). The TDS has a similar trend with chloride in groundwater evolution, but its tendency might be
disturbed due to potential precipitation of certain ions when reaching their saturation conditions.
According to the salination classification of water, all the groundwater samples collected in this study
were fresh water in type (TDS < 1000 mg/L). Thus evident precipitation of major ions could be weak
in the Otindag groundwaters.

In this study, a groundwater-sampling project was designed in field along an N-S section of a





palaeo-channel located at the transition zone between the Dali and Otindag (Figs. 1, 2). The channel is
located near the south distal reach of the Xilamulun River and was named "PCSX" in this study. The
north part of the channel, named as NPCSX, is located at the riverhead of the Xilamulun River and the
south part (SPCSX) is close to the eastern margin of the Yinshan Mountains (Figs. 1, 2).

Regarding to the topographical gradient in the Otindag, the GPS elevation of the northernmost

sampling site in the NPCSX (g11, about 1317 m a.s.l.) was much lower than that of the southernmost
site in the SPCSX (g1, 1396 m a.s.l.) (Fig. 2 and Table 1). It is about an 80-meter drop between the
NPCSX and the SPCSX. Under such slope, the underground hydraulic gradient for groundwater flow
can be roughly parallel with that of the surface water flow, namely that the groundwater flow should
move downwards from the SPCSX area to the NPCSX area. Thus we can speculate that groundwater in
the NPCSX would have higher values of chloride and TDS in concentration than those in the SPCSX,
if the groundwater was flowing from the SPCSX to the NPCSX.

In order to check up this speculation, the actual variations of the environmental tracers (chloride

and TDS) were detected along the PCSX section. The sampling site g1 was defined as the initial point
and the distances between g1 and other sampling sites along the PCSX section were calculated, based
on their GPS geographical coordinate records measured in the field. The results were shown in Fig. 10.
It was very clear that the variations of chloride and TDS concentrations in groundwater did not increase
along the palaeo-channel from south to north (Fig. 10). On the contrary, both the values of chloride and
TDS were lower in the NPCSX area than those in the SPCSX area. Such kind of spatial variations in
the chloride and TDS values was contradicted to the speculated patterns abovementioned, suggesting a
complicated movement of groundwater in the study area. It also indicated that the hydraulic linkage
was weak in the groundwaters between the NPCSX and SPCSX areas.

The stable and radioactive isotopic data were also used here as tracers to differentiate the

groundwaters between the two regions. Before we use the stable isotopic signals, however, it is
necessary to think about the effect of evaporation process on the fractionation of stable isotopes.
During the evaporation process, dissolved chloride, the conservative ion, will be enriched along with
the heavy isotopes, which is manifested as a correlation between the chloride concentration and the
deuterium content in groundwater (Sklash and Mwangi, 1991; Taylor and Howard, 1996). Based on
this consideration, a bivariate diagram was built using the chloride and deuterium data of the
groundwater samples in this study, as shown in Fig. 11. The Groundwater samples from the PCSX
section showed a very weak correlation between the chloride and deuterium (Fig. 11). This indicated
that the groundwaters studied were not affected by evapotation process in a deep degree.

Compared between the NPCSX and SPCSX regions, the stable isotopic values ($\delta^{18}$O and $\delta^2$H) of

groundwaters in the SPCSX region varied greatly with a large amplitude, while those in the NPCSX
were relatively constant (Fig. 12). This indicated that the recharge sources of groundwater in the
SPCSX were diversity than those in the NPCSX. The constant variations   indicated that the recharge
source of groundwater in the NPCSX is relatively unitary. The isotopic values in the SPCSX were
much lighter than those in the NPCSX along the distance section from south to north (Fig. 12). The
heaviest valuses occurred in the sample g11 collected from the NPCSX (Fig. 12), indicating a water
being firsthand recharged. The spring water sample s2, a representation of discharge water, was



characterized by medium values of $\delta^2H$ and $\delta^{18}O$. Similarly, the deuterium excess values of these
groundwaters also showed such spatial patterns in the two regions (Fig. 13). These results indicated
that the groundwaters in the SPCSX area, with relatively enriched isotopic signals in $\delta^2H$ and $\delta^{18}O$ than
those in the NPCSX area, were an mixture of the groundwaters in the NPCSX and other waters, thus
resulting in the spring water sample s2 in the discharge zone being characterized by an intermediate
isotopic signal (Figs. 12, 13). A similar case was also observed by Abdalla (2009), who reported that
the isotopic compositions had decreased progressively along a regional-scale flow path of groundwater
in the semi-arid central Sudan, because of the mixture of groundwaters between the heavier-isotope
recharged and the lighter-isotope recharged.

In addition to stable isotopes, the tritium contents were broadly positively related to the values of
deuterium excess in the groundwater samples in the PCSX section (Fig. 14a). The deuterium excess or
d-excess, computed from the equation $d = \delta^2H – 8\delta^{18}O$ (Dansgaard, 1964), is controlled primarily by
the mean relative humidity of the air masses formed above the water surface (Merlivat and Jouzel,
1979) and generally reflects the rate of evaporation process experienced during the flowing paths
(Dansgaard, 1964). For a water experienced evaporation process, the d-excess value will increase in the
evaporated water vapor, but will decrease in the residual water body. In this study, except for sample
g11 (a sample very close to the riverhead area), the positive relationship between the tritium and the
deuterium excess generally showed that the d-excess values were higher in the groundwaters collected
from the NPCSX, but were lower in those from the SPCSX (Fig. 14a). The distribution pattern
indicated that the groundwaters in the NPCSX were relatively younger and had experienced less degree
of evaporation than those in the SPCSX. The d-excess gradient, increasing from the south to north in
the PCSX, further confirmed that groundwater did not flow from the SPCSX area to the NPCSX area.

In Fig. 14b, the tritium contents of groundwater increased while the TDS decreased from the south
to north in the PCSX (Fig. 14b). This distribution pattern of the two environmental tracers further
proved that the groundwaters in the NPCSX were younger and fresher than those in the SPCSX. The
reason why the older gourndwate has a higher TDS value can be attributed to the fact that most
minerals dissolve slowly in aquifer and the older groundwater have more contacting time to act
between water solution and soluble minerals, leading to a higher TDS (Fitts, 2002). Many studies (e.g.,
Boronina et al., 2005; Kazemi et al., 2006) have demonstrated that groundwater will flow in the
direction in which it gets older. In view of this point, groundwaters in the PCSX region should
theritically flow from the NPCSX area to the SPCSX area, evidently being paradoxical with the S-N
topographical gradient in the PCSX region.

Overall, it implied that the hydraulic gradient of groundwater in topography is not consistent with
the isotopic and hydrogeochemical gradient of groundwater in the eastern Otindag.

**5. 5. Potential sources of groundwater recharge in the Otindag: the Daxinganling and Yinshan**
**Mountains:**

The discussions above indicated that groundwater in the eastern Otindag has a close relationship
with river water in the Dali Basin in terms of the isotopic perspective, and both the river water and
groundwater in the two regions might be recharged from a common source derived from other place.



Meanwhile, the isotopic and hydrochemical characteristics of groundwaters both in the NPCSX and
SPCSX areas indicated that the groundwaters between the Dali (togethern with the northeast Otindag)
and the southeast Otindag were different and the groundwater systems in the two regions were not
integrated.
For the Dali catchment, the Dali Lake and its surrounding rivers are the most important water
bodies in the Dali Basin. There are two large permanent rivers and lots of small intermittent streams
entering the Dali Lake (Xiao et al., 2008), including the Xilamulun River to the south and the Gongger
River to the north, both of which are stemming from the Greater Khingan Mountains (Daxinganling
Mountains in Chinese term, 1,100-1,400 m above seal level) (Fig. 1). The Xilamulun River, 380 km in
length and $32.54 \times 10^3$ km² in area, is a neighboring river both to the southeastern Dali and to the
northeastern Otindag (Figs. 1 and 2). It carries a large amount of water (about $6.58 \times 10^8$ m³/y) from the
Daxinganling Mountains flowing through the east margins of the Dali and Otindag (Wu et al., 2014).
This is an important clue linking groundwaters in the northeastern Otindag and the river waters and
groundwaters in the Dali.
Variation of the elevation from the Dali Lake to the riverhead of the Xilamulun River can be
clearly found along a land surface topographical section (Fig. 15). The channel of the Xilamulun River
is located in a fault called the Xilamulun River Fault or the Xar Moron River Fault, which is a part of
the Xilamulun-Changchun-Yanji plate suture zone (Sun et al., 2004) or the Solonker Suture Zone
(Eizenhöfer et al., 2014) in the regional tectonical settings (Figs. 1 and 2). When rivers stem from the
Daxinganling Mountains and flow downward to the marginal areas of the Dali and Otindag, leakage
water from these rivers can recharge the desert land through thick unconsolidated aquifers (Fig. 15). A
strong isotopic evidence is that the lake and river waters in the Dali Basin share the same evaporation
line (EL2) with the groundwaters in the PCSX area. Although groundwaters in the SPCSX area were
different from those in the NPCSX area, their isotopic data points still fell onto the EL2, which further
indicated that the groundwaters in the SPCSX were a mixture of waters from the Daxinganling
Mountain and other source.
Another source for groundwater recharge in the SPCSX can be speculated to flash floods derived
from the north Yinshan Mountains (Fig. 1), because it can be clearly observed from digitial maps that
many transient rivers or streams originated from the Yinshan Mountrains flow into the south and
southeastern Otindag (Fig. 1). A key clue for this view can also be obtained from the isotopic signals of
local precipitation and groundwater samples collected from the areas near to the Yinshan Mountains in
this study.
It has been reported that the isotope-depleted signals of $\delta^2$H and $\delta^{18}$O in waters from mountain
areas can be passed into the groundwater in a plain area (Harrington et al., 2002; Vanderzalm et al.,
2011; Liu and Yamanaka, 2012; Rattray, 2015; Khalid and Hamid, 2017). Rattray (2015) attributed this
isotopic signature to the altitude effect on precipitation, because temperature and altitude can deeply
affect the deuterium and oxygen-18 compositions in precipitation. The values of $\delta^2$H and $\delta^{18}$O in
precipitation from the mountain areas will be depleted when compared with those in precipitation from
the piedmont areas (Rattray, 2015). For the Yinshan Mountain Range, there is lack of the data of stable
isotopes in precipitation from the mountains in this study. However, based on the altitude effect of





temperature on isotopic signals, we can theoretically estimate the values using the precipitation sample
(p1), which was collected from the piedmont area of the Yinshan Mountains in this study. For example,
the GPS elevation of the sample location of p1 is about 1260 m a.s.l. and that of the top of the Yinshan
mountain range is around 1700-1800 m a.s.l., thus the elevation drop is approximately 500 m between
the two sites. Based on this elevation drop and a potential effect of elevation change on temperature
that elevation arises will lead to a decrease of temperature by 0.65 °C per 100 m, the temperature
difference between the two sites is about 3.25 °C. According to an empirical estimation for
precipitation in NW China that the $\delta^{18}$O-temperature gradient is 0.37 ‰/°C and the $\delta^{18}$O-elevation
gradient is -0.13‰/100 m (Liu et al., 2014), the $\delta^{18}$O value in precipitation at the Yinshan Mountains
shall be 1.85 ‰ lower than that in the sample p1, namely -8.99 ‰ in $\delta^{18}$O for the Yinshan mountain
precipitation. This value is very similar to that of the groundwater (-9 ‰) in the SPCSX area. It
indicates that the Yinshan Mountains are a potential source area for the groundwater recharge in the
SPCSX area.

In general, the above analyses revealed that the highland water resources from the Daxing'Anling
and Yinshan Mountains were isotopically and geochemically traced to be a major source for the
groundwater in the Otindag. It means that the modern indirect recharge mechanism, instead of the
direct recharge and the palaeowater recharge, is responsible for groudnwater recharge in the desert land
in northern China. This implies that the tectoic settings, but not the climate control, was significant for
the groundwater origin in the Otindag.

**6. Conclusions**

Water resources in arid lands of the world are generally scarce and highly uncertain. In the
middle-latitude desert zone of northern China, however, many deserts are unexpectedly rich in
incommensurate groundwater resources, such as the Otindag and the Badanjilin Deserts, although they
have been under an arid or hyper-arid climate for a long geological period. How the groundwaters are
originated and recharged in a desert environment are thus becaming a key question longtime ago, but it
is still under an endless debate at present in the acadamic circle. For some of the earth scientists, the
direct recharge is thought to be very important for groundwaters in the wide desert lands of northern
China, due to lack of surface runoffs. However, the groundwater availability is very much as function
of the local- and regional-scale geological and climatic components. Integrated understanding of the
groundwater recharge and their controlling mechanism is of great significance. In this study, an effort
to explore the groundwater recharge was carried out using multiple environmental tracers in the eastern
Otindag of northern China, where is under the control of the East Asian Summer Monsoon (EASM)
climate. The results showed that (1), the natural waters in the study area were fresh water (TDS < 1000
mg/L) and were neutral to slightly alkaline. The major water types were the Ca–HCO$_3$ and Ca/Mg–SO$_4$.
There were no Cl-type and Na-type waters occurring in the study area, indicating a primary stage of
water evolution in terms of the hydrogeochemical perspective. (2) Compared to the modern summer
precipitation, the groundwaters, river waters and spring waters were depleted in $\delta^2$H and $\delta^{18}$O, while
the lake waters were enriched in $\delta^2$H and $\delta^{18}$O. All these waters, however, shared a same line of
evaporation in the Craig diagram, indicating a genetic relationship on their recharge sources. The more



depleted stable isotopic signals in the groundwaters than those in the modern summer precipitation suggested that the groundwaters studied could only be sourced from a colder water other than the EASM precipitation. The contribution from local winter precipitation was very small due to its weak rainfall effect. The high contents (5-25 TU) of tritium in these groundwaters indicated that they were young and could not be recharged by palaeowaters formed during the past glacial periods. (3) Clear difference in the isotopic signals occurred between the groundwaters in the north (NPCSX) and south (SPCSX) parts of the study area, but the signals were silimar between the groundwaters in the NPCSX and its neighbouring catchment, the Dali Basin. (4) Combined analysis was further performed using the isotopic and physiochemical data of natural waters collected from the Dali Basin and the surrounding mountains. The resulsts indicated that the major sources of the groundwaters in the NPCSX, as well as the river waters and groundwaters in the Dali Basin, were mainly derived from the Daxin'Anling Mountains, by leaking the Xilamulan River water through thick aquifer in the eastern margins of the Otindag. While the groundwaters in the SPCSX were mainly recharged from two sources, the flash floods from the Yinshan Mountains and the river waters from the Daxin'Anlin Mountains. (5) The modern indirect recharge mechanism, instead of the direct recharge and the palaeowater recharge, was significant for groudnwater recharge in the eastern Otindag. It indicates that the tectoic settings at a regional scale, but not the climate, was responsible for the groundwater origin in the Otindag. This study provided a new sight into the origin and evolution of groundwater resources in the middle-latitude desert zone of northern China.

### Acknowledgements

This study was financially supported by the National Natural Science Foundation of China (41771014) and the National Key Research and Development Program of China (2016YFA0601900). We thank the China Meteorological Data Sharing Service system for providing the weather data. Sincere thanks are also extended to Profs. Xiaoping Yang, Xunming Wang, Jule Xiao and other workmates, e.g., Qiuhong Li, Ziting Liu, Hongwei Li, and Deguo Zhang for their generous help in the research work.

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

Chinese).







**Figure Captions:**
**Fig. 1.** The Geographical location of the Otindag Desert in northern China. (a) The study area shown in a bigger scale, and (b) the study area shown in a smaller scale, with detailed information about the boundary and tectonic settings of the desert land. 1, the palaeo lake area of the megalake Dali; 2, the boundary of the Otindag; 3, the modern lake area; 4, the boundary of Fig. 2; 5, the boundary between the westerlies and the East Asian Summer Monsoon (EASM) climate systems. ①, the Xilamulun River. ②, the Gonggeer River. ③, the Shepi River. ④, the Tuligen River. The boundary between the westerlies and the EASM in (a) and (b) is modified from Chen et al. (2010). The palaeo lake area of the megalake Dali and the palaeo channel in (b) is modified from Yang et al (2015). The location of the Xar Moron Fault is referenced from Eizenhöfer et al. (2014). Section S1 is an elevation section starting from the upstream of the Dali Lake and ending at a spring sample (s2) in the riverhead of Xilamulun River.





**Fig. 2.** The locations of the water sampling sites in this study.

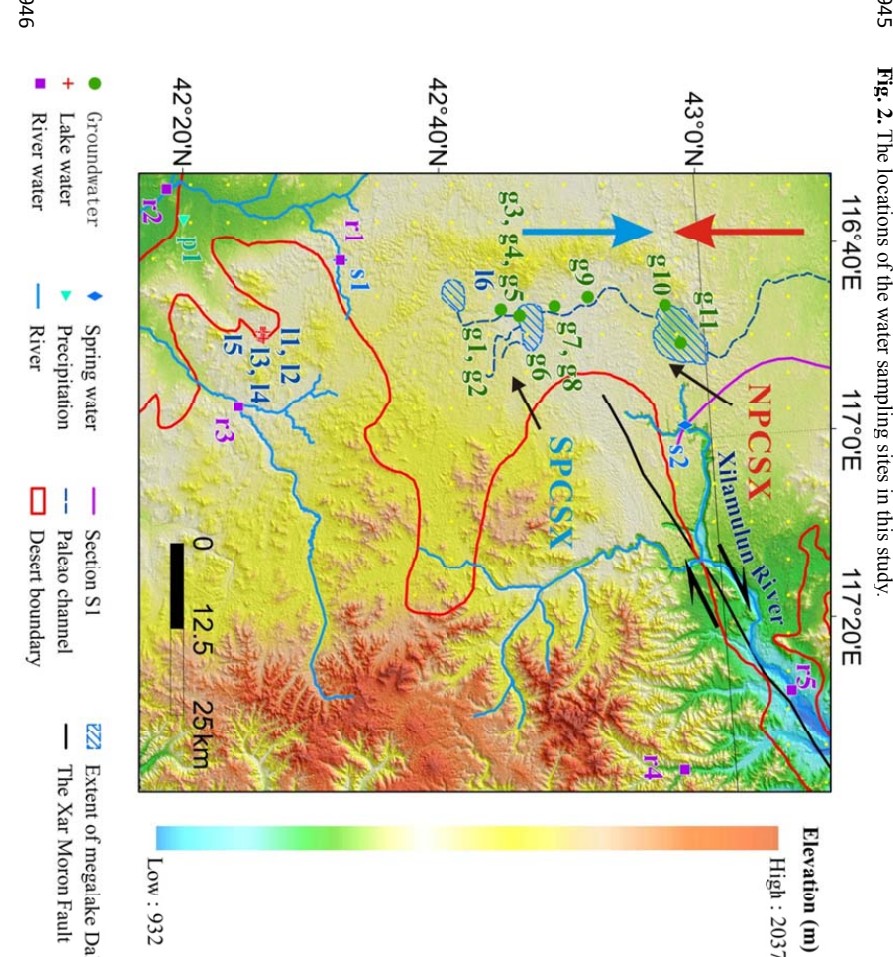






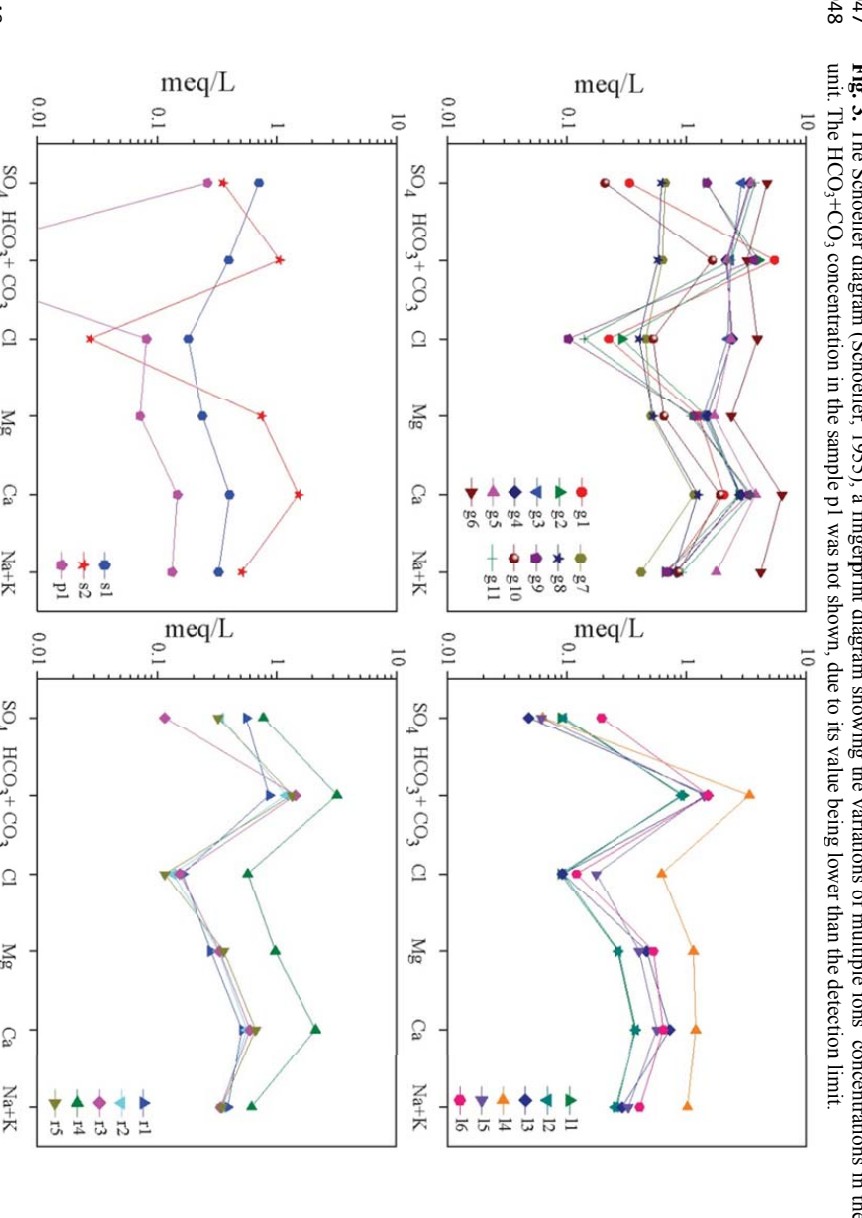

**Fig. 3.** The Schoeller diagram (Schoeller, 1955), a fingerprint diagram showing the variations of multiple ions' concentrations in the studied water samples in an equivalent unit. The $HCO_3+CO_3$ concentration in the sample p1 was not shown, due to its value being lower than the detection limit.





**Fig. 4.** The Piper diagram (Piper, 1944) showing the relative abundances of major cations and anions in the studied water samples. Major water types are also shown in this diagram.


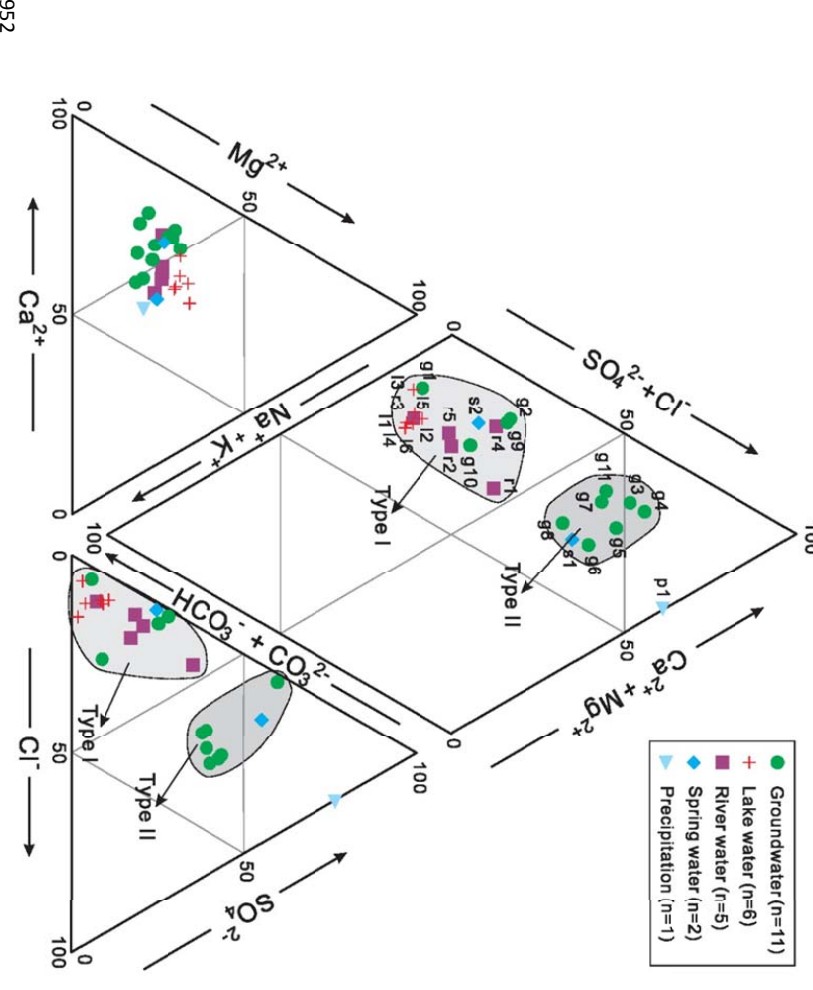





**Fig. 5.** An Expanded Durov diagram (Durov, 1948; Lioyd and Heathcote, 1985; Al-Bassam et al., 1997; Chadha, 1999; Al-Bassam and Khalil, 2012) showing the linear dissolution or mixing process for groundwater and the ion-exchange process occurred in the groundwater and other waters in the study area.

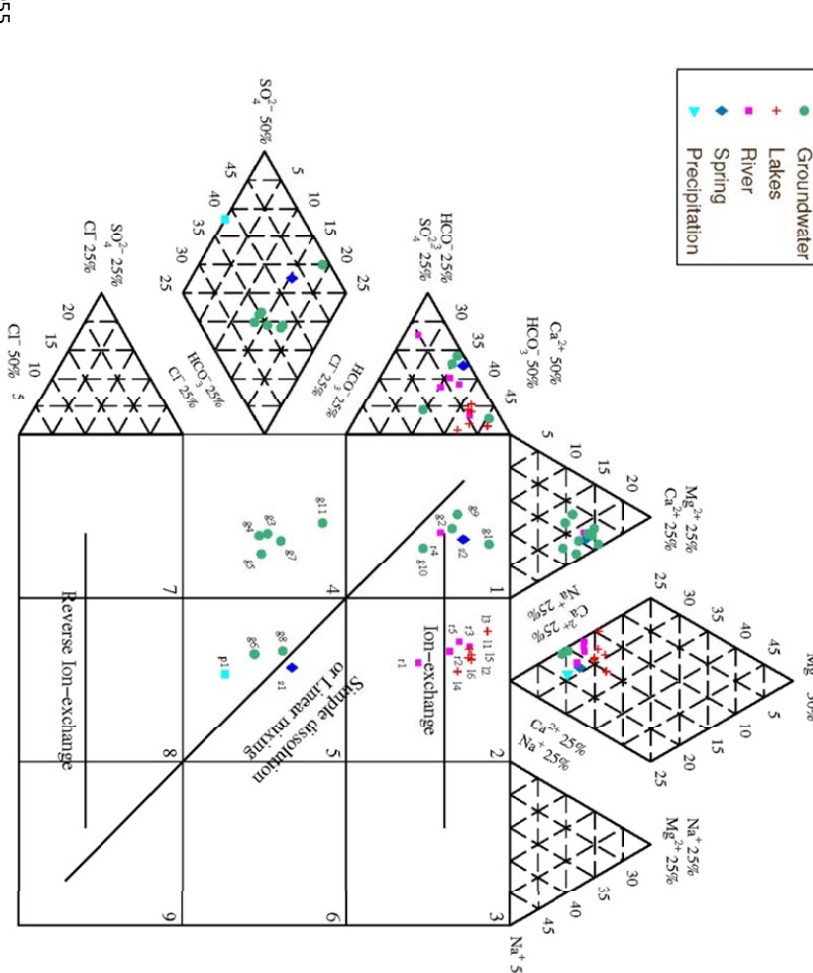

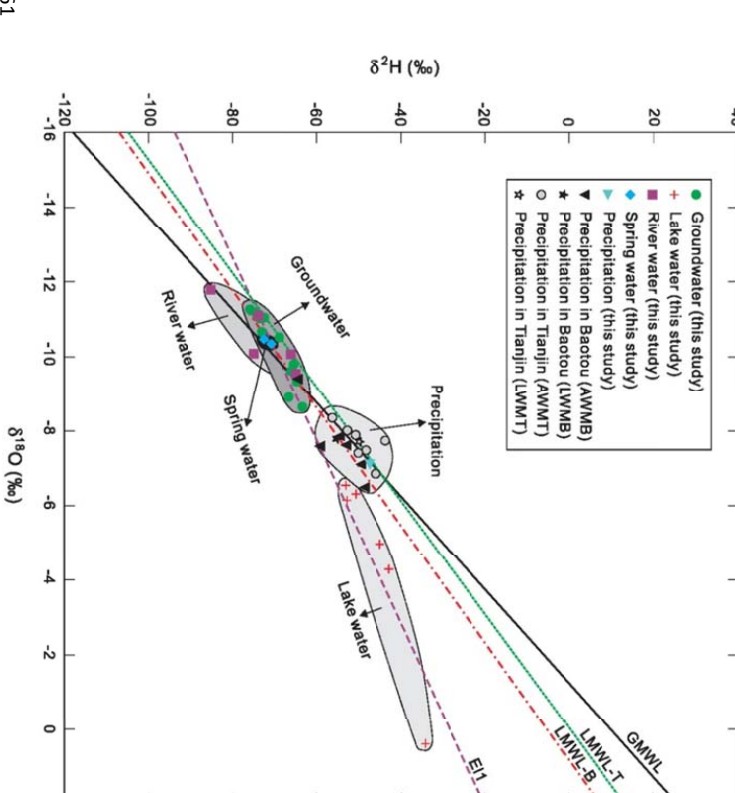

**Fig. 6.** The bivariate diagram of of δ²H and δ¹⁸O, i.e. the Craig diagram, for the natural water samples in this study. Different relationships between the groundwaters, lake waters, river waters, spring waters and the precipitation waters are emphasizedly illustrated. AWMB, the annual weighted mean value at the Baotou station; AWMT, the annual weighted mean value at the Tianjin station; LWMB, the long-term weighted means at the Baotou station; LWMT, the long-term weighted means at the Tianjin station; GMWL, the Global Meteoric Water Line; LMWL-B, the local meteoric water line calculated based on the data from the Baotou station; LMWL-T, the local meteoric water line calculated based on the data from the Tianjin station; ELl, the evaporation line calculated based on the data of water samples collected in the eastern Otindag.






**Fig. 7.** The seasonal mean distributions of precipitation (a), surface air temperature (b) and water vapor pressure (c) from the Baotou and Tianjin weather stations (station sites seen in **Fig. 1a**) in the surrounding areas of the Otindag in recent thirty years (1981-2010).






**Fig. 8.** The seasonal mean distributions of δ¹⁸O (a) and δ²H (b) values in precipitation from the Baotou and Tianjin weather stations in the surrounding areas of the Otindag in
recent sixteen years (1986-2001).


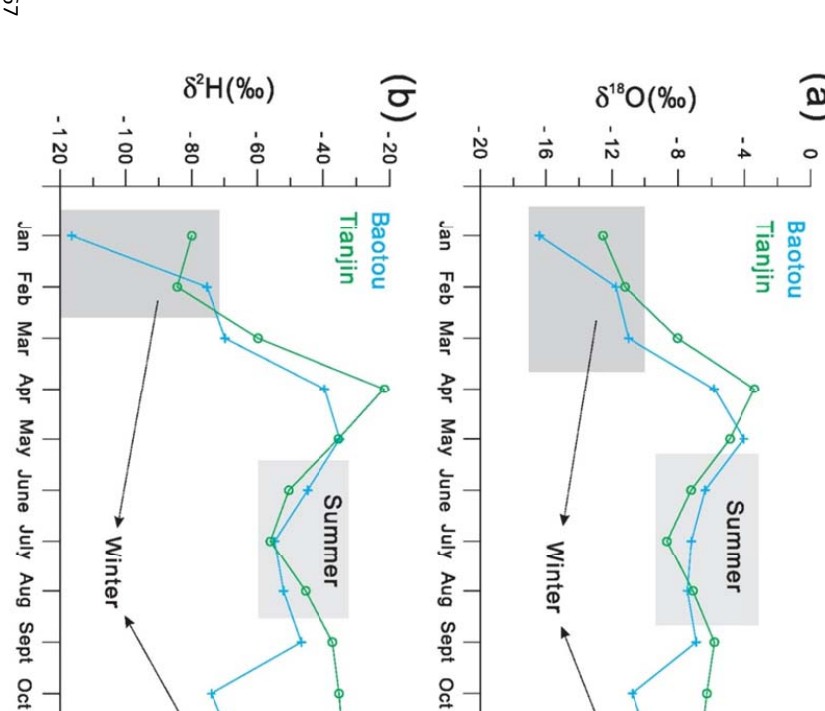

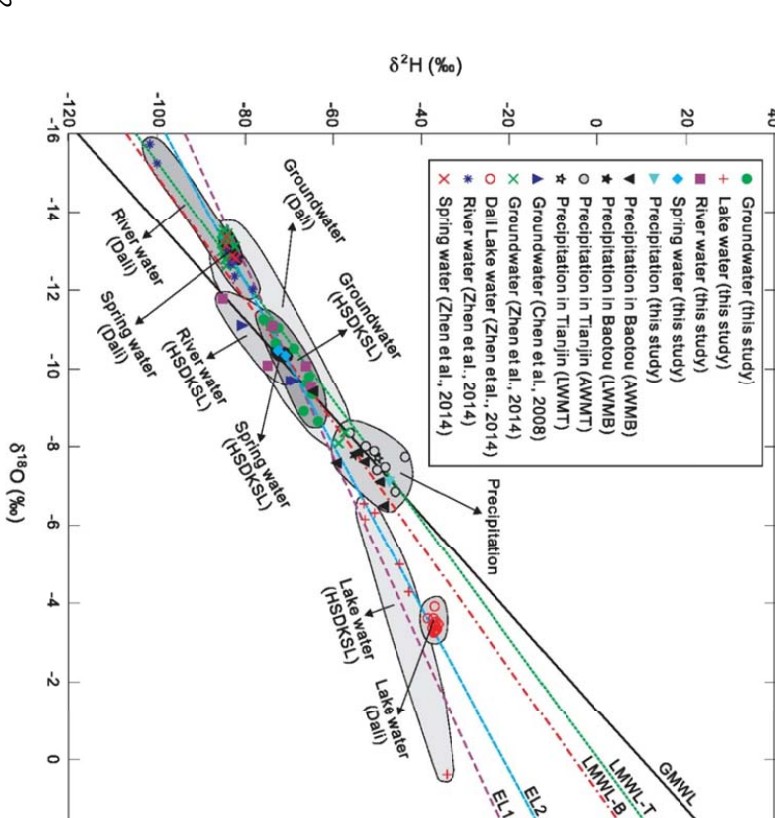

**Fig. 9.** The bivariate diagram of of δ²H and δ¹⁸O, i.e. the Craig diagram, for the natural water samples in this study. Different relationships between the groundwaters, lake waters, river waters, spring waters and the precipitation waters are emphasizedly illustrated. AWMB, AWMT, LWMB, LWMT, GMWL, LMWL-B, LMWL-T, and EL1 are same to **Fig. 6**. EL2, the evaporation line calculated based on the data from the groundwater, lake water, river water and spring water samples in the Otindag and in the Dali Basin. The data of the Dali are cited from previous studies (Chen et al., 2008; Zhen et al., 2014).



**Fig. 10.** (a) The sketch map showing the relationship between the groundwaters in the NPCSX and SPCSX areas, based on the chloride (a) and the TDS (b) concentrations of these water samples versus their distances away from the water sample g1 along the palaeo river channel (PCSX) from south to north.





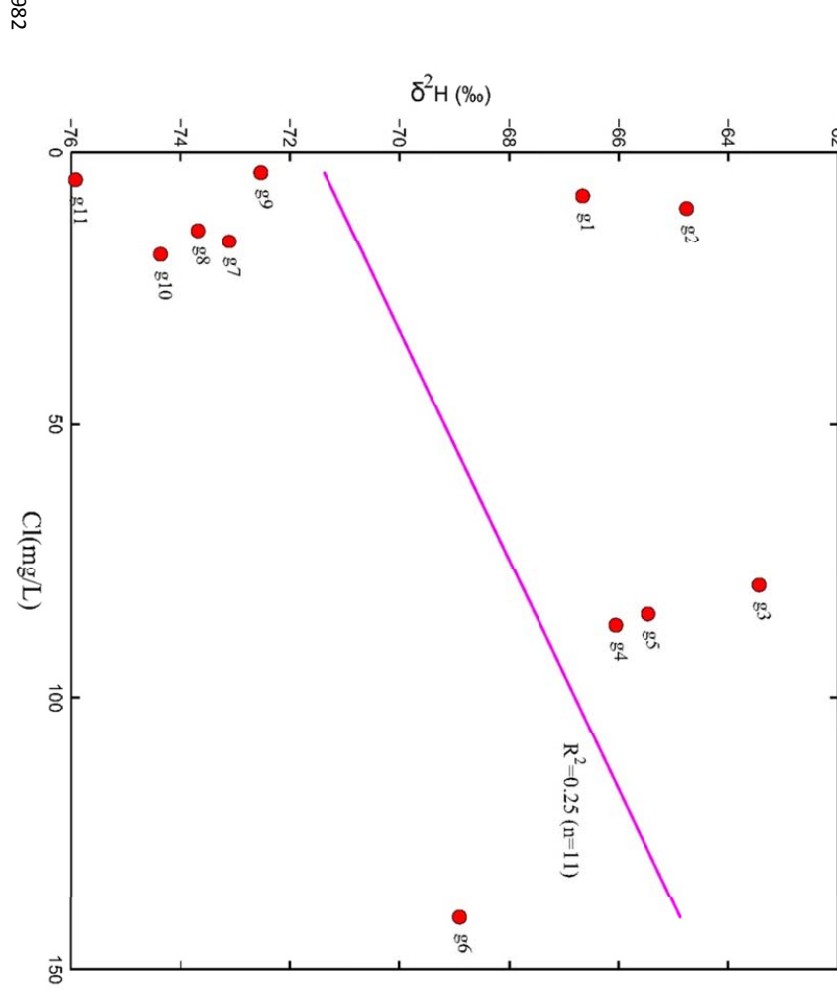

**Fig. 11.** The bivariate plot of Cl vs. $\delta^2$H in the groundwaters from the PCSX region, which showing that no significant evaporation process has been experienced by these groundwaters.





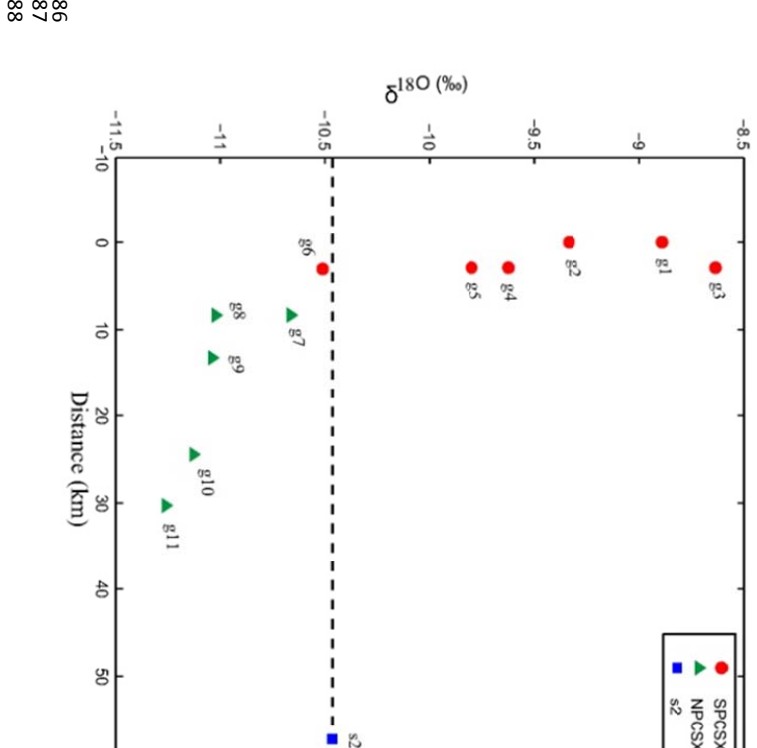

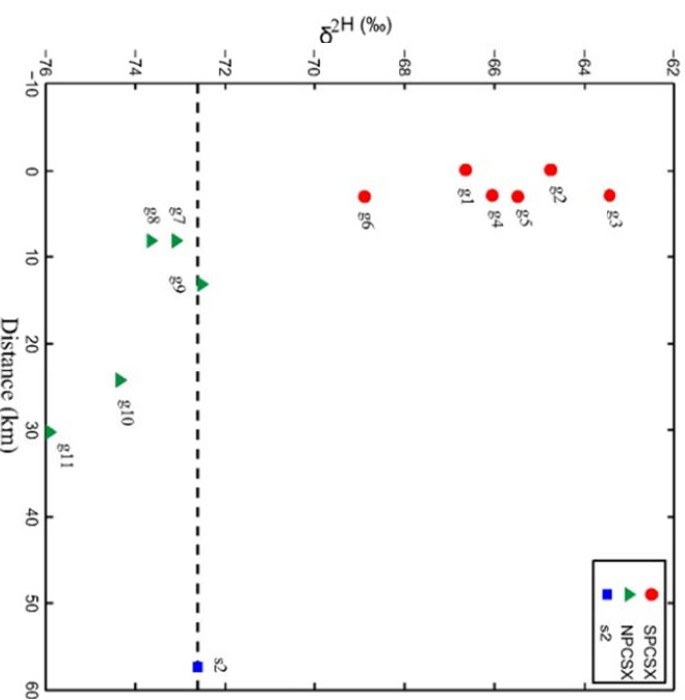

**Fig. 12.** Variations of $\delta^{18}O$ (a) and $\delta^2H$ (b) in the groundwaters versus their distances away from the groundwater sample g1 along the palaeo river channel (PCSX) from south to north. The dashed line represents the corresponding values of the spring water sample s2, which is just well divided the samples into the NPCSX and the SPCSX parts.






999
998
997

**Fig. 13.** The bivariate plots of $\delta^2$H (a) and $\delta^{18}$O (b) vs. deuterium excess for the groundwaters in the PCSX area.





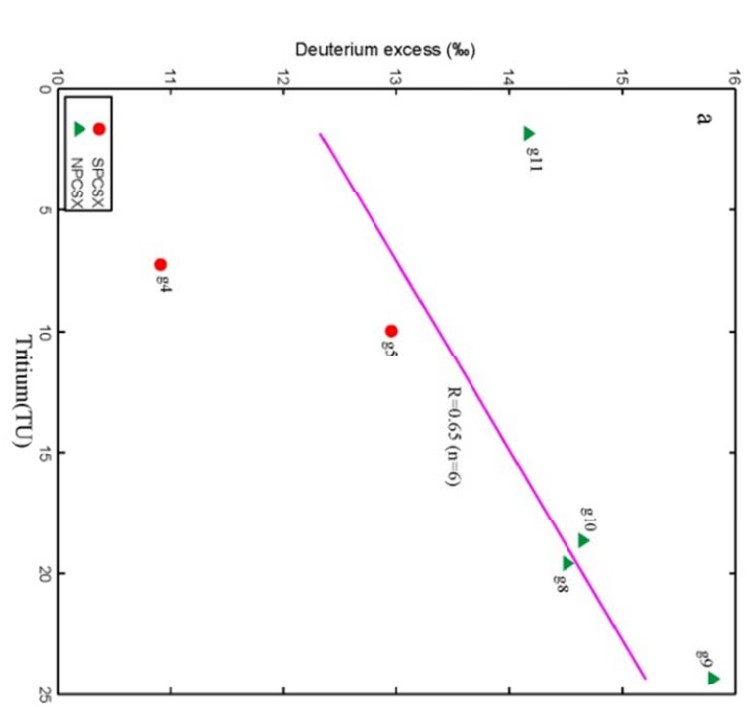

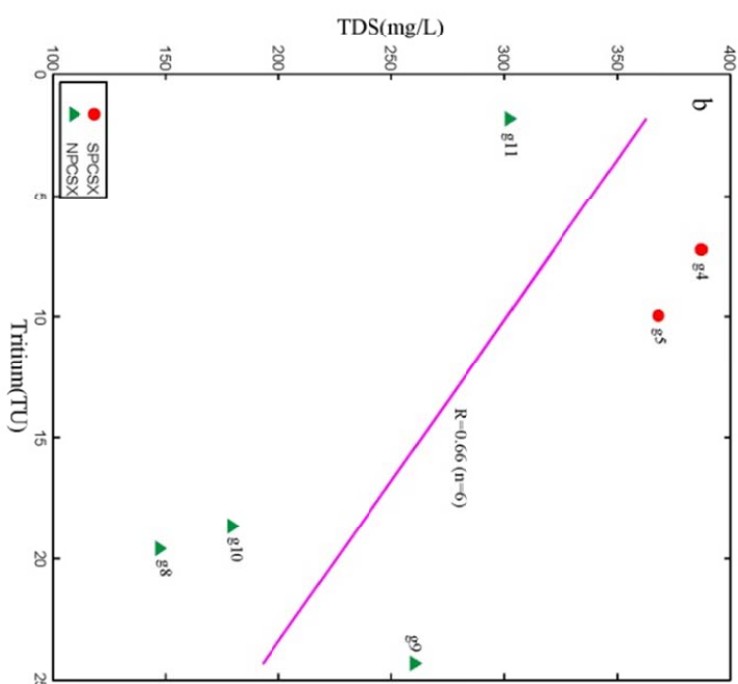

**Fig. 14.** Variations of tritium contents vs. deuterium excess (a) and TDS (b) for the groundwater samples in the study area. The sample g6 was excluded because of its potential contaminatedion.



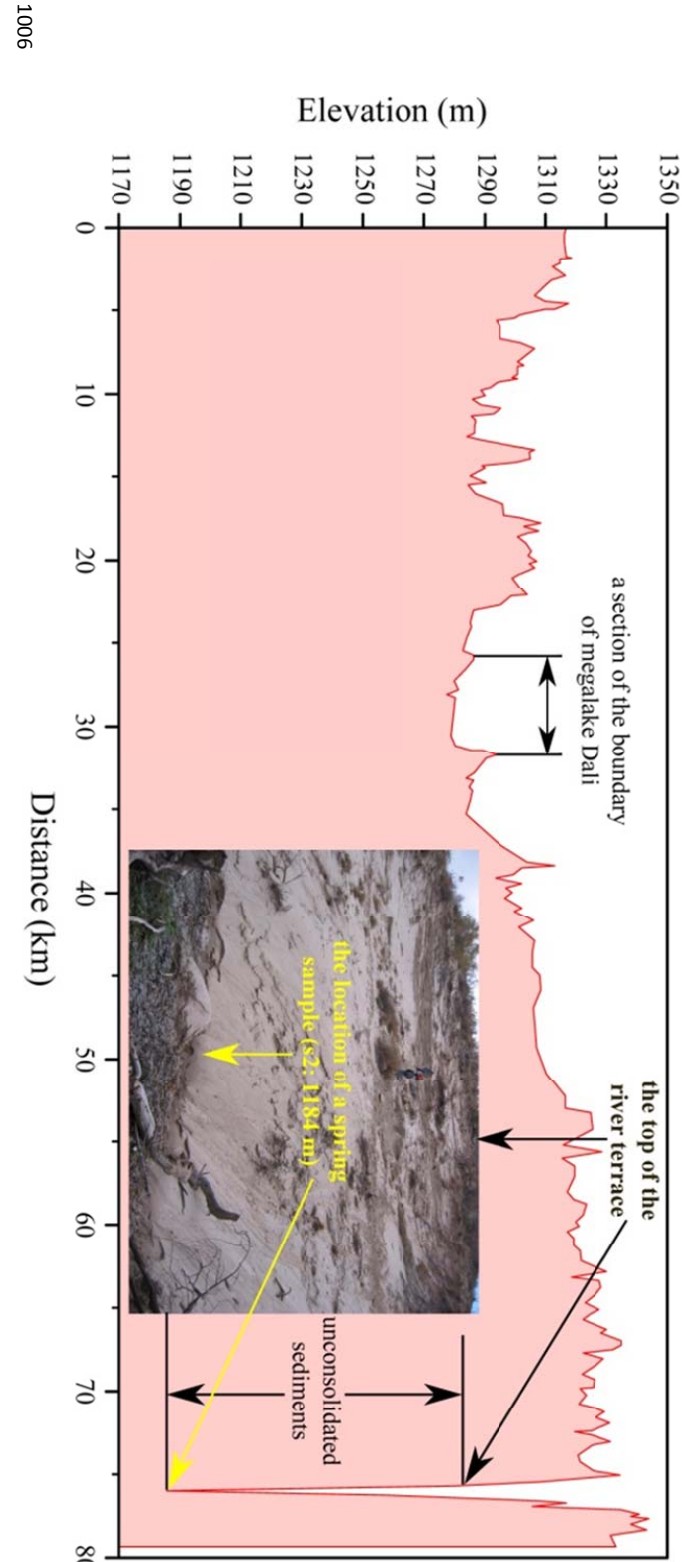

**Fig. 15.** Variation of the topographical elevation along the section S1 (see Fig.1b) from the upstream of the Dali Lake to the location site of the spring water sample (s2) in the riverhead of the Xilamulun River.



**Table Captions:**

**Table 1.** The physical parameters measured for the natural water samples in the study area.

| Sample ID | Water type | Latitude (N, degree) | Longitude (E, degree) | Elevation (m a.s.l) | Depth (m) | Temperature (°C) | pH | Eh (mV) | EC (µS/cm) | TDS (mg/L) | Salinity (%) | Alkalinity (meq/L) | Hardness (°dH) |
|---|---|---|---|---|---|---|---|---|---|---|---|---|---|
| g1 | Groundwater | 42.736306 | 116.747333 | 1396 | 12 | 5.8 | 6.72 | 3 | 769 | 410 | 0.6 | 5.47 | 9.42 |
| g2 | Groundwater | 42.736306 | 116.747333 | 1396 | 26 | 6.0 | 6.91 | -10 | 736 | 393 | 0.5 | 4.07 | 11.96 |
| g3 | Groundwater | 42.760194 | 116.760139 | 1355 | 32 | 7.7 | 6.88 | -6 | 725 | 384 | 0.5 | 2.39 | 11.94 |
| g4 | Groundwater | 42.759694 | 116.760417 | 1360 | 7 | 10.0 | 6.74 | 1 | 725 | 387 | 0.5 | 2.20 | 12.28 |
| g5 | Groundwater | 42.759556 | 116.760556 | 1362 | 27 | 7.6 | 6.46 | 16 | 691 | 368 | 0.5 | 2.23 | 15.57 |
| g6 | Groundwater | 42.760111 | 116.760250 | 1365 | 7 | 10.3 | 6.26 | 22 | 1240 | 660 | 0.8 | 3.25 | 24.45 |
| g7 | Groundwater | 42.806361 | 116.747806 | 1352 | 20 | 6.8 | 6.71 | 2 | 297 | 158 | 0.2 | 0.63 | 4.70 |
| g8 | Groundwater | 42.806361 | 116.747806 | 1352 | 16 | 6.5 | 6.92 | -8 | 276 | 147 | 0.2 | 0.58 | 5.00 |
| g9 | Groundwater | 42.850333 | 116.735722 | 1347 | 30 | 7.2 | 6.74 | -1 | 487 | 260 | 0.4 | 3.73 | 12.68 |
| g10 | Groundwater | 42.949861 | 116.759194 | 1321 | 37 | 9.9 | 6.75 | -2 | 337 | 179 | 0.2 | 1.66 | 7.23 |
| g11 | Groundwater | 42.967111 | 116.827528 | 1317 | 60 | 8.6 | 6.99 | -14 | 571 | 302 | 0.4 | 2.40 | 12.94 |
| l1 | Lake water | 42.424611 | 116.769194 | 1368 | / | 16.9 | 9.44 | -151 | 126 | 67 | 0.1 | 0.95 | 1.79 |
| l2 | Lake water | 42.424611 | 116.769194 | 1368 | / | 19.6 | 9.18 | -137 | 132 | 70 | 0.1 | 0.92 | 1.82 |
| l3 | Lake water | 42.424611 | 116.757806 | 1365 | / | 20.2 | 7.38 | -36 | 196 | 105 | 0.1 | 1.53 | 3.36 |
| l4 | Lake water | 42.427083 | 116.757639 | 1366 | / | 20.5 | 7.87 | -64 | 448 | 238 | 0.2 | 3.42 | 6.61 |
| l5 | Lake water | 42.421806 | 116.756917 | 1360 | / | 20.1 | 8.23 | -83 | 173 | 92 | 0.1 | 1.43 | 2.73 |
| l6 | Lake water | 42.736389 | 116.747222 | 1374 | / | 10.7 | 8.35 | -89 | 194 | 103 | 0.1 | 1.53 | 3.30 |
| r1 | River water | 42.530917 | 116.641250 | 1355 | / | 20.6 | 7.31 | -33 | 180 | 96 | 0.1 | 0.88 | 2.23 |
| r2 | River water | 42.310883 | 116.494817 | 1231 | / | 14.9 | 7.67 | -52 | 178 | 95 | 0.1 | 1.21 | 2.50 |
| r3 | River water | 42.385778 | 116.886194 | 1362 | / | 9.5 | 7.62 | -48 | 177 | 94 | 0.1 | 1.45 | 2.62 |
| r4 | River water | 42.931417 | 117.585306 | 1217 | / | 10.5 | 7.97 | -69 | 474 | 252 | 0.3 | 3.22 | 8.73 |
| r5 | Lake water | 43.079083 | 117.457389 | 1006 | / | 12.9 | 7.87 | -62 | 191 | 101 | 0.1 | 1.37 | 2.88 |
| s1 | Spring water | 42.530917 | 116.641250 | 1359 | / | 20.9 | 6.63 | 5 | 165 | 88 | 0.1 | 0.40 | 1.81 |
| s2 | Spring water | 42.965417 | 116.975361 | 1184 | / | 19.0 | 7.47 | -46 | 371 | 195 | 0.2 | 1.07 | 6.40 |
| p1 | Precipitation | 42.330750 | 116.551694 | 1260 | / | 20.2 | 4.61 | 109 | 78 | 42 | 0.0 | / | 0.61 |





**Table 2.** The concentrations of major cations and anions measured for the water samples in the study area.

| Sample | F⁻ (mg/L) | Cl⁻ (mg/L) | NO₂⁻ (mg/L) | NO₃⁻ (mg/L) | SO₄²⁻ (mg/L) | CO₃²⁻ (mg/L) | HCO₃⁻ (mg/L) | Li⁺ (mg/L) | Na⁺ (mg/L) | NH₄⁺ (mg/L) | K⁺ (mg/L) | Mg²⁺ (mg/L) | Ca²⁺ (mg/L) |
|---|---|---|---|---|---|---|---|---|---|---|---|---|---|
| g1 | 0.13 | 7.90 | 2.32 | 0.48 | 16.10 | 0.00 | 334.60 | 0.02 | 13.79 | 10.54 | 4.59 | 15.52 | 41.81 |
| g2 | 0.21 | 10.21 | 0.00 | 6.15 | 70.61 | 0.10 | 247.70 | 0.02 | 13.36 | 6.56 | 3.45 | 17.91 | 56.04 |
| g3 | 0.11 | 79.56 | 0.00 | 0.00 | 140.76 | 0.00 | 145.40 | 0.01 | 17.92 | 2.28 | 1.76 | 17.06 | 57.29 |
| g4 | 0.10 | 86.90 | 0.00 | 5.73 | 164.80 | 0.00 | 133.70 | 0.02 | 18.02 | 0.00 | 2.02 | 18.50 | 57.32 |
| g5 | 0.07 | 84.82 | 0.00 | 0.76 | 169.30 | 0.00 | 136.20 | 0.00 | 39.68 | 1.02 | 2.72 | 20.94 | 76.86 |
| g6 | 0.07 | 140.54 | 0.00 | 0.00 | 228.80 | 0.00 | 198.20 | 0.00 | 79.80 | 0.00 | 29.47 | 29.25 | 126.68 |
| g7 | 0.37 | 16.31 | 0.00 | 0.00 | 32.01 | 0.00 | 38.70 | 0.06 | 7.83 | 0.00 | 3.09 | 6.21 | 23.37 |
| g8 | 0.29 | 14.28 | 0.00 | 0.00 | 29.89 | 0.00 | 35.50 | 0.02 | 16.21 | 0.11 | 3.38 | 6.44 | 25.14 |
| g9 | 0.10 | 3.66 | 0.15 | 1.19 | 71.56 | 0.00 | 227.40 | 0.06 | 12.92 | 0.55 | 4.50 | 14.06 | 67.52 |
| g10 | 0.24 | 18.80 | 0.00 | 49.49 | 9.97 | 0.00 | 101.10 | 0.00 | 18.54 | 0.00 | 2.09 | 7.92 | 38.68 |
| g11 | 0.28 | 4.94 | 0.00 | 0.00 | 181.53 | 0.00 | 146.20 | 0.05 | 20.40 | 2.59 | 2.06 | 13.30 | 70.59 |
| l1 | 0.16 | 3.15 | 0.00 | 0.07 | 4.32 | 0.00 | 57.90 | 0.01 | 5.42 | 0.00 | 0.86 | 3.24 | 7.49 |
| l2 | 0.16 | 3.30 | 0.00 | 1.66 | 4.57 | 0.00 | 55.80 | 0.00 | 5.33 | 0.00 | 0.84 | 3.29 | 7.61 |
| l3 | 0.11 | 3.27 | 0.00 | 0.61 | 2.33 | 0.00 | 93.30 | 0.01 | 5.88 | 0.00 | 1.19 | 5.68 | 14.66 |
| l4 | 0.17 | 22.12 | 0.00 | 0.39 | 3.04 | 0.10 | 207.60 | 0.00 | 9.21 | 0.70 | 24.21 | 14.02 | 24.18 |
| l5 | 0.09 | 6.24 | 0.00 | 0.65 | 2.97 | 0.10 | 86.80 | 0.01 | 6.72 | 0.00 | 1.16 | 4.91 | 11.41 |
| l6 | 0.18 | 4.29 | 0.00 | 0.80 | 9.34 | 0.10 | 93.00 | 0.01 | 8.41 | 0.00 | 1.36 | 6.47 | 12.95 |
| r1 | 0.30 | 5.76 | 0.00 | 2.38 | 26.67 | 0.30 | 52.40 | 0.01 | 7.15 | 0.00 | 2.99 | 3.41 | 10.34 |
| r2 | 0.19 | 4.82 | 0.00 | 0.65 | 16.40 | 0.10 | 73.10 | 0.01 | 6.82 | 0.00 | 1.92 | 3.96 | 11.36 |
| r3 | 0.64 | 5.46 | 0.00 | 0.43 | 5.57 | 0.00 | 88.10 | 0.01 | 7.11 | 0.00 | 1.13 | 4.04 | 12.06 |
| r4 | 1.08 | 20.39 | 0.00 | 19.27 | 37.25 | 0.50 | 195.00 | 0.01 | 13.02 | 0.00 | 1.96 | 11.90 | 42.81 |
| r5 | 0.19 | 4.10 | 0.00 | 1.08 | 15.57 | 0.00 | 82.60 | 0.01 | 6.71 | 0.00 | 2.08 | 4.38 | 13.40 |
| s1 | 0.16 | 6.44 | 0.00 | 1.95 | 34.25 | 0.00 | 24.30 | 0.02 | 6.56 | 0.00 | 1.62 | 2.92 | 8.10 |
| s2 | 0.05 | 0.98 | 0.00 | 0.45 | 17.15 | 0.00 | 64.90 | 0.02 | 9.87 | 0.00 | 3.32 | 9.10 | 30.79 |
| p1 | 0.61 | 2.90 | 0.00 | 9.46 | 12.65 | 0.00 | 0.00 | 0.00 | 2.09 | 2.07 | 1.64 | 0.88 | 2.95 |





**Table 3.** The analytical data of stable and radioactive isotopes measured for the water samples in this study.

| Sample ID | $\delta^2H$ (‰) | σ‰ | $\delta^{18}O$ (‰) | σ‰ | deuterium excess (d) | Tritium ($^3H$)(TU) |
|---|---|---|---|---|---|---|
| g1 | -66.664 | 0.199 | -8.895 | 0.026 | 4.496 | / |
| g2 | -64.758 | 0.291 | -9.336 | 0.039 | 9.930 | / |
| g3 | -63.424 | 0.269 | -8.635 | 0.008 | 5.656 | / |
| g4 | -66.055 | 0.149 | -9.621 | 0.062 | 10.913 | 7.250 |
| g5 | -65.462 | 0.111 | -9.802 | 0.027 | 12.954 | 9.975 |
| g6 | -68.913 | 0.287 | -10.514 | 0.039 | 15.199 | 22.908 |
| g7 | -73.105 | 0.298 | -10.662 | 0.041 | 12.191 | / |
| g8 | -73.676 | 0.220 | -11.023 | 0.037 | 14.508 | 19.611 |
| g9 | -72.530 | 0.181 | -11.041 | 0.015 | 15.798 | 24.345 |
| g10 | -74.362 | 0.201 | -11.127 | 0.026 | 14.654 | 18.681 |
| g11 | -75.924 | 0.340 | -11.260 | 0.015 | 14.156 | 1.860 |
| l1 | -53.128 | 0.229 | -6.553 | 0.002 | -0.704 | / |
| l2 | -50.721 | 0.304 | -6.320 | 0.026 | -0.161 | / |
| l3 | -42.877 | 0.239 | -4.292 | 0.034 | -8.545 | / |
| l4 | -34.155 | 0.243 | 0.381 | 0.040 | -37.203 | / |
| l5 | -45.057 | 0.206 | -4.987 | 0.009 | -5.161 | / |
| l6 | -52.866 | 0.187 | -6.150 | 0.049 | -3.666 | / |
| r1 | -66.157 | 0.118 | -10.069 | 0.015 | 14.395 | / |
| r2 | -64.996 | 0.148 | -9.549 | 0.012 | 11.396 | / |
| r3 | -73.790 | 0.315 | -11.083 | 0.021 | 14.874 | / |
| r4 | -85.155 | 0.244 | -11.781 | 0.005 | 9.093 | / |
| r5 | -74.978 | 0.195 | -10.084 | 0.003 | 5.694 | / |
| s1 | -70.832 | 0.074 | -10.340 | 0.007 | 11.888 | / |
| s2 | -72.601 | 0.281 | -10.468 | 0.046 | 11.143 | / |
| p1 | -47.435 | 0.374 | -7.141 | 0.017 | 9.693 | / |




**Table 4.** The statistical frequency of rainfall events being >20 mm per year during the recent 30 years from 1985 to 2014. The data come from the China Meteorological Data Sharing Service System.

| Station | One time/year | Two times/year | Three times/year | Four times/year | Five times/year | Six times/year | Seven times/year | Mean times/year |
|---|---|---|---|---|---|---|---|---|
| Duolun | 2 | 8 | 8 | 4 | 4 | 3 | 1 | 3.4 |
| Xilinhaote | 8 | 5 | 2 | 6 | 3 | 2 | 0 | 2.5 |

**Table 5.** The measured contents of tritium in the groundwater samples studied and the calculated ages of these samples.

| Sample-ID | Tritium content (T.U.) | Possible ages (years) |
|---|---|---|
| g1 | not measured | not clear |
| g2 | not measured | not clear |
| g3 | not measured | not clear |
| g4 | 7.25 | 20–40 |
| g5 | 9.97 | 13–33 |
| g6 | 22.91 | 0–20 |
| g7 | not measured | not clear |
| g8 | 19.61 | 0–20 |
| g9 | 24.34 | 0–17 |
| g10 | 18.68 | 0–22 |
| g11 | 1.86 | 40–65 |