# Peer review of "Direct or indirect recharge on groundwater in the middle-latitude desert of Otindag, China?"

_Hydrology and Earth System Sciences, 2018_

## Referee Comment (RC1) · Anonymous Referee #1 · 22 May 2018

The manuscript describes interesting results about the recharge mechanisms of arid zones in China, especially considering the importance of the topic. Despite the multi-disciplinary approach, which is very useful in groundwater recharge studies, there are many weak points which have to be improved for a publication in HESS. The main points are listed below: 1) The datasets belong to sampling campaigns carried out in different moments (years) and seasons and for this reason in my opinion cannot be discussed together, without a clear distinction between the different phases. 2) A re-construction of the piezometric morphology as well as a stratigraphy of the considered study areas should be reported. This could help also the discussion of the groundwater preferential pathways. 3) The organization of the paper is still at a draft level, since

there is not a clear distinction between the results and discussion paragraphs. Many paragraphs need to be summarized and better explained. 4) The number of figures should be reduced (probably putting together some and deleting others). 5) The English is very poor and there are many typo errors. The reported delta notation is wrong. Due to the consideration of these main points the manuscript can be accepted only if major revision will be reported.

―――――――――――――――――

---

## Referee Comment (RC2) · Anonymous Referee #2 · 6 Jun 2018

Groundwater availability in arid and semi-arid regions is one of the key issues in hydrogeology and is becoming even more important because of the expected climate changes. Within this context, the contribution by Zhu and Ren provides an interesting analysis on the possible recharge supporting the availability of significant groundwater resources in the Otindag desert, north-eastern China. The analyses have been carried out using hydrogeochemical tracers and isotopic measurements on water samples collected from groundwater, surficial (river, lake, and spring) waters, and precipitation water, as well as in-situ records of temperature, pH, conductivity, and TDS concentration. The various steps implemented by the authors to reject possible hypotheses on the groundwater origin (e.g., water flowing from another nearby arid area, precipitation,

paleo-water resources) are presented in detail and discussed. Zhu and Ren concludes that, based on the available evidences, the groundwater resources in this region are recharged by the leakage through the bed on incise rivers bounding the desert to the east and conveying downward the waters originated from the precipitation on Daxin-ganling Ranges. Hence, an "indirect" recharge is the main mechanism supporting the water availability in the study arid lands.

Two are the main weaknesses of this ms: 1) the chemical/isotopic investigations seem not supported by a (at least minimum) knowledge of the hydrogeological setting. This is likely one of the reasons why the analyses carried out by the authors are mainly able to exclude recharge mechanisms, but not definitely explain from where this water is originated. The last part of Section 5.5 provides a list of speculative mechanisms (lines 614-652): how the Xilamulun river can recharge the Dali lake when Fig. 15 shows that the bed of the former is less elevated than that of the latter? What support the "speculation" about the "flash floods" in the southern portion of the desert? How you only "theoretically estimate" the isotopic firm of the precipitation on the Yinshan Ranges? 2) the contribution is over-long. The introduction addresses the topic with a too-wide perspective, concepts are repeated, with verbose descriptions. There are also too many figures that can be fruitfully combined. The English form must be improved too.

Moreover, the location of the study area is unclear: Fig 1a is obscure, the various portions of the desert are not provided in the maps shown in Figs. 1b and 2, a large part of the toponymy cited in the text is not added to the maps.

Because of this, the ms need a major revision.

---

## Author Comment (AC1) · 30 Jun 2018

Comments from Referee #1 Interactive comment on "Direct or indirect recharge on groundwater in the middle-latitude desert of Otindag, China?" by Bing-Qi Zhu and Xiao-Zong Ren, Anonymous Referee #1, The manuscript describes interesting results about the recharge mechanisms of arid zones in China, especially considering the importance of the topic. Despite the multidisciplinary approach, which is very useful in groundwater recharge studies, there are many weak points which have to be improved for a publication in HESS. The main points are listed below: 1) The datasets belong to sampling campaigns carried out in different moments (years) and seasons and for this reason in my opinion cannot be discussed together, without a clear distinction between the different phases. 2) A reconstruction of the piezometric morphology as well as a stratigraphy of the considered study areas should be reported. This could help also the discussion of the groundwater preferential pathways. 3) The organization of the paper is still at a draft level, since there is not a clear distinction between the results and discussion paragraphs. Many paragraphs need to be summarized and better explained. 4) The number of figures should be reduced (probably putting together some and deleting others). 5) The English is very poor and there are many typo errors. The reported delta notation is wrong. Due to the consideration of these main points the manuscript can be accepted only if major revision will be reported. Interactive comment on Hydrol. Earth Syst. Sci. Discuss., https://doi.org/10.5194/hess-2018-71, 2018.

The authors' responses to the comments from Referee #1

Dear Dr/Professor Referee #1: On behalf of my co-authors, we thank you very much for giving us an opportunity to revise our manuscript. We appreciate you very much for your positive and constructive comments and suggestions on our manuscript (hess-2018-71). We have studied your comments carefully and have made revision which marked in red in the revised manuscript. We tried our best to revise our manuscript according to the comments point by point. Attached please find the revised version, which we would like to submit for your kind consideration. Thank you and best regards.

1) The datasets belong to sampling campaigns carried out in different moments (years) and seasons and for this reason in my opinion cannot be discussed together, without a clear distinction between the different phases. Our response: AGREE AND NO CHANGES MADE. Firstly, we thank you very much for this comment from you and we truly agree this point that water samples collected in different moments (years) and seasons cannot be discussed together without a clear distinction between the different water phases. In fact, although we stated in the manuscript that our fieldwork had taken place during the summer season of 2011 and the spring season of 2012, we collected the natural water samples at the same time for the same phases in the study area. For example, (1) all the groundwater samples discussed in this paper were collected during the 2011 summer in five days in the Otindag Desert. For other natural water samples discussed in this study, the detailed sampling methods are as follow: (2) all the spring water samples and (3) the precipitation water sample (p1) discussed in this paper were also collected during the 2011 summer in five days in the study area, and (4) all the river water samples and (5) lake water samples were collected during the spring season of 2012 in three days in the study area. This is to say that the water samples within the same phase are discussed together in the paper.

2) A reconstruction of the piezometric morphology as well as a stratigraphy of the considered study areas should be reported. This could help also the discussion of the groundwater preferential pathways. Our response: AGREE AND CHANGES MADE. We thank you very much for this comment. And yes, according to this comment, we revised the manuscript and focused on reporting the geological (tectonic, lithological, sedimentological and structural), geomorphological, hydrogeological and stratigraphical settings of the study area. Please see the section 2 "Regional setting" of the revised manuscript in its pages 2-4 lines 103-188.

3) The organization of the paper is still at a draft level, since there is not a clear distinction between the results and discussion paragraphs. Many paragraphs need to be summarized and better explained. Our response: AGREE AND CHANGES MADE. We thank the you very much for this comment. And yes, we have revised the manuscript accordingly. The structure and content of the paper has been thoroughly reorganized in the revised manuscript, especially for the results and discussion sections, to make the content and context of the paper being more logic, coherent and readable. And yes, almost all of the paragraphs in the paper are newly summarized and explained. The detailed changes can be easily observed in the revised manuscript by reading one of the two resubmitted MS-Word files with the "changes marked" version (in contrast, another version is "clear copy").

4) The number of figures should be reduced (probably putting together some and deleting others). Our response: AGREE AND CHANGES MADE. We thank you very much for this comment. And yes, we have revised the manuscript accordingly. We reduced the number of figures in the revised manuscript by putting some figures together and deleting several figures. At last the revised manuscript has 11 figures compared with the original manuscript that including 15 figures. For example, the Figs. 5, 11, 13, 14a in the original manuscript are deleted in the revised manuscript, and the Figs. 7 and 8, the Figs. 10, 12 and 14a are combined, respectively. In addition, two newly-built figures are added into the revised manuscript according to the second comment from you (the detailed content of this comment can be seen above). The specific changes and the final results of these figures can be seen in the newly submitted revised manuscript.

5) The English is very poor and there are many typo errors. The reported delta notation is wrong. Our response: AGREE AND CHANGES MADE. We thank you very much for this comment. We are very sorry for our poor and incorrect English writing in the original manuscript. For the shortcomings of the English presentation and the grammatical edit in the first paper, we have checked and revised the whole manuscript carefully to avoid language errors, and finally we have got the help of a native English speaking professional to check and improve the English quality of the revised manuscript. We believe that the language is now acceptable for the publishing purpose. In addition, the wrong use of the dalta notation in the original manuscript, such as $\delta$2H, has been corrected as "$\delta$D" in the revised manuscript.

6) Due to the consideration of these main points the manuscript can be accepted only if major revision will be reported. Our response: AGREE AND CHANGES MADE. Special thanks to you for your good comments. We have tried our best to improve the manuscript and made specific changes in the revised manuscript according to the comments from you one by one. These changes will not influence the content and framework of the paper. And here we did not list the changes but marked in red in the revised paper. We hope that the correction will meet with approval. Once again, thank

you very much for your comments and suggestions.

Please also note the supplement to this comment:
https://www.hydrol-earth-syst-sci-discuss.net/hess-2018-71/hess-2018-71-AC1-
supplement.pdf

―――――――――――――――――

[Figure]

The map figure shows geographical coordinates and features.

Legend:
- Groundwater
- Lake water
- River water
- Spring water
- Precipitation
- The Xar Moron Fault
- Section S1
- Palaeo channel
- River
- City
- 1
- 2
- 3
- 4
- 5

Elevation (m)
High : 2042
Low : 597

**Fig. 1.** Fig. 1. The Geographical location of the Otindag Desert in northern China. (a) The study area shown at a large scale, and (b) the study area shown at a smaller scale, with detailed information about t

**Fig. 2.** Fig. 2. (a) Tectonic framework of the north China-Mongolian segment of the Central Asian Orogenic Belt (modified after Jahn, 2004). (b) Geological sketch map of the northern China-Mongolia tract (modi

[Figure]

**Fig. 3.** Fig. 3. The hydrogeological division map of the Otindag Desert.

Map legend:

Pore water-bearing aquifer in unconsolidated rocks
- The watery degree of aquifer: very weak
- The watery degree of aquifer: weak
- The watery degree of aquifer: strong

Pore/fracture water-bearing aquifer in clastic rocks (aquifer in clastic rocks)
- The watery degree of aquifer: weak
- The watery degree of aquifer: medium
- The watery degree of aquifer: strong

Fracture water-bearing aquifer in magmatic rocks (aquifer in intrusive rocks)
- The watery degree of aquifer: weak

Fracture water-bearing aquifer in magmatic rocks (aquifer in extrusive rocks)
- The watery degree of aquifer: weak
- The watery degree of aquifer: medium

Fracture water-bearing aquifer in metamorphic rocks
- The watery degree of aquifer: weak

- Sandy land boundaries
- Lake
- City
- < 10  Depth of unconfined groundwater (m)

**Fig. 4.** Fig. 4. The locations of the water sampling sites in this study.

[Figure]

**Fig. 5.** Fig. 5. The fingerprint diagram showing the variations of multiple ions' concentrations in the studied water samples in an equivalent unit. The HCO3+CO3 concentration in the sample p1 was not shown, d

[Figure]

**Fig. 6.** Fig. 6. The Piper diagram showing the relative abundances of major cations and anions in the studied water samples. Major water types are also shown in this diagram.

[Figure]

**Fig. 7.** Fig. 7. The bivariate diagram of δD and δ18O, i.e. the Craig diagram, for the natural water samples in this study. Different relationships between the groundwaters, lake waters, river waters, spring w

**Fig. 8.** Fig. 8. The seasonal mean distributions of (a) precipitation, (b) surface air temperature and (c) water vapor pressure from the Baotou and Tianjin weather stations (station sites seen in Fig. 1a) in t

[Figure]

**Fig. 9.** Fig. 9. The bivariate diagram of $\delta D$ and $\delta 18O$, i.e. the Craig diagram, for the natural water samples collected in the Otindag (this study) and the Dali Basin. Different relationships between the ground

[Figure]

**Fig. 10.** Fig. 10. (a) Sketch map showing the relationship between the groundwaters in the NPCSX and SPCSX areas, based on variations of (a) the chloride concentrations, (b) the TDS concentrations, (c) the $\delta$18O

[Figure]

**Fig. 11.** Fig. 11. Variation of the topographical elevation along the section S1 (see Fig. 1b) from the upstream of the Dali Lake to the location site of the spring water sample (s2) in the riverhead of the Xil

**Supplement:**

The manuscript describes interesting results about the recharge mechanisms of arid zones in China, especially considering the importance of the topic. Despite the multidisciplinary approach, which is very useful in groundwater recharge studies, there are many weak points which have to be improved for a publication in HESS. The main points are listed below: 1) The datasets belong to sampling campaigns carried out in different moments (years) and seasons and for this reason in my opinion cannot be discussed together, without a clear distinction between the different phases. 2) A reconstruction of the piezometric morphology as well as a stratigraphy of the considered study areas should be reported. This could help also the discussion of the groundwater preferential pathways. 3) The organization of the paper is still at a draft level, since there is not a clear distinction between the results and discussion paragraphs. Many paragraphs need to be summarized and better explained. 4) The number of figures should be reduced (probably putting together some and deleting others). 5) The English is very poor and there are many typo errors. The reported delta notation is wrong. Due to the consideration of these main points the manuscript can be accepted only if major revision will be reported.

**The authors' responses to the comments from Referee #1**

Dear Dr/Professor Referee #1:

On behalf of my co-authors, we thank you very much for giving us an opportunity to revise our manuscript. We appreciate you very much for your positive and constructive comments and suggestions on our manuscript (hess-2018-71). We have studied your comments carefully and have made revision which marked in red in the revised manuscript. We tried our best to revise our manuscript according to the comments point by point. Attached please find the revised version, which we would like to submit for your kind consideration. Thank you and best regards.

1) The datasets belong to sampling campaigns carried out in different moments (years) and seasons and for this reason in my opinion cannot be discussed together, without a clear distinction between the different phases.

Our response: AGREE AND NO CHANGES MADE.

Firstly, we thank you very much for this comment from you and we truly agree this point that water samples collected in different moments (years) and seasons cannot be discussed together without a clear distinction between the different water phases. In fact, although we stated in the manuscript that our fieldwork had taken place during the summer season of 2011 and the spring season of 2012, we collected the natural water samples at the same time for the same phases in the study area. For example, (1) all the groundwater samples discussed in this paper were collected during the 2011 summer in five days in the Otindag Desert. For other natural water samples discussed in this study, the detailed sampling methods are as follow: (2) all the spring water samples and (3) the precipitation water sample (p1) discussed in this paper were also collected during the 2011 summer in five days in the study area, and (4) all the river water samples and (5) lake water samples were collected during the spring season of 2012 in three days in the study area. This is to say that the water samples within the same phase are discussed together in the paper.

2) A reconstruction of the piezometric morphology as well as a stratigraphy of the considered study areas should be reported. This could help also the discussion of the groundwater preferential pathways.

Our response: AGREE AND CHANGES MADE.

We thank you very much for this comment. And yes, according to this comment, we revised the manuscript and focused on reporting the geological (tectonic, lithological, sedimentological and structural), geomorphological, hydrogeological and stratigraphical settings of the study area. Please see the section 2 "Regional setting" of the revised manuscript in its pages 2-4 lines 103-188.

3) The organization of the paper is still at a draft level, since there is not a clear distinction between the results and discussion paragraphs. Many paragraphs need to be summarized and better explained.

**Our response: AGREE AND CHANGES MADE.**

We thank the you very much for this comment. And yes, we have revised the manuscript accordingly. The structure and content of the paper has been thoroughly reorganized in the revised manuscript, especially for the results and discussion sections, to make the content and context of the paper being more logic, coherent and readable. And yes, almost all of the paragraphs in the paper are newly summarized and explained. The detailed changes can be easily observed in the revised manuscript by reading one of the two resubmitted MS-Word files with the "changes marked" version (in contrast, another version is "clear copy").

4) The number of figures should be reduced (probably putting together some and deleting others).

**Our response: AGREE AND CHANGES MADE.**

We thank you very much for this comment. And yes, we have revised the manuscript accordingly. We reduced the number of figures in the revised manuscript by putting some figures together and deleting several figures. At last the revised manuscript has 11 figures compared with the original manuscript that including 15 figures. For example, the Figs. 5, 11, 13, 14a in the original manuscript are deleted in the revised manuscript, and the Figs. 7 and 8, the Figs. 10, 12 and 14a are combined, respectively. In addition, two newly-built figures are added into the revised manuscript according to the second comment from you (the detailed content of this comment can be seen above). The specific changes and the final results of these figures can be seen in the newly submitted revised manuscript.

5) The English is very poor and there are many typo errors. The reported delta notation is wrong.

**Our response: AGREE AND CHANGES MADE.**

We thank you very much for this comment. We are very sorry for our poor and incorrect English writing in the original manuscript. For the shortcomings of the English presentation and the grammatical edit in the first paper, we have checked and revised the whole manuscript carefully to avoid language errors, and finally we have got the help of a native English speaking professional to check and improve the English quality of the revised manuscript. We believe that the language is now acceptable for the publishing purpose.

In addition, the wrong use of the dalta notation in the original manuscript, such as  $\delta^2H$ , has been corrected as " $\delta D$ " in the revised manuscript.

6) Due to the consideration of these main points the manuscript can be accepted only if major revision will be reported.

**Our response: AGREE AND CHANGES MADE.**

Special thanks to you for your good comments. We have tried our best to improve the manuscript and made specific changes in the revised manuscript according to the comments from you one by one. These changes will not influence the content and framework of the paper. And here we did not list the changes but marked in red in the revised paper. We hope that the correction will meet with approval. Once again, thank you very much for your comments and suggestions.

**Direct or indirect recharge on groundwater in the
middle-latitude desert of Otindag, China?**

Bing-Qi Zhu1\*, Xiao-Zong Ren2, Patrick Rioual3

1KLWCRESP, IGSNRR, CAS, Beijing, China

2SGS, TYNU, Jinzhong, China

3KLCGE, IGGCAS, Beijing, China

*Correspondence to:* Bing-Qi Zhu ([zhubingqi@sina.com](mailto:zhubingqi@sina.com))

**Abstract.** The Otindag Desert is essential to livestock-economy and ecoenvironment of northern China.
Although surface water is the traditional source for China's socio-economy in arid areas, the
groundwater resources underlying the desert are increasingly burdened by groundwater pumping,
which increases interest in the status of the groundwater resources. Widespread fresh groundwater deep
to 60 m was found at the eastern part of the Otindag Desert. The occurrence of this massive fresh
groundwater raises doubts on the often-made assumption in the literature that regional atmospheric
precipitation or palaeowater, namely the direct recharge, is the source of water in the middle-latitude
desert aquifers of northern China and makes further investigation necessary. Knowledge on the origin
and recharge of this fresh groundwater is key in assessing the possibility of groundwater exploitation
and utilization. In this study we conducted hydrogeochemical and isotopical analyses to assess possible
origin and recharge of these groundwaters. It is concluded that the fresh groundwater can neither
originate from regional atmospheric precipitation derived from the Asian Summer Monsoon system,
nor from palaeowater that formed during the last glacial period. Our results indicate that with
groundwater dating it is possible to originate from remote mountain areas via the faults of the Solonker
Suture zone, including the Daxing'Anlin and Yinshan Mountains. Furthermore, it is deduced that the
hydrological connection between desert aquifers and mountain systems through the suturezone is
crucial to the hydrogeological functioning of the Otindag aquifer. This suggests that the modern indirect
recharge mechanism, instead of the direct recharge and the palaeo-water recharge, is the most
significant for groundwater recharge in the Otindag Desert. This study provides a new perspective into
the origin and evolution of groundwater resources in the middle-latitude desert zone of Asian continent.

**Keywords:** fresh groundwater recharge; atmospheric precipitation; direct recharge; indirect recharge;
palaeowater recharge; fault hydrology; middle-latitude desert; Otindag Desert.

**1. Introduction**

In a semi-arid to arid region where rainfall is insufficient to supply the needs of a growing
population and a higher standard of living, the deficit is normally made up by extracting groundwater.
Many areas in the middle-latitude desert zone of northern China such as the Badanjilin Desert, the Mu
US sandy Land and the Hobq Desert (Chen et al., 2012a; Chen et al., 2012b), are unexpectedly rich with large groundwater resources although they have been under arid or hyper-arid climate for a long
time (Sun et al., 2010). How these groundwaters originated and how they are recharged in these deserts
are thus fundamental scientific questions. Until now, however, no consensus has been achieved in
academic circles.

The Otindag Desert is one of the largest sandy lands located at the monsoon margin of northern
China and is the geographical centre of the northeastern Asian Continent (Fig. 1), which can be
regarded as a significant repository of information relating to the groundwater recharge in the arid
Inner Asia. At present, the eastern Otindag is also a typical case for its unexpected groundwater
resources, because there is abundant groundwater in this desert land and even rivers originate there due
to the spillover of spring water, such as the tributaries of Xilamulun River in its north and the Shandian
River in its south (Fig. 1). Climatically, the monsoon margin of northern China refers to a strip along
the present East Asian Summer Monsoon (EASM) limits and is considered to be sensitive to climate
change (Wang and Feng, 2013). Geologically, the Otindag Desert lies in a tectonic depression of the
central Solonker suture zone with a few faults stretching east and west (Fig. 2), with its northern
margin along a fault marked by a series of lake basins. Thus, the large-scale hydrogeological conditions
of the Otindag Desert belong to a fault zone under the influence of the EASM climate.

Until now, however, whether the climate or other factors affected the groundwater recharge in the
Otindag are still not known. Little data about the groundwater and its origin is available in the literature,
and knowledge and reliable data on various hydrogeological characteristics of the desert such as the
catchment extent, input/output, the hysteretic hydraulic functions, the transient hydraulic conditions,
in-homogeneities, and on transfer functions to overcome scale problems are also missing. Under such
conditions, conventional methods such as water balance and hydraulic methods sometimes fail in
determining groundwater recharge, particularly in extreme environments (arid, semi-arid, or cold)
(Drever, 1997). Because pristine aquatic conditions may significantly differ from managed conditions
in arid environment, and thus groundwater recharge is not a fixed number, but may vary with the
boundary conditions of the recharge system (Seiler and Gat, 2007).

Groundwater recharge can be broadly classified into two categories: the direct recharge by native
water resources and the indirect recharge by external water resources (Herczeg and Leaney, 2011).
Water infiltration of atmospheric precipitation through the unsaturated zone to the groundwater is
hydrologically defined as the direct recharge, and the indirect recharge is defined as recharge from
mappable features such as rivers, canals, lakes and originates from remote areas (Scanlon et al., 2006;
Healy, 2010). It is well known that groundwater recharge can be influenced by environmental factors,
including climate change, underlying soil and geology, land cover and the growth in human population
that affects withdrawal and economic development (Zhu et al., 2015, 2017). Among these
environmental factors, climate and land cover largely determine precipitation and evapotranspiration,
whereas the underlying soil and geology dictate whether a water surplus (precipitation minus
evapotranspiration) can be transmitted and stored in the subsurface (Doll, 2008, 2009; Giordano, 2009).

For some earth scientists, the direct recharge is thought to be very important for groundwaters in
the wide desert lands of north China due to the lack of surface runoffs (Yang et al., 2010; Yang and
Williams, 2003; Zhao et al., 2017). They argued that although the amount of atmospheric precipitation is small, the vast catchment area in the desert region could concentrate the rainfall into large inland
basins, creating an aquifer with large storage capacity and great thickness. However, some hydrologists
estimated by the chloride mass balance method that the direct recharge was 1.4 mm/year, which
represents approximately only 1.7% of the mean annual precipitation in a cold large desert (Badanjilin)
in northern China (Gates et al., 2008). A similar estimation of 1 mm/year was given for Gobi deserts
from the Hexi Corridor to the Inner Mongolia Plateau in northwestern China (Ma et al., 2008).
Consequently, they thought that heavy potential evaporation and little precipitation make it difficult for
direct recharge to meet the supply of groundwater in these desert areas. Thus, the indirect recharge is
considered to be an important mechanism for groundwater recharge in these desert areas. For example,
Zhao et al. (2012) suggested that little precipitation had recharged into groundwaters in the Badain
Jaran Desert. Chen et al. (2004) argued that the groundwaters in the Badanjilin Desert were recharged
by palaeo-glacial melt water through faults and deep carbonate layers far away from the local desert.
Many studies also suggested that palaeowaters stored in an aquifer during wetter climate periods could
recharge to groundwater under certain conditions in arid lands (Edmunds et al., 2006; Ma and Edmunds,
2006). Other kinds of indirect recharge, such as mountain front recharge from adjacent mountain
blocks, are also proposed to offer an important inflow to aquifers within arid to semiarid catchments
(Blasch and Bryson, 2007).

In this paper, we focus to answer the question that whether groundwater recharge in Otindag is
mainly direct or indirect, using hydrochemical and isotopic indicators as tracers to offer a valuable
support for identifying the contributions of precipitation recharge on groundwater, since these
indicators reflect the composition of water molecules and are sensitive to physical processes such as
mixing and evaporation (Sultan et al., 2000; Guendouz et al., 2003; Petrides et al., 2006; Scanlon et al.,
2006; Zhu et al., 2007, 2008; Jobbágé et al., 2011). The detailed objectives are: (1) to recognize the
major sources of groundwater in the area, and (2) to identify the key mechanism of groundwater
recharge in the desert.

**103 **2. Regional settings**

Geographic setting. The Otindag Desert lies between latitudes 42° and 44°N and longitudes 112°
and 118° E (Fig. 1). It forms a part of the great middle-latitude desert belt in northern China which
stretches from the Taklamakan Desert of northwestern China to the Kelqin Desert of northeastern
China, near the west coast of the Pacific Ocean. The desert has an area of approximately 21,400 square
kilometers located in the eastern Inner Mongolia and at the monsoon margin of northern China (Fig. 1).
It is the fourth largest sandy lands in China (Yang et al., 2012) and is bordered by a flat steppe terrain
of Dali Basin to the north, the Yinshan Mountains and mountainous loess landscape to the south, and
the the Greater Khingan (Daxing'Anling) Mountains to the east (Fig. 1). The Otindag Desert is
essential to livestock-economy and ecoenvironment of northern China. Settlements in this desert are
restricted to areas to permanent springs, shallow groundwater and oases to areas where irrigation is
possible. Some nomads continue to eke out a precarious existence grazing livestock in the desert.

Topography and geomorphology. The Otindag Desert has a varied relief, combining extensive
dune fields with rugged mountains along the eastern, southern and southeastern rims. In the east, the

Daxing'Anling Mountains stretch from the Heilong River Valley into the upper reach valleys of the
Xilumulun River from northeast to southwest, gradually increasing in height northwards from about
180 m near Huma to Huanggangliang, where the highest peaks reach 2,029 m with an average
elevation range from 1,100 to 1,400 m. In the south and southeast, the Yinshan Mountains decline
gradually near Duolun and Zhenglanqi, and in some areas leave wide alluvial plains. The terrain of the
Otindag Desert is less rough and elevations decrease from ca. 1300 m in the southeast to ca. 1000 m in
the northwest. Over the greater part of this desert the ground cover consists of fixed and semi-fixed
sandy dunes, with a few mobile dunes in area of little vegetation. The dominated dune types are
represented from parabolic to barchans, linear and grid-formed types, ranging from a few meters to
over 40 m in height (Zhu et al., 1980; Yang et al., 2008).

Climate, vegetation and soil. The climate of the Otindag Desert was not uniform in geological
period, with much sand movement, occasional rainy years, and several wetter intervals during the
Holocene (Yang et al., 2015; Tian et al., 2017). At present the whole desert belongs to the arid and
semi-arid temperate zone, with a meanannual temperature of 2 °C in the north and 4°Cin the south (Liu
and Yang, 2013). At the regional scale, the climate of the desert is typically controlled by the East
Asian Monsoon system, characterized by a warm summer, with precipitation transported by the EASM,
and by a cold and dry winter under the influence of the East Asian Winter Monsoon (EAWM). The
rainfall in the desert exhibits a wide variation in space and time. Influence of the EASM changes from
southeast to northwest in the desert, and varies with latitude and distance from the Pacific Ocean,
leading to the mean annual rainfall decreasing from ~450 mm in the southeast to ~150 mm in the
northwest (Yang et al., 2013). This uneven distribution of precipitation has a major influence on the
availability of near-surface moisture, consequently on the distribution of vegetation, soil and the animal
husbandry potential of local communities. The basic soil cover consists of grey desert soil in the west
and changes to sierozems and chernozem or chestnut soil in the east. Through the desert, vegetation is
sparse in the west and relatively abundant in the east. The natural vegetation is characteristic of desert
or semi-deserts, with scrub woodland in the east and steppe in the west. Due to the scarcity of surface
water, the growing season is affected by temperature, rainfall and elevation, and hence cultivation is
restricted mainly to flood plains.

Geology. The Otindag Desert is located in a tectonic depression of the Solonker Suture Zone (Jian
et al., 2010) bounded by the Northern Early to Mid-Paleozoic Orogen Zone and the Hatug Uul Block to
the north, the Southern Early to Mid-Paleozoic Orogen Zone and the North China Craton system to the
south (Fig. 2). A few faults such as the Xar Moron Fault and Chifeng-Bayan Obo Fault stretch east and
west, with its northern margin along the Solonker Suture Zone marked by a series of lake basins (Figs.
1 and 2). The tectonostratigraphic units and overall structural trends are mainly oriented NE–SW (Fig.
2), which may be interpreted as resulting from overall compressive stresses oriented principally in the
NW–SE quadrants during orogenesis (Jian et al., 2010; Zhang et al., 2015). Diverse rock types from
unlithified and lithified clastic sediments through to carbonate, crystalline, and volcanic rocks are
distributed in and around the Otindag Desert (Zhang et al., 2015) (Figs. 2 and 3). Tertiary and
Quaternary sandstones and mudstones are the common basement rocks under the dunes of the Otindag,
and extensive volcanic basalts forming flat terrains are to the north (Zhu et al., 1980; Li et al., 1995).

Hydrology and hydrogeology. The Otindag Desert originated during the Late Quaternary (Yang
et al., 2015) and various alluvial fans formed at the margins of this desert during the early to middle
Holocene. These are composed of conglomerate and sand deposits, where major periodic steams or
wadis debouched into the Otindag. At present two rivers run through the eastern margin of the Otindag
Desert, i.e. the Xilamulun River in the north and the Shandian River and its two tributaries, the Shepi
River and Tuligen River in the south. Both stem from the eastern and southeastern parts of the Otindag
(Fig. 1). The Xilamulun River, 380 km in length and  $32.54 \times 10^3 \text{ km}^2$  in area, is a neighboring river both
to the northeastern Otindag and the southeastern Dali Basin, the northern catchment of the Otindag
Desert. The Xilamulun River flows to the east and finally goes into the Xiliao River, with an annual
mean runoff of  $6.58 \times 10^8 \text{ m}^3$  (Wu et al., 2014). The Shandian River is the upper reach of the Luan
River, with a length of 254 km and a catchment area of  $4.11 \times 10^3 \text{ km}^2$  (Yao et al., 2013). Along the low,
flat and sandy shorelines of some lakes in the Otindag, salt flats or sabkhas have formed in shallow
depressions. Due to the high rate of evaporation, salt crusts develop which have been locally exploited
where the salt is relatively free from sand. During rainy season, some rain and floodwaters (generally
coming from the Yinshan piedmonts) are retained in low-lying areas, which may temporarily recharge
shallow aquifers. Under storm conditions, occasional heavy, short rainstorms cause floods in soil-rich
wadi channels. Under other conditions, sand dunes and sand sheets bury the ground and sabkhas.

The Otindag Desert can depend on several water-bearing formations and units (aquifers) for their
groundwater resources (Fig. 3). Coarse- to fine-grained sedimentary rocks, magmatic rocks and
metamorphic rocks of the Inner Mongolia-Daxing'Anling Orogenic Belt (Zhang et al., 2015) form the
major regional aquifer unit (Fig. 3). They are composed mainly of alluvial sediments (mid-Permian
Zhesi Formation), melange (Solonker suture zone), A-type granite (early Permian), bimodal volcanic
rocks with sedimentary intercalations (early Permian Dashizhai Formation), diorite-quartz
diorite-granodiorite rocks (Carboniferous-Permian) and metamorphic complex (predominantly gneiss,
early Paleozoic) (Fig. 2). The aquifer is generally unconfined in dune fields of the Otindag Desert,
unconfined to semi-confined in the Yinshan Mountains' piedmont, and semi-confined to confined in the
Daxing'Anling uplands (Fig. 3). Water-level measurement in June 2010 indicated that the general depth
of unconfined groundwater level ranges between 10 to 70 m in the Otindag Desert (Fig. 3). Local
granular aquifers in the central desert are composed of coarse fluvial, lacustrine and aeolian sediments,
but their extent and thickness vary throughout the watershed (Zhu et al., 1980; Li et al., 1995). The
generally coarse-grained texture of the unconsolidated rock formations provides primary porosity in
terms of groundwater flow in the desert.

**190 **3.Methods**

The hydrochemistry of natural water in the Otindag Desert, as related to the prevailing EASM
climate, as well as, the dominant topographical, geological (tectonic) and hydrogeological conditions,
are discussed here and interpreted, using chemical and isotope analyses of water samples from rain,
springs, shallow aquifers and deep aquifers, rivers and lakes, and are represented on relevant graphs
and diagrams. Fieldworks took place during the summer season of 2011 and the spring season of 2012.
Water samples were mainly retrieved from shallow and deep wells located over a wide area in dune fields of the study regions. The detailed locations of the sampling sites are shown in Fig. 4.

Two groups of parameters are measured to characterize the chemistry of any water analysis:
field-measured parameters and lab-measured parameters. The filed-measured parameters include
temperature (°C), hydrogen-ion concentration (pH), electrical conductivity (EC in micro-Siemens per
centimeter or  $\mu\text{S}/\text{cm}$ ) and total dissolved solid (TDS, mg/L). The values of these parameters change
when they are not directly measured in the field. The number lab-measured parameters depend on the
purpose of study. However, the measurement of major cations ( $\text{F}^-$ ,  $\text{Cl}^-$ ,  $\text{NO}_2^-$ ,  $\text{NO}_3^-$ ,  $\text{SO}_4^{2-}\text{HCO}_3^-$ ,
$\text{CO}_3^{2-}$  and  $\text{H}_2\text{PO}_4^-$ ) and anions ( $\text{Li}^+$ ,  $\text{Na}^+$ ,  $\text{NH}_4^+$ ,  $\text{K}^+$ ,  $\text{Mg}^{2+}$  and  $\text{Ca}^{2+}$ ) are determined in most chemical
analyses. Analysis for stable ( $^2\text{H}$  and  $^{18}\text{O}$ ) and radioactive isotopes ( $^3\text{H}$ ) in rain and groundwater are
also included. The analytical data of the physiochemical parameters and the stable and radioactive
isotopes of the water samples collected in this study are listed in Tables 1, 2 and 3, respectively.

**209 **4. Results and Discussions**

**211 **4.1. Hydrochemical characteristics of natural waters**

The natural water samples collected in this study are generally neutral to slightly alkaline, with the
pH values varying between 6.26 and 9.44 (except the precipitation sample p1, 4.61) (Table 1) and a
median value of 7.27. The TDS values range between 67 and 660 mg/L (average 211 mg/L) (Table 1),
all belonging to fresh water (TDS < 1000 mg/L) in the salination classification of natural water
(Meybeck, 2004). The variations in ion concentrations of the major cations and anions in the studied
water samples were displayed in a fingerprint diagram with a semi-logarithm y-axis (Fig. 5). The rain
water sample is the most depleted in ions among these samples. The groundwater samples have the
highest concentrations of cations and anions and the lake, river and spring waters had intermediate
values. The calcium concentration is the highest among cations in almost all of the water samples, and
the  $\text{HCO}_3^-+\text{CO}_3^-$  concentration (bicarbonate + carbonate, alkalinity) is the highest among anions in most
of the water samples. For several groundwater samples (g3, g4, g5, g6 and g11), spring sample (s1) and
precipitation sample (p1), they have higher  $\text{SO}_4^{2-}$  concentrations than alkalinity (Fig. 5).

Two chemically distinct water types are recognized for the studied waters via a Piper diagram (Fig.
6), calcium bicarbonate and calcium sulphate. No Chloride-type and sodium-type waters occur in the
study area (Fig. 6). Based on more than 10,000 chemical analyses of groundwater samples from the
world, Chebotarev (1955) observed that the global groundwater tends to evolve chemically towards the
composition of seawater. He also observed that this evolution is associated with regional changes in
dominant anions but not cations, as the concentration of cations may exhibit a wide range of
fluctuations in groundwater and is not as steady as the changes in anion dominance. Freeze and Cherry
(1979) illustrated the Chebotarev's (1955) general evolution of groundwater as a anion evolution line:
$\text{HCO}_3^- \rightarrow \text{HCO}_3^- + \text{SO}_4^{2-} \rightarrow \text{SO}_4^{2-} + \text{HCO}_3^- \rightarrow \text{SO}_4^{2-} + \text{Cl}^- \rightarrow \text{Cl}^- + \text{SO}_4^{2-} \rightarrow \text{Cl}^-$ , which travels
along the flow paths and increasing ages. On this evolution line, bicarbonate water is generally
characteristic of low salinity, renewable water resources and low residence time, while sulphate waters
predominate in groundwater passing through gypsum and anhydrite aquifers, and is usually associated
with intermediate salinity in unconfined aquifers (Clark, 2015). The distribution pattern of water chemical types occurred in the studied area indicates a primary stage of groundwater evolution in the
Otindag Desert.

The  $\delta D$  values of the groundwater samples collected in this study varied from -63.42‰ to -75.92‰
(Table 3), with an average -69.53‰. The  $\delta^{18}\text{O}$  values ranged between -8.64‰ and -11.26‰ (Table 3),
with an average -10.17‰. The spring water samples were relatively concentrated in  $\delta D$  and  $\delta^{18}\text{O}$  and
were greatly similar to those of the groundwater samples (Fig. 7). The  $\delta D$  and  $\delta^{18}\text{O}$  values in the river
water samples were slightly more variable and were also similar to those of the groundwater (Fig. 7).
The lake water samples were enriched in  $\delta D$  and  $\delta^{18}\text{O}$  by comparison to the groundwater samples (Fig.
6). The precipitation sample p1 was also enriched in  $\delta D$  and  $\delta^{18}\text{O}$  by comparison to the groundwater
samples (Fig. 7). The content of radioactive isotope of tritium ( $^3\text{H}$ ) measured in seven well
groundwater samples with 6-60 m depth ranged from 1.86 to 24.35 TU (Table 3), with an average
14.95 TU, higher than the mean tritium concentration (9.8 TU) of groundwater in the Vienna Basin,
Austria (Stolp et al., 2010), the seat of the International Atomic Energy Agency (IAEA).

If we plot the relationships between oxygen and hydrogen isotopes of groundwater, spring, river
and lake water samples, we observed that the regression line that fits all data points can be described by
the equation:  $\delta D = 4.09\delta^{18}\text{O} - 28.31$  ( $R^2=0.93$ ,  $n=24$ ) (EL1 in Fig. 7). This local groundwater line
(LGWL) is different from the Global Meteoric Water Line (GMWL,  $\delta D = 8\delta^{18}\text{O} + 10$ ) and the
Mediterranean Meteoric Water Line (MMWL,  $\delta D = 8\delta^{18}\text{O} + 20$ ) estimated by Craig (1961), but it is
similar to the local groundwater lines established for other deserts in northern China and central Asia
with a same slope but different Y-intercepts, such as  $\delta D = 4.17\delta^{18}\text{O} - 31.3$  for the Badanjilin Desert (Jin
et al., 2018),  $\delta D = 4.8\delta^{18}\text{O} - 15.2$  for the Ejina Desert in China (Wang et al., 2013), and  $\delta D = 4.26\delta^{18}\text{O}$
+ 9.23 for the Rub Al Khal Desert in the United Arab Emirates (Rizk and El-Etr, 1997). The scatter of
stable isotope data points for the lake water samples (Fig. 7) in the Otindag suggests that the lake
waters are affected by evaporation, but the other waters in the desert are not so.

**262 **4.2. Local precipitation recharge on groundwater in the Otindag**

To incorporate the isotopic analysis of precipitation with similar areas in the studied area, local
data (p1) was plotted with those of Baotou (Fig. 7). The isotopic composition of rainfall in Baotou, the
nearest long-term station to the Otindag Desert, was monitored for the period 1986-2001 within the
scope of the International Atomic Energy Agency/World Meteorological Organization (IAEA/WMO)
global survey. The stable isotope data available from this station was used to provide basic
characteristics of the stable isotopic composition of the present-day meteoric water, especially in the
westward inland areas of the Otindag Desert (Fig. 1). Stable isotope data of the Tianjin station was also
used to characterize precipitation of the eastern coastal areas of the Otindag Desert (Fig. 1).

Based on the isotopic data from the Baotou station, the local meteoric water lines can be
statistically expressed as the isotopic regression equation of  $\delta D = 6.36\delta^{18}\text{O} - 5.21$  (LMWL-B). It can
also be expressed as  $\delta D = 6.57\delta^{18}\text{O} + 0.31$  (LWML-T), based on the data from the Tianjin station (Fig.
7). The precipitation sample p1 collected in this study fell onto the GMWL (Fig. 7). It also showed
similar  $\delta D$  and  $\delta^{18}\text{O}$  values to those of the precipitation collected in the GNIP stations of Baotou and
Tianjin (Fig. 7).

      Compared to the precipitation data from the GNIP stations and from the local precipitation (p1),
      the groundwater, spring, and river water samples were evidently depleted in heavy stable isotopes in
      the Otindag (Fig. 7). Except for the lake water samples, most of the groundwater, river water and
      spring water samples in the Otindag fall on or lay between the LMWL-B and the LMWL-T lines, and
      are located at the lower left area of the precipitation points (Fig. 7).

      Because the isotopic evolution of  $\delta D$  and  $\delta^{18}O$  in water illustrated in the Craig line represents a
      one-way and irreversible process, the water bodies distributed at the upper right area of the Craig line
      can not be recharge sources for the water bodies distributed at the lower left area of the line. Such
      results indicate that the groundwater, river water and spring water in the Otindag are not recharged by
      the regional precipitation, namely no significant modern direct recharge has taken place for
      groundwater in the Otindag.

      Dogramaci et al. (2012) documented that only intense and remarkable rainfall events  $>20$  mm
      could recharge groundwater in the semi-arid Hamersley Basin of northwest Australia, while the rainfall
      events  $<20$  mm had limited influences on groundwater recharge. Chen et al. (2014) described that
      rainfall events  $\leq 5$  mm in the arid and semi-arid region of northern China would be evaporated into
      the atmosphere rapidly before it is infiltrated into the groundwater system. Based on the analysis on the
      data records from two meteorological stations around the Otindag, i.e. the Duolun and Xilinhaote
      stations (see Fig. 1a), we observed that rainfall events  $>20$  mm on average only occur 2.5-3.4 times per
      year (Table 4). In some years (e.g. from 2005 to 2007 at the Xilinhaote Station), no rainfall events  $>20$
      mm even occurred. It further indicated the limited contribution of regional precipitation on
      groundwater recharge in the Otindag.

      In addition to groundwater, the river and spring water samples from the Otindag also deviated
      from the local precipitation in the Craig diagram (Fig. 7). These water samples came from the
      Xilamulun, Shepi and Tuligen rivers. They shared the same evaporation line (EL1) with the
      groundwater and lake water samples (Fig. 7). Generally speaking, natural waters that have a same
      recharge source are distributed on a same line of evaporation in the  $\delta^2$  and  $\delta^{18}O$  diagram (Chen et al.,
      2012b). This indicates that the recharge sources of groundwater, river water, spring water and lake
      water in the Otindag are genetically associated each other and differ from the local precipitation.

**306       **4.3. Winter precipitation and palaeowater recharge on groundwater in the Otindag**

      Since the groundwater samples in the Otindag are depleted in their  $\delta D$  and  $\delta^{18}O$  values even more
      than those of the local rainfall (Fig. 7), they must be sourced from other waters characterized by similar
      or more depleted signals in their stable isotopes compositions. Due to the temperature effect (such as
      evaporation) on isotopic fractionation, only the waters issued from colder environments can be more
      depleted in their  $\delta D$  and  $\delta^{18}O$  values even more than those of the local rainfall.

      Because the Otindag Desert is under the control of the EASM climate (Fig. 1), the local rainfall in
      the desert is mainly sourced from summer precipitation. This can also be illustrated by the seasonal
      distributions in annual mean precipitation (Fig. 8a), in annual mean air temperature (Fig. 8b) and in
      annual mean water vapor pressure (Fig. 8c) over the last forty years at the two surrounding GNIP
      weather stations in Baotou and Tianjin. The seasonal distributions of stable isotopes in the two stations (Fig. 8d-e) show that the summer rainfall is evidently positive in its signals of  $\delta D$  and  $\delta^{18}O$  by
comparison with those of the winter rainfall, further suggesting that the waters issued from cold
environments can be more depleted in their  $\delta D$  and  $\delta^{18}O$  values than those of the summer rainfall. Thus
we speculate that groundwater in the Otindag can be potentially derived from (1) modern precipitation
in winter, (2) palaeowater formed in the past glacial period, or (3) remote/mountains waters that
emanate in colder and wetter conditions.

The annual mean values of  $\delta D$  and  $\delta^{18}O$  over the last forty years are more depleted in winter
precipitation than in summer precipitation at the Baotou and Tianjin stations (Fig. 8d-e). This isotopic
signal qualifies the regional winter precipitation to be a potential source of groundwaters in the Otindag.
However, the precipitation amounts and the water vapor pressures (effective moisture) in winter
months are much lower than those in the summer months at both the Baotou and Tianjin stations (Fig.
8a and 8c). It indicates that the winter seasons in these regions are relatively colder and drier but not
colder and wetter. A colder-wetter winter season is a necessary condition for winter precipitation to be a
water source for the formation of groundwater under a summer monsoon climate. This is because the
bigger amounts of summer precipitation will easily remove or weaken the depleted isotopic signals of
winter precipitation in groundwater. In this regard, modern winter precipitation is unlikely to be an
important source of groundwater in the Otindag.

As to the palaeowaters formed in colder and wetter periods such as the last glacial, it has been
proposed to be a potential water source for groundwaters in the wide arid lands of the world. The
depleted signals of stable isotopes ( $\delta D$  and  $\delta^{18}O$ ) in groundwater have been recognized in global arid
and semi-arid regions, such as the Sinai Desert in Egypt (Gat and Issar, 1974), Israel (Gat, 1983), South
Australia (Love et al., 1994, 2000), northern China (Ma et al., 2010), Saudi Arabia (Bazuhair and Wood,
1996) and North Africa (Guendouz et al., 2003). These signals are very often explained as
palaeo-groundwater that recharged by precipitation during past wetter and colder periods (Love et al.,
1994, 2000; Herczeg and Leaney, 2011).

Here we use the tritium data as a environmental tracer to estimate the groundwater age in the
Otindag. The tritium data at the GNIP stations of the Baotou and Tianjin are also referenced as the
background values in precipitation of recent years. The residence time of groundwater in aquifer and
the residual tritium of a water body can be calculated by  $N = N_0 e^{-\lambda t}$  (Yang and Williams, 2003). Where
$N$  = content of residual tritium in water sample,  $\lambda = 0.0565$ , the radioactive decay constant,  $N_0$  =
content of tritium at the time of rainfall and  $t$  = years after precipitation. Based on this equation, the
residual tritium was theoretically calculated and the standard for tritium dating was established for
seven groundwater samples in the Otindag Desert (Table 3). As a result, ages of 0-60 years were
obtained for these groundwater samples (Table 5). This indicates that recent recharge took place several
decades after the peak in global nuclear tests. We thus conclude that groundwater is generally not older
than 70 years in the study area. It means that groundwater in the Otindag are not palaeowater
recharged.

Both the modern summer and winter precipitation recharge and the palaeowater recharge can be
refuted, indicating that direct recharge is not a major mechanism controlling the groundwater recharge
in the Otindag.

**4. 4. Remote water recharge on groundwater in the Otindag: Dali Basin**

The third hypothesis that “remote/mountains waters emanate under colder and wetter conditions”
is further considered here. In essence, it is an indirect recharge mechanismas as water originates from
remote areas (Healy, 2010; Herczeg and Leaney, 2011).

It is worth noting that the values of deuterium and oxygen-18 for groundwater in the north part of
the study area are more depleted in  $\delta\text{D}$  and  $\delta^{18}\text{O}$  than those in the south part (Table 3). It suggests that
the Otindag groundwater might be potentially recharged by water resouces coming from the northern
neighboring catchment, such as the Dali Basin.

Recently published data of  $\delta\text{D}$  and  $\delta^{18}\text{O}$  in groundwater, lake water, river water and spring water
sampled from the Dali Basin (e.g., Chen et al., 2008; Zhen et al., 2014) were compiled in this study and
were co-analyzed with the data from the Otindag. About 70 natural water samples from the Dali and
Otindag with  $\delta\text{D}$  and  $\delta^{18}\text{O}$  values are shown in a Craig diagram (Fig. 9). All of these samples fell on or
lied near the evaporation line EL2 in the Craig diragram (Fig. 9), with a regression equation of  $\delta\text{D} =$
$4.81\delta^{18}\text{O} - 21.55$  and a high correlation coefficient ( $R^2=0.98$ ,  $n=70$ ). Compared to the groundwater
samples in the Otindag, water samples from the groundwaters, rivers and springs from the Dali Basin
are more depleted in  $\delta^{18}\text{O}$  and  $\delta\text{D}$  (Fig. 9). Such results further indicate that, in terms of itsisotopic
signature, the groundwater in the Otindag has a close relationship with the natural waters in the Dali
Basin.

The similar signals of  $\delta\text{D}$  and  $\delta^{18}\text{O}$  between the groundwater in the Otindag and the river water in
the Dali (Fig. 9) point towards the idea that the groundwater in the Otindag might be sourced from the
river water in the Dali Basin, since the Dali has more depleted isotopic signals in water than the
Otindag (Fig. 9). Considering the topographical gradient of elevations between the two regions,
however, river water in the Dali Basin cannot flow into the eastern Otindag, because the terrain
elevation of the Dali Basin is lower than that of the Otindag (Fig. 1). This is also the reason why the
huge Dali Lake that lies in the Dali Basin has no equivalent in the Otindag (Fig. 1). If there is a
hydraulic linkage between the two regions, water should flow from the Otindag into the the Dali, but
not conversely.

In view of the hydraulic gradient, river water in the Dali Basin could not be a recharge source for
groundwater in the Otindag. However, in view of the isotopic gradients, groundwater in the Otindag
could not conversely be the source of river water in the Dali (Fig. 9). Thus, the similar isotopic signals
between the river water in Dali and the groundwater in Otindag indicate that these waters might be
recharged from a common source.

Similar isotopic signals also occurred in the groundwaters between the Otindag and the Dali Basin
(Fig. 9). In order to understand the linkage of groundwaters between the two regions, the potential
movement of groundwater in the transition zone of the two regions need to be known. In this study, a
groundwater-sampling project was designed in the field along a N-S section of a palaeo-channel
located at the transition zone between the Dali and Otindag (Figs. 1, 2). The channel was named
“PCSX” in this study, with its north part named “NPCSX” and the south part named “SPCSX”.

The GPS elevation of the northernmost sampling site in the NPCSX (g11, about 1317 m a.s.l.) was much lower than that of the southernmost site in the SPCSX (g1, 1396 m.a.s.l.) (Fig. 2 and Table 1).
Regarding to the topographical gradient in the channel, there is a drop of about 80 m between the
NPCSX and the SPCSX. Under such slope, the underground hydraulic gradient for groundwater flow
can be roughly parallel with that of the surface water flow, namely that the groundwaterflow should
move downwards from the SPCSX area into the NPCSX area. Thus we can speculate that groundwater
in the NPCSX would have higher salinity than those in the SPCSX under such flowing direction. In
order to verify this speculation, actual variations of water salinity (chloride and TDS) were detected
along the PCSX section. The sampling site g1 was defined as the initial point and the distances between
g1 and other sampling sites along the PCSX section were calculated, based on their GPS geographical
coordinates measured in the field. The results are shown in Fig. 10a-b. It is clear that the variations of
chloride and TDS concentrations in groundwater do not increase along the palaeo-channel from south
to north (Fig. 10a-b). On the contrary, both the values of chloride and TDS are lower in the NPCSX
area than those in the SPCSX area. Such kind of spatial variations in the chloride and TDS values
contradict the speculated patterns abovementioned, suggesting that the hydraulic gradient of
groundwater flowing path in this region is not controlled by the topographical gradient between the
NPCSX and SPCSX areas.

Compared between the NPCSX and SPCSX regions, the stable isotopic values ( $\delta^{18}\text{O}$  and  $\delta\text{D}$ ) of
groundwaters in the SPCSX region vary greatly with a large amplitude, while those in the NPCSX are
relatively constant (Fig. 10c-d). The constant variations indicate that the recharge source of
groundwater in the NPCSX is relatively unitary. The isotopic values in the SPCSX are much lighter
than those in the NPCSX along the distance section from south to north (Fig. 10c-d). The heaviest
values occurred in the sample g11 collected from the NPCSX (Fig. 10c-d), indicating a water being
earlier recharged. The spring water sample s2, a representation of discharge water, is characterized by
medium values of  $\delta\text{D}$  and  $\delta^{18}\text{O}$ . These results indicate that the groundwaters in the SPCSX area, with
relatively enriched isotopic signals in  $\delta\text{D}$  and  $\delta^{18}\text{O}$  by comparison with those in the NPCSX area, are
composed of a mixture of the groundwaters in the NPCSX with other waters.

The tritium contents were broadly and positively related to the values of deuterium excess in the
groundwater samples in the PCSX (Fig. 10e). For water that experiences an evaporation process, the
d-excess value will increase in the evaporated water vapor, but will decrease in the residual water body
(Dansgaard, 1964; Merlivat and Jouzel, 1979). In this study, except for sample g11 (a sample very
close to the riverhead area), the positive relationship between the tritium and the deuterium excess
generally shows that the d-excess values are higher in the groundwaters collected from the NPCSX, but
are lower in those from the SPCSX (Fig. 10e). This distribution pattern indicates that the groundwaters
in the NPCSX are relatively younger and experienced a lower degree of evaporation than those in the
SPCSX. The d-excess gradient, increasing from south to north in the PCSX, further suggests that
groundwater does not flow from the SPCSX area to the NPCSX area, namely out of the topographical
control.

Many studies (e.g., Boronina et al., 2005; Kazemi et al., 2006) have demonstrated that
groundwater flows in the direction in which it gets older. In view of this point, groundwaters in the
PCSX region should flow from the NPCSX area to the SPCSX area, in opposition to the S-N

topographical gradient between the Otindag and Dali regions. Thus groundwater in the Dali are not the
source of groundwater in the Otindag. The similar isotopic signals between groundwaters in the two
regions indicate that these waters might be recharged from a common source in other place.

**441 **4. 5. Remote water recharge on groundwater in the Otindag: mountains waters**

The discussions above revealed that both the groundwaters in the Otindag and Dali Basin might be
recharged from a common source derived from another place. Considering the third hypothesis
abovementioned that “remote/mountains waters emanate under colder and wetter conditions”, we
propose that this “common source” of the two regions are from mountain areas surrounding the
Otindag and Dali Basin.

There are two large permanent rivers and lots of small intermittent streams entering the Dali Basin
(Xiao et al., 2008), including the Xilamulun River to the south and the Gongger River to the north, both
of which are stemming from the Greater Khingan Mountains (Daxing’Anling Mountains in Chinese
pinyin, 1,100-1,400 m above seal level) (Fig. 1). The Xilamulun River carries a large amount of water
(about  $6.58 \times 10^8 \text{ m}^3/\text{y}$ ) from the Daxing’Anling Mountains flowing through the east margins of the Dali
and Otindag (Wu et al., 2014). This is an important clue linking natural waters between the Otindag
and Dali Basin.

Variation in the elevation from the Dali Lake to the riverhead of the Xilamulun River can be
clearly found along a land surface topographical section (Fig. 11). The channel of the Xilamulun River
is located in the Xar Moron Fault (Fig. 1), which is a part of the Solonker Suture Zone (Eizenhöfer et
al., 2014) or the Xilamulun-Changchun-Yanji plate suture zone (Sun et al., 2004) in the regional
tectonical settings (Fig. 2). Outcrop observations indicate that fault zones commonly have a
permeability structure suggesting they should act as complex conduit-barrier systems in which
along-fault flow is encouraged and across-fault flow is impeded (Bense et al., 2013). Thus the
hydraulic gradient of groundwater flow in the Eastern margins of the Otindag and Dali Basin must be
controlled by the fault zone hydrogeology. This may be the reason why the hydraulic gradient of
groundwater represented by the isotopic and hydrogeochemical gradients of groundwater samples in
this study is not consistent with the local topographical gradient in the Otindag Desert. On the other
hand, the regional aquifer is generally unconfined in dune fields of the Otindag Desert but
semi-confined to confined in the Daxing’Anling uplands (Fig. 3), thus the thick unconsolidated
aquifers in the study area (Figs. 3 and 11) will be favourable conditions for groundwater storage and
transportation along the Solonker Suture Zone. When rivers stem from the Daxing’Anling
Mountains and flow downward to the marginal areas of the Dali and Otindag, leakage water from these
rivers can recharge the desert land through thick unconsolidated aquifers. A strong isotopic evidence is
that the lake and river waters in the Dali Basin share the same evaporation line (EL2) with the
groundwaters in the PCSX area.

Although groundwaters in the SPCSX area are different from those in the NPCSX area, their
isotopic data points still fell onto the EL2 (Fig. 9), which further indicates that the groundwaters in the
SPCSX are a mixture of waters from the Daxing’Anling Mountain and other sources. Another source
for groundwater recharge in the SPCSX could be represented by remote water such as flash floods coming from the north Yinshan Mountains, because it can be clearly observed from digital maps that
many transient rivers or streams originated from the Yinshan Mountains flow into the south and
southeastern Otindag (Fig. 1). Supportive evidence for this idea can also be observed in the summer
rainy season. During rainy days or under storm conditions, occasional heavy, short rainstorms cause
floods in soil-rich wadi channels and low-lying depressions in the unconfined to semi-confined areas of
the Yinshan Mountains' piedmont. These waters may temporarily recharge shallow aquifers in the
SPCSX area.

**485 **5. Conclusions**

In the middle-latitude desert zone of northern China, many deserts such as the Otindag and
Badanjilin Deserts, are unexpectedly rich in groundwater resources, although they have no surface
runoff and have been under an arid or hyper-arid climate for a long period of time. How groundwaters
originated and recharged in these deserts are thus key questions that are still under debate. For some
earth scientists, the direct recharge is thought to be very important for groundwaters in the wide desert
lands of northern China, due to the lack of surface runoffs. However, groundwater availability is very
much a function of the local- and regional-scale geological and climatic settings. To achieve an
integrated understanding of the groundwater recharge and its controlling mechanisms is of great
significance. In this study, groundwater recharge was explored using multiple environmental tracers in
the Otindag Desert of northern China, a region that is under the influence of the East Asian Summer
Monsoon (EASM) climate. Compared to modern summer precipitation, the groundwaters, river waters
and spring waters are depleted in  $\delta D$  and  $\delta^{18}O$ . All these waters shared a same Craig line, indicating a
genetic relationship on their recharge sources. The stable isotopic signals of the groundwaters is more
depleted than those of the modern summer precipitation and this suggests that the groundwaters studied
could only be sourced from cold water different from the EASM precipitation. In general, the analyses
revealed that the highland remote water resources from the Daxing'anling and Yinshan Mountains
were isotopically and geochemically traced to be a major source for the groundwater in the Otindag. It
suggests that the modern indirect recharge mechanism, instead of the direct recharge and the
palaeo-water recharge, is the most significant for groundwater recharge in the eastern Otindag. This
study provides a new perspective into the origin and evolution of groundwater resources in the
middle-latitude desert zone of northern China.

**508 **Acknowledgements**

This study was financially supported by the National Natural Science Foundation of China
(41771014 and 41602196) and the National Key Research and Development Program of China
(2016YFA0601900). We thank the China Meteorological Data Sharing Service system for providing
the weather data. Sincere thanks are also extended to Profs. Xiaoping Yang, Xunming Wang, Jule Xiao
and other workmates, e.g., Ziting Liu, Hongwei Li, and Deguo Zhang for their generous help in the
research work.

[revised manuscript text omitted]

**Fig. 9.** The bivariate diagram of  $\delta D$  and  $\delta^{18}O$ , i.e. the Craig diagram, for the natural water samples collected in the Otindag (this study) and the Dali Basin. Different relationships between the groundwaters, lake waters, river waters, spring waters and the precipitation waters are clearly illustrated. AWMB, AWMT, LWMB, LWMT, GMWL, LMWL-B, LMWL-T, and EL1 are the same as in Fig. 7. EL2, the evaporation line calculated based on the data from the groundwater, lake water, river water and spring water samples collected from the Otindag and Dali Basin. The data for the Dali were taken from Chen et al. (2008) and Zhen et al. (2014).

**Fig. 10.** (a) Sketch map showing the relationship between the groundwaters in the NPCSX and SPCSX areas, based on variations of (a) the chloride concentrations, (b) the TDS concentrations, (c) the  $\delta^{18}\text{O}$  values and (d) the  $\delta\text{D}$  values of these water samples versus their distances away from the water sample g1 along the palaeo river channel (PCSX) from south to north. The dashed line in (c) and (d) represents the corresponding values of the spring water sample s2, and divides samples into the NPCSX and SPCSX parts. (e) Variations of tritium contents vs. deuterium excess for the groundwater samples in the study area. The sample g6 was omitted due to its potential contamination.

[revised manuscript text omitted]

Bing-Qi Zhu1\*, Xiao-Zong Ren2, Patrick Rioual3

1KLWCRESP, IGSNRR, CAS, Beijing, China

2SGS, TYNU, Jinzhong, China

3KLCGE, IGGCAS, Beijing, China

Correspondence to: Bing-Qi Zhu (zhubingqi@sina.com)

**Abstract.** Although rainfall is scarce in most desert lands of the world, the Otindag Desert in the
middle-latitude desert zone of northern China in Northern Hemisphere (NH) has abundant of water
resources, mainly groundwater. To gain an insight into the origin of groundwater origin in this desert,
stable and radioactive isotopes and major ion hydrochemistry of groundwater, as well as other natural
waters including river water, spring water, lake water and precipitation water, were investigated in the
eastern part of the Otindag. The results showed that the groundwaters in the Otindag were freshwater
(TDS < 700 mg/L) and were depleted in  $\delta^{2}H$  and  $\delta^{18}O$ , when compared with the modern precipitation.
The major water types are the  $Ca-HCO_3$  and  $Ca/Mg-SO_4$  types waters. No  $Cl$  type and  $Na$  type
waters occurred in the study area. The ionic and depleted stable isotopic signals in groundwater, as well
as the high values contents in of tritium contents (5.25 TU), indicated that
these groundwaters studied were young but not of meteoric origin, i.e., out of control by the modern
and paleo direct recharge. Clear differences in the isotopic signals were observed between the
groundwaters in the north (NPCSX) and south (SPCSX) parts of the study area, but the signals were
similar in between the groundwaters between in the NPCSX and its neighbouring catchment, the Dali
Basin. The topographical elevation is decreasing from the SPCSX (1396 m a.s.l.) to the NPCSX
(1317 m a.s.l.) and the Dali (1226 m a.s.l.). Groundwaters in the NPCSX were characterized by
lower elevations, the lower chloride and TDS concentrations, higher tritium contents, higher deuterium
excess, and more depleted values of  $\delta^{2}H$  and  $\delta^{18}O$  than those in the SPCSX. The spatial distribution
pattern of these environmental parameters This indicates a discrepancy between in the one
hand the hydraulic gradient of groundwater and in the other hand the isotopic and
hydrochemical gradients of groundwater in the desert eastern Otindag. It also suggests that the
groundwaters have different water recharge sources between the two areas parts in the study area.
However, the groundwaters in the two areas shared a common evaporation line (EL2) in the Craig
diagram of  $\delta^{2}H$  and  $\delta^{18}O$ , indicating a genetic relationship in their recharge sources. Combined analysis
was further performed using the isotopic and physico-chemical data of natural waters collected from the
Dali Basin and the surrounding mountains. It indicated that the major recharge sources of the
groundwaters in the NPCSX, as well as the river waters and groundwaters in the Dali Basin, were mainly
derived from the Daxin'Anling Mountains, by leaking of the Xilamulan River water through a thick
aquifer in the eastern margins of the Otindag. By contrast, While the groundwaters in the

SPCSX were mainly recharged from two sources. One was the flash floods derived from the Yinshan
Mountains and the river waters other was the Xilamulun River waters derived from the Daxin'Anlin
Mountains. It indicates that the modern indirect recharge mechanism, instead of the direct recharge and
the palaeo-water recharge, is the most significant for groundwater recharge in the eastern Otindag. This
suggests that the tectonic settings at a regional scale, but not the climate, is at the origin of what
was responsible for the groundwater origin in the Otindag. This study provides a new perspective into
the origin and evolution of groundwater resources in the middle latitude desert zone of NH. The Otindag
Desert is essential to the livestock-economy and ecoenvironment of northern China. Although surface
water is the traditional source for China's socio-economy in arid areas, the groundwater resources
underlying the desert are increasingly burdened by groundwater pumping, which increases interest in
the status of the groundwater resources. Widespread fresh groundwater deep to 60 m was found at the
eastern part of the Otindag Desert. The occurrence of this massive fresh groundwater raises doubts on
the often-made assumption in the literature that regional atmospheric precipitation or palaeowater,
namely the direct recharge, is the source of water in the middle-latitude desert aquifers of northern
China and makes further investigation necessary. Knowledge on the origin and recharge of this fresh
groundwater is key in assessing the possibility of groundwater exploitation and utilization. In this study
we conducted hydrogeochemical and isotopical analyses to assess possible origin and recharge of these
groundwaters. It is concluded that the fresh groundwater can neither originate from regional
atmospheric precipitation derived from the Asian Summer Monsoon system, nor from palaeowater that
formed during the last glacial period. Our results indicate that with groundwater dating it is possible to
originate from remote mountain areas via the faults of the Solonker Suture zone, including the
Daxing'Anlin and Yinshan Mountains. Furthermore, it is deduced that the hydrological connection
between desert aquifers and mountain systems through the suture zone is crucial to the hydrogeological
functioning of the Otindag aquifer. This suggests that the modern indirect recharge mechanism, instead
of the direct recharge and the palaeo-water recharge, is the most significant for groundwater recharge in
the Otindag Desert. This study provides a new perspective into the origin and evolution of groundwater
resources in the middle-latitude desert zone of Asian continent.

**Keywords:** fresh groundwater recharge origin atmospheric precipitation; direct recharge; indirect
recharge; palaeowater recharge; fault hydrology; middle-latitude desert; direct and indirect recharge;
stable and radioactive isotope; ion hydrochemistry; climate control; tectonic control Otindag Desert.

**68 1. Introduction**

Water Resources. In a semi-arid to arid region where rainfall is insufficient to supply the needs of
a growing population and a higher standard of living, the deficit is normally made up by extracting
groundwater. [Alsharhan, 2001, Hydrogeology of an Arid Region The Arabian Gulf and Adjoining
Areas.] As rainfall events are infrequent in arid and semi-arid regions of the world, surface runoff
and related water resources are globally scarce and ephemeral. These areas thus rely heavily on
groundwater as the primary water resource to support local ecosystems (Herezeg and Leaney, 2011;
Seanlon et al., 2006). It has been widely proved that the origin, quality and quantity of groundwater in

arid lands can be deeply influenced by environmental factors/processes, which controlling the
groundwater recharge and evolution, such as in the arid lands of northwestern China and Central Asia
(Zhu et al., 2015, 2016, 2017). For this reason these factors/processes become an essential component in
the understanding of regional hydrological systems and the management of water resources
(Dogramaci et al., 2012). For example, groundwater recharged by modern precipitation can refill
quickly but is vulnerable to contamination by the surface wastes; inversely, groundwater containing
mostly ancient water may not recharge to a useful extent over human timescales and cannot be affected
by surface waters (Bethke and Johnson, 2008). Therefore, different strategies on groundwater
resources management should be adopted when the different recharge mechanisms of groundwater
occurring.

In general, groundwater recharge can be broadly classified into two ways, the direct
recharge, namely diffuse recharge by native water resources, and the indirect recharge, namely focus
recharge by external water resources. The direct recharge is replenished by precipitation infiltration
through the unsaturated zone and the indirect recharge is defined as recharge from mappable features
such as rivers, canals, and lakes originated from remote areas (Healy, 2010). It is well known that
groundwater recharge can be influenced by environmental factors, including climate change,
underlying soil and geology, land cover and population growth, over withdrawal and economic
development (Zhu et al., 2015, 2017), thus the amount of groundwater in arid and semi-arid regions
decrease rapidly while human demands on the limited water resources increase rather than decrease
(Ma et al., 2013). Between environment and groundwater recharge, climate and land cover largely
determine precipitation and evapotranspiration, whereas the underlying soil and geology dictate
whether a water surplus (precipitation minus evapotranspiration) can be transmitted and stored in the
subsurface (Giordano, 2009; Doll, 2009). Modelled estimates of diffuse recharge globally (Doll and
Fedler, 2008; Wada et al., 2010) range from 13,000 to 15,000 km3/yr, equivalent to ~30% of the
world's renewable freshwater resources (Doll, 2009) or a mean per capita groundwater recharge of
2100 to 2500 m3/yr. These estimates represent potential recharge fluxes as they are based on a water
surplus rather than measured contributions to aquifers. Furthermore, these modelled global recharge
fluxes do not include focused recharge, which, in semi-arid and arid environments, can be substantial
(Scanlon et al., 2006; Favreau et al., 2009). For keeping sustainable management of water resources, it
requires urgently to understand both diffuse and focused recharge and meet both human and ecosystem
needs in arid areas of the world, particularly in Central Asia and Northern China.

Many areas in the middle-latitude desert zone of northern China such as, many areas of these
lands the Badanjilin Desert, the Mu Us sandy Land and the Hobq Desert (Chen et al., 2012a; Chen et
al., 2012b), are unexpectedly rich in incommensurate with large groundwater resources, such as the
Badanjilin Desert, the Mu Us Sandy Land and the Hobq Desert (Chen et al., 2012a; Chen et al., 2012b),
although they have been under arid or hyper-arid climate for a long time (Sun et al., 2010). How these
groundwater are originated and how they are recharged in these deserts are thus fundamental
scientific becoming a keyquestions. Until now, however, no consensus has been achieved it has long
been altered in the academic circles.

For some of the earth scientists, the direct recharge is thought to be very important for groundwaters in the wide desertlands of northwestern China due to lack of surface runoff (Yang et al.,
2010; Yang and Williams, 2003; Zhao et al., 2017). They argued that although the amount of
atmospheric precipitation is small, the vast catchment area in the desert region could concentrate the
rainfall into large inland basins, creating an aquifer with large storage capacity and great
thickness. However, some hydrologists suggested that the estimate of direct recharge used by the
chloride mass balance method was 1.4 mm/year, approximately only 1.7% of the mean annual
precipitation in a cold large desert (Badanjilin) in northern China (Gates et al., 2008). A similar
estimation was only 1 mm/year for Gobi deserts from the Hexi Corridor to the Inner Mongolia Plateau
in northwestern China (Ma et al., 2008). Consequently, they thought that heavy potential evaporation
and little precipitation make it difficult for direct recharge to meet the supply of groundwater in these
desert areas. Thus, the indirect recharge is considered to be an important mechanism for groundwater
recharge in these desert areas. For example, based on isotopic compositions of natural waters, Zhao et al.
(2012) suggested that little precipitation had recharged into groundwaters in the Badain Jaran
Desert. Chen et al. (2004) argued that the groundwaters in the Badanjilin Desert were recharged by
paleo glacial melt water through faults and deep carbonate layers far away from the local desert. Many
studies also suggested that paleowaters stored in aquifer during wetter climate periods could recharge
to groundwater under certain conditions in arid lands (Edmunds et al., 2006; Ma and Edmunds,
2006). Other kinds of indirect recharge, such as mountain front recharge from adjacent mountain
blocks, are also proposed to offer an important inflow to aquifers within arid to semiarid
catchments (Blasch and Bryson, 2007).

The Otindag Desert is one of the largest sandy desert lands located at the monsoon margin of
northern China and is the geographical centre of the northeastern Asian Continent (Fig. 1), which can
be regarded as a significant repository of information relating to the groundwater recharge in the arid
Inner Asia. At present, the eastern Otindag is also a typical case for its unexpected incommensurate
groundwater resources, because there is abundant groundwater in this desert land and even rivers
originate there due to the spillover of spring water, such as the tributaries of Xilamulun River in its
north and the Shandian River in its south (Fig. 1). Climatically, the monsoon margin of northern China
refers to a strip along the present East Asian Summer Monsoon (EASM) limits and is considered to be
sensitive to climate change (Wang and Feng, 2013). Geologically, the Otindag Desert lies in a tectonic
depression of the central Solonker suture zone with a few faults stretching east and west (Fig. 2), with
its northern margin along a fault marked by a series of lake basins. Thus, the large-scale
hydrogeological conditions of the Otindag Desert belong to a fault zone under the influence of the
EASM climate.

Until now, however, whether the climate or other factors the tectonic faults affected the origin of
groundwater recharge and the fluid flow patterns in groundwater aquifers in the Otindag are still not
known. Because at present, until now, however, little data and documents about the groundwater
and its origin is available in the literature in Otindag, and knowledge and reliable data on various
hydrogeological characteristics of the desert such as the catchment extent, input/output, the hysteretic
hydraulic functions, the transient hydraulic conditions, in-homogeneities, and on transfer functions to

overcome scale problems are also missing. Under such conditions, conventional methods such as water
balance and hydraulic methods sometimes fail in determining groundwater recharge, particularly in
extreme environments (arid, semi-arid, or cold) (Drever, 1997).   pristine aquatic
conditions may significantly differ from managed conditions in arid environment, and thus
groundwater recharge is not a fixed number, but may vary with the boundary conditions of the recharge
system (Seiler and Gat, 2007).

. It is well known that groundwater
Among these environmental factors, climate and land cover

. They argued that although the amount of atmospheric precipitation is small, the vast catchment area in
. A similar estimation of 1 mm/year was given for Gobi deserts from the Hexi
. Consequently, they thought that heavy potential evaporation and little precipitation make it difficult for
Other kinds of indirect


Bryson, 2007:

In this paper, we focus to answer the a question that whether groundwater recharge in Otindag is
mainly direct or indirect, using hydrochemical and isotopic indicators as tracers to offer a valuable
support for identifying the contributions of precipitation recharge on groundwater, since these
indicators reflect the composition of water molecules and are sensitive to physical processes such as
mixing and evaporation (Lawrence et al., 1976; Coplen, 1993; Sultan et al., 2000; Guendouz et al., 2003;
Petrides et al., 2006; Scanlon et al., 2006; Zhu et al., 2007, 2008; Jobbágy et al., 2011; Zhai et al., 2013;
Eissa et al., 2014). T, as the abovementioned hot question for other deserts in China, is also unknown.

It should be kept in mind that virgin aquatic conditions may significantly differ from managed
conditions in arid environment, because groundwater recharge is not a fixed number, but may vary with
the boundary conditions of the recharge system (Seiler and Gat, 2007). Conventional methods such as
water balance and hydraulic methods sometimes fail in determining groundwater recharge in extreme
environments (arid, semi-arid, or cold) (Drever, 1997), because of missing knowledge and the lack of
reliable data on various characteristics such as the catchment extent, input/output, the hysteretic
hydraulic functions, the transient hydraulic conditions, in homogeneities, and on transfer functions to
overcome scale problems (Seiler and Gat, 2007). Under such conditions, tracer methods offer a
valuable support for natural water studies.

Geochemical elements and environmental isotopes have been widely used as effective tracers to
determine the sources of groundwater recharge, which could be attributed to infiltration by rainfall,
surface waters or both of them (Zhu et al., 2007, 2008; Zhu et al., 2017). For example, by comparing the
composition of stable isotopes of hydrogen and oxygen in local meteoric waters with these in
groundwaters, many studies successfully applied in identifying whether the rainfall play a vital role in
recharging groundwater or not (Zhu et al., 2007; Petrides et al., 2006; Jobbágy et al., 2011; Zhai et al.,
2013). Also, investigating the spatial distribution of groundwater age represented by the concentration
of tritium or radioactive carbon ( $^{14}\text{C}$ ) can provide a way to understand the recharge relationship
between the modern rainfall and the groundwater (Sultan et al., 2000; Zhu et al., 2008). For the indirect
recharge, the groundwater flow regimes or its movement pathway deduced from hydrochemical and
isotopical tracers can indicate its origin and recharge processes. For example, the groundwater
mineralisation will increase as a result of dissolution of evaporite minerals along flow lines that begin
with the recharge area (Guendouz et al., 2003). While, the geochemical and isotopic composition of
groundwaters will be much complex at interface zones between groundwaters with different
hydrochemistry or ages, they will show distinct physiochemical characteristics indicating how they
mixed (Lawrence et al., 1976; Eissa et al., 2014).

The detailed The objectives of this study are: (1) to examine the distribution patterns of
environmental signals in the stable and radioactive isotopes and the major ionic hydrochemistry of
groundwater in the eastern Otindag drainage system, and (2) to recognize the major sources of
groundwater in the area, and (3) to identify the key mechanism of groundwater recharge in the desert
land, particularly to discriminate whether the direct recharge or the indirect recharge being the major
control on groundwater recharge in the desert land.

自动设置, 图案: 清除

自动设置

自动设置, 图案: 清除

自动设置, 图案: 清除

自动设置

**2. Regional settings**

**Geographic setting.** The Otindag Desert lies between latitudes 42° and 44°N and longitudes 112°
and 118° E (Fig. 1). It forms a part of the great middle-latitude desert belt in northern China which
stretches from the Taklamakan Desert of northwestern China to the Kelqin Desert of northeastern
China, near the west coast of the Pacific Ocean. The desert has an area of approximately The Otindag
Desert (~21,400 square kilometers km2) is, a middle latitude sandy land desert located in the eastern
of the Inner Mongolia Plateau and at situated at the monsoon margin of northern China as the
geographical centre of the northeastern Asian Continent (Fig. 1). It is the fourth largest sandy lands in
China (Yang et al., 2012) (Yang et al., 2012) and is bordered by a flat steppe terrain of Dali Basin to the
north, the Yinshan Mountains Range and mountainous loess landscape to the south, and the the Greater
Khingan (Daxing'Anling) Mountains Range to the east (Fig. 1). The Otindag Desert is essential to
livestock-economy and ecoenvironment of northern China. Settlements in this desert are restricted to
areas to permanent springs, shallow groundwater and oases to areas where irrigation is possible. Some
nomads continue to eke out a precarious existence grazing livestock in the desert.

**Topography and geomorphology.** The Otindag Desert has a varied relief combining extensive
dune fields with rugged mountains along the eastern, southern and southeastern rims. In the east, the
Daxing'Anling Mountains stretch from the Heilong River Valley into the upper reach valleys of the
Xilumulun River from northeast to southwest, gradually increasing in height northwards from about
180 m near Huma to Huanggangliang, where the highest peaks reach 2,029 m with an average
elevation range from 1,100 to 1,400 m. In the south and southeast, the Yinshan Mountains decline
gradually near Duolun and Zhenglanqi, and in some areas leave wide alluvial plains. The terrain of the
Otindag Desert is less rough and elevations decrease from ca. 1300 m in the southeast to ca. 1000 m in
the northwest. Over the greater part of this desert the ground cover consists of fixed and semi-fixed
sandy dunes, with a few mobile dunes in area of little vegetation. The dominated dune types are
represented from parabolic to barchans, linear and grid-formed types, ranging from a few meters to
over 40 m in height (Zhu et al., 1980; Yang et al., 2008).

**Climate, vegetation and soil.**

**The climate of the Otindag Desert was not uniform in geological period, with much sand**
**movement, occasional rainy years, and several wetter intervals during the Holocene (Yang et al., 2015;**
**Tian et al., 2017). At present, the Sandy Land is in a tectonic depression with a few faults stretching**
**east and west, with its northern margin along a fault marked by a series of lake basins. Tertiary and**
**Quaternary sandstones and mudstones are the common basement rocks under the dunes, and extensive**
**volcanic basalts forming flat terrains are to the north (Zhu et al., 1980; Li et al., 1995). (Yang et al.,**
**2007, Catena)**

**The Otindag's elevation is variable, ranging from ca. 1300 m in the southeast to ca. 1000 m in the**
**northwest. The whole desert belongs to the temperate arid and semi-arid temperate zone of northern**
**China, with a meanannual temperature of 2 °C in the north and 4°C in the south (Liu and Yang,**
**2013) (Liu and Yang, 2013). At the regional scale, the climate of the desert climate is typically**
**controlled by the East-Asian Monsoon system, characterized by a warm summer, with precipitation**
**transported by the EASM, and by a cold and dry winter under the influence of the East Asian Winter**

自动设置

自动设置

自动设置

自动设置

自动设置

自动设置

自动设置, 图案: 清除

自动设置

自动设置, 图案: 清除

自动设置

自动设置

自动设置, 图案: 清除

自动设置

自动设置

自动设置

New Roman, 10 磅, 无下划线, 字
体颜色: 自动设置

自动设置

自动设置

自动设置

Monsoon (EAWM). The rainfall in the desert exhibits a wide variation in space and time. Influence of the EASM is changes from southeast to northwest in the desert, and varies with latitude
and distance from the Pacific Ocean, leading to the mean annual rainfall decreasing from ~450 mm in
the southeast to ~150 mm in the northwest (Yang et al., 2013) (Yang et al., 2013). This uneven
distribution of precipitation has a major influence on the availability of near-surface moisture,
consequently on the distribution of vegetation, soil and the animal husbandry potential of local
communities. The basic soil cover consists of grey desert soil in the west and changes to sierozems and
chernozem or chestnut soil in the east. Through the desert, vegetation is sparse in the west and
relatively abundant in the east. The natural vegetation is characteristic of desert or semi-deserts, with
scrub woodland in the east and steppe in the west. Due to the scarcity of surface water, the growing
season is affected by temperature, rainfall and elevation, and hence cultivation is restricted mainly to
flood plains. Fixed and semi-fixed sandy dunes are dominate the landscaped in the desert land, with a
few mobile dunes in area of little vegetation. Several dune types are represented various from
parabolic to barchans, linear and grid formed types, ranging from a few meters to over 40 m in height
(Yang et al., 2008; Zhu et al., 1980).

**Geology.** The Otindag Desert is located in a tectonic depression of the Solonker Suture Zone (Jian et al., 2010) bounded by the Northern Early to Mid-Paleozoic Orogen Zone and the Hatug Uul Block to the north, the Southern Early to Mid-Paleozoic Orogen Zone and the North China Craton system to the south (Fig. 2). A few faults such as the Xar Moron Fault and Chifeng-Bayan Obo Fault stretch east and west, with its northern margin along the Solonker Suture Zone marked by a series of lake basins (Figs. 1 and 2). The tectonostratigraphic units and overall structural trends are mainly oriented NE-SW (Fig. 2), which may be interpreted as resulting from overall compressive stresses oriented principally in the NW-SE quadrants during orogenesis (Jian et al., 2010; Zhang et al., 2015). Diverse rock types from unlithified and lithified clastic sediments through to carbonate, crystalline, and volcanic rocks are distributed in and around the Otindag Desert (Zhang et al., 2015) (Figs. 2 and 3XX).

(Bense et al., 2013, ESR-1)

**Tertiary and Quaternary sandstones and mudstones are the common basement rocks under the**
**dunes of the Otindag, and extensive volcanic basalts forming flat terrains are to the north (Zhu et al.,**
**1980; Li et al., 1995).**

**Hydrology and hydrogeology**

**The Otindag Desert originated during the Late Quaternary (Yang et al., 2015) and various**
**alluvial fans formed at the margins of this desert during the early to middle Holocene. These are**
**composed of conglomerate and sand deposits, where major periodic steams or wadis debouched into**
**the Otindag. At present two rivers run through the eastern margin of the Otindag Desert, i.e. the**
**Xilamulun River in the north and the Shandian River and its two tributaries, the Shepi River and**
**Tuligen River in the south. Both stem from the eastern and southeastern parts of the Otindag (Fig. 1).**
**The Xilamulun River, 380 km in length and  $32.54 \times 10^3 \text{ km}^2$  in area, is a neighboring river both to the**
**northeastern Otindag and the southeastern Dali Basin, the northern catchment of the Otindag Desert.**
**The Xilamulun River flows to the east and finally goes into the Xiliao River, with an annual mean**
**runoff of  $6.58 \times 10^8 \text{ m}^3$  (Wu et al., 2014). The Shandian River is the upper reach of the Luan River.**

with a length of 254 km and a catchment area of  $4.11 \times 10^3 \text{ km}^2$  (Yao et al., 2013). Along the low, flat
and sandy shorelines of some lakes in the Otindag, salt flats or sabkhas have formed in shallow
depressions. Due to the high rate of evaporation, salt crusts develop which have been locally exploited
where the salt is relatively free from sand. During rainy season, some rain and floodwaters (generally
coming from the Yinshan piedmonts) are retained in low-lying areas, which may temporarily recharge
shallow aquifers. Under storm conditions, occasional heavy, short rainstorms cause floods in soil-rich
wadi channels. Under other conditions, sand dunes and sand sheets bury the ground and sabkhas.

The Otindag Desert can depend on several water-bearing formations and units (aquifers) for their
groundwater resources (Fig. 3). Coarse- to fFault zone hydrogeology, 【Bense et al., 2013, ESR】

Outerop observations indicate that fault zones commonly have a permeability structure suggesting
they should act as complex conduit barrier systems in which along fault flow is encouraged and
across fault flow is impeded. 【Bense et al., 2013, ESR】

Hydrogeological observations of fault zones reported in the literature show a broad qualitative
agreement with outerop-based conceptual models of fault zone hydrogeology. 【Bense et al., 2013, ESR】

Nevertheless, the specific impact of a particular fault permeability structure on fault zone
hydrogeology can only be assessed when the hydrogeological context of the fault zone is considered
and not from outerop observations alone. 【Bense et al., 2013, ESR】

Diverse rock types from unlithified and lithified clastic sediments through to carbonate, crystalline,
and volcanic rocks are distributed in and around the Otindag Desert (Fig. XX). 【Bense et al., 2013, ESR】

Fine-grained sedimentary rocks, magmatic rocks and aeolian sediments metamorphic rocks of the
Inner Mongolia-Daxing'Anling Orogenic Belt (Zhang et al., 2015) XXXX geological province, form the
major regional aquifer unit (Fig. 3). They are composed mainly of alluvial sediments (mid-Permian
Zhesi Formation), melange (Solonker suture zone), A-type granite (early Permian), bimodal volcanic
rocks with sedimentary intercalations (early Permian Dashizhai Formation), diorite-quartz
diorite-granodiorite rocks (Carboniferous-Permian) and metamorphic complex (predominantly gneiss,
early Paleozoic) (Fig. 2). The aquifer is generally unconfined in dune fields of the Otindag Desert,
unconfined to semi-confined in the YinshanXXX Mountains' piedmont, and semi-confined to confined
in the Daxing'AnlingXXX uplands (Fig. 3). Water-level measurement in June 2010 indicated that the
general depth of unconfined groundwater level ranges between 10 to 70 m in the Otindag Desert (Fig.
3). Local granular aquifers in the central desert are composed of coarseee coarse fluvial, lacustrine and
aeolian sediments, but their extent and thickness vary throughout the watershed (Zhu et al., 1980; Li et
al., 1995). 【Benoit et al., 2014, CWRJ】 The generally coarse-grained texture of the unconsolidated
rock formations provides primary porosity in terms of groundwater flow in the desert.

Most of the tectonic fabric of the Appalachians was generated by compression or low angle
thrusting; in those areas where major faults are strike-slip in nature, deformation is largely limited to
rocks adjacent to the faults. The tectonostratigraphic units and overall structural trends are mainly
oriented NE-SW, which may be interpreted as resulting from overall compressive stresses oriented
principally in the NW-SE quadrants during orogenesis (Faure et al., 2004, 2006). The generally
fine-grained texture of the rock formations provides negligible primary porosity in terms of

groundwater flow (Benoit et al. 2008). [Benoit et al., 2014, CWRJ].

Deformation along faults in the shallow crust (<1 km) introduces permeability heterogeneity anisotropy, which has an important impact on processes such as regional groundwater flow. Fault zones have the capacity to be hydraulic conduits connecting shallow and deep geological environments, but simultaneously the fault cores of many faults often form effective barriers to flow. The direct evaluation of the impact of faults to fluid flow patterns remains a challenge and requires a multidisciplinary research effort of structural geologists and hydrogeologists. [Bense et al., 2013, ESP]

The Uyundag Desert depend on several water bearing formations and units (aquifers) for their groundwater resources. They are composed mainly of sandstone and limestone. **[Alsharhan, 2001]**

Hydrogeology of an Arid Region The Arabian Gulf and Adjoining Areas

During rainy season, some rain and flood waters are retained behind dune dams and recharge shallow aquifers.

Two rivers inrun through the Otindag, i.e. the Xilamulun River in the north and the Shandian River and its with two tributaries of the Shepi River and the Tuligen River in the south. Both stem from the eastern and southeastern parts of the Otindag (Fig. 1). The Xilamulun River flows to the east and finally goes into the Xiliao River, with a catchment area of  $32.54 \times 10^3 \text{ km}^2$  and an annual mean runoff of  $6.58 \times 10^8 \text{ m}^3$  (Wu et al., 2014). The Shandian River is the upper reach of the Luan River, with a length of 254 km and a catchment area of  $4.11 \times 10^3 \text{ km}^2$  (Yao et al., 2013).

**3. Methods**

The hydrochemistry of natural water in the Otindag Desert, as related to the prevailing EASM climate, as well as, the dominant topographical, geological (tectonic) and hydrogeological conditions, are discussed here and interpreted, using chemical and isotope analyses of water samples from rain, springs, shallow aquifers and deep aquifers, rivers and lakes, and are represented on relevant graphs and diagrams. Fieldworks took place during the summer season of 2011 and the spring season of 2012. The water samples selected in this study were all collected from natural water, including the groundwater, river water, lake water, spring water and precipitation water in types A. Total of twenty five water samples were analyzed collected for ion chemical, stable and radioactive isotopic analysis for in this study. Water Groundwater samples is the major type among these waters, which were mainly retrieved taken from shallow and deep wells widely located over a wide area in dune fields of the study regions area. The detailed locations of the sampling sites are shown in Fig. 4.

Two groups of parameters are measured to characterize the chemistry of any water analysis: field-measured parameters and lab-measured parameters. The filed-measured parameters include temperature (°C), hydrogen-ion concentration (pH), electrical conductivity (EC in micro-Siemens per centimeter or  $\mu\text{S}/\text{cm}$ ) and total dissolved solid (TDS, mg/L). The values of these parameters change when they are not directly measured in the field. The number lab-measured parameters depend on the purpose of study. However, the measurement of major cations ( $\text{F}^-$ ,  $\text{Cl}^-$ ,  $\text{NO}_2^-$ ,  $\text{NO}_3^-$ ,  $\text{SO}_4^{2-}$ ,  $\text{HCO}_3^-$ ,  $\text{CO}_3^{2-}$  and  $\text{H}_2\text{PO}_4^-$ ) and anions ( $\text{Li}^+$ ,  $\text{Na}^+$ ,  $\text{NH}_4^+$ ,  $\text{K}^+$ ,  $\text{Mg}^{2+}$  and  $\text{Ca}^{2+}$ ) are determined in most chemical analyses. Analysis for stable ( $^2\text{H}$  and  $^{18}\text{O}$ ) and radioactive isotopes ( $^3\text{H}$ ) in rain and groundwater are

自动设置

自动设置

自动设置, 图案: 清除

自动设置

自动设置, 图案: 清除

自动设置

自动设置，下标

自动设置, 下标

自动设置 上标

自动设置, 上标

自动设置, 上标

自动设置

also included. The analytical data of the physiochemical parameters and the stable and radioactive
isotopes of the water samples collected in this study are listed in Tables 1, 2 and 3, respectively.

The surface waters were mainly sampled from rivers and lakes in the Otindag, and the spring
waters were collected from the riverhead of the Xilamulun River, the Shepi River and the Tuligen River.
One rainfall sample of the local atmospheric precipitation (p1) was also collected at the southeastern
margin of the Otindag in the 2011 summer season. Water samples were filtered using 0.45  $\mu$ m
membrane filters for cation and anion analysis, and were acidified with 1% HNO3 for cation
analysis. Water samples for stable and radioactive isotope analysis were collected in the field with a
polyethylene bottles of 0.5 L in volume, respectively. Variables Some kinds of analysis were measured
on site with a portable instrument (Eijkelkamp). These determinations included temperature, pH,
oxidation reduction potential (Eh), electrical conductivity (EC), and total dissolved solid (TDS). The
measurement errors bars were  $\pm 0.1$  °C for temperature,  $\pm 1\%$  for pH,  $\pm 5\%$  for Eh,  $\pm 5\%$  for EC, and
$\pm 0.5\%$  for TDS, respectively.

The concentrations of major anions (F-, Cl-, NO3-, NO2-, SO42- and H2PO4-) and cations (Li+, Na+,
NH4+, K+, Mg2+ and Ca2+) were determined by electrochemical detectors of an ion chromatography
(Dionex 600) in the Institute of Geology and Geophysics, Chinese Academy of Sciences, with
measurement errors bars  $\pm 3\%$  for anions and  $\pm 2\%$  for cations. The concentrations of carbonate
(alkaline) ions of HCO3- and CO32- were measured by titration with HCl (0.1 M) following the Gran
Method (Gran, 1952), with an error bar  $\pm 5\%$ . The hardness (HD, German standards) of these water
samples was calculated based on the equation HD = ([Mg2+]  $\times$  100/24.305 + [Ca2+]  $\times$  100/40.08)/17.847,
[Mg2+] and [Ca2+] referring to the concentration of Mg2+ and Ca2+ with unit of mg/L.

Two stable isotopes of 2H and 18O, as being expressed in  $\delta$  notation ( $\delta^{2H} = ^2H/H - ^2H/H_{VSMOW}$ ,  $\delta^{18O} = ^{18}O/^6O - ^{18}O/^6O_{VSMOW}$ )
relative to the Vienna standard mean water (VSMOW), were measured for all of the water samples
collected in this study using MAT 252 in the Laboratory for Stable Isotope Geochemistry, Institute of
Geology and Geophysics, Chinese Academy of Sciences, with  $\pm 0.374\%$  for  $\delta^{2H}$  and  $\pm 0.062\%$  for
$\delta^{18O}$  and  $\delta^{18O}$ , respectively.

Several groundwater samples (500 ml each), collected from wells (6–60 m deep) in the study area,
were prepared for the analysis of radioactive isotope (tritium) analysis. 300 ml of water sample, added
with addition of 1 g KMnO4, were distilled to remove any impurities. In order to increase the tritium
concentration to an easily measurable level, electrolytic enrichment was applied (Kaufman, 1954;
Baeza et al., 1999). A volume of 250 ml of previously distilled sample with 2.5 g NaOH was then put
to the electrolysis apparatus containing electrolytic cells with co-axial stainless steel electrodes.
Electrolysis was carried out until the volume of electrolyte was reduced to 8 ml and all runs were
performed at a temperature of 2–5 °C to prevent the loss of tritiated water molecules by evaporation.
After electrolysis CO2 was bubbled through the cell to neutralize the water because the medium in
which the electrolysis took place earlier is alkaline. The water sample was separated from the
electrolyte by distilling. The pretreated samples were measured by a low level background liquid
scintillation counter (Quantulus 1220-003) according to the manufacturer's guidelines. The error bar of
the measurement errors are should be  $\pm 3\%$ . The tritium data of several groundwater samples collected
in this study had been partially mentioned by Yang et al. (2015) as one of the supplementary materials.

It was systematically discussed in this study.

**438 **4. Results and Discussions**

The analytical data of the physicochemical parameters and the stable and radioactive isotopes of
the water samples collected in this study were listed in Tables 1, 2 and 3, respectively. The study area
and the sampling sites location for each sample analyzed were showed in Figs. 1 and 2, respectively.

**444 **4.1. Hydrochemical characteristics of natural the ground and surface waters in the Otindag**

The pH values of the water samples studied varied from 6.26 to 9.44 (except sample p1,
precipitation, 4.61) (Table 1) with a median value of 7.27, indicating that the waters are generally
neutral to slightly alkaline. The TDS ranged between 67 mg/L and 660 mg/L (average 211 mg/L) (Table
1), all belonging to fresh water (TDS < 1000 mg/L) in the salination classification of natural water
(Meybeck, 2004). The natural water samples collected in this study are generally neutral to slightly
alkaline, with the pH values varying between 6.26 and 9.44 (except the precipitation sample p1, 4.61)
(Table 1) and a median value of 7.27. The TDS values range between 67 and 660 mg/L (average 211
mg/L) (Table 1), all belonging to fresh water (TDS < 1000 mg/L) in the salination classification of
natural water (Meybeck, 2004).

The variations in ion concentrations of the major cations and anions in the studied water samples
were displayed in a Schoeller diagram (Schoeller, 1955), a fingerprint diagram with a semi-logarithm
offy-axis (Fig. 3). The rain water sample is the most depleted in ions among these samples. The general
The the groundwater samples have had the highest concentrations of cations and anions while the
precipitation sample (p1) had the lowest concentrations, and the lake, river and spring waters had
intermediate the medium values. The calcium concentration is was the highest among cations in
almost all of the water samples, and the  $\text{HCO}_3 + \text{CO}_3$  concentration (bicarbonate + carbonate, alkalinity)
is was the highest among anions in most of the water samples. For except for several groundwater
samples (g3, g4, g5, g6 and g11), and one of the spring sample (s1) and the precipitation sample (p1),
they have which had the higher  $\text{SO}_4$  concentrations than the alkalinity (Fig. 5).

Two chemically distinct water types are recognized for the studied waters via a Piper diagram
(Fig. 6) (Piper, 1944), calcium bicarbonate and calcium sulphate (Fig. 4). The relative differences in
abundance of ion concentrations between different water can be detectable revealed in a Piper
diagram (Piper, 1944). The water samples studied can be classified into two water types in the Piper
diagram (Fig. 4). Type I, the  $\text{Ca}-\text{HCO}_3$  water, which generally represents the typical bicarbonate
water experienced affected by near surface mineral weathering, and type II, the  $\text{Ca}/\text{Mg}-\text{SO}_4$  water, which
indicates saline water dominated by alkaline earth metals (Zhu et al., 2011, 2012; Clark, 2015). For
water type I, the weak acids exceeded the strong acids; the carbonate hardness (secondary alkalinity)
exceeded 50% and was dominated by the alkaline earths. While for water Type II, the strong acids
exceeded the weak acids and no carbonate hardness exceeded 50%. The alkaline earths ( $\text{Ca}+\text{Mg}$ )
exceeded the alkalis ( $\text{Na}+\text{K}$ ) in all the water samples studied. There were no any Chloride-type and
sodium Na-type waters occurring in the study area (Fig. 6). Based on more than 10,000 chemical

$\delta D = 6.36 \delta^{18}\text{O} + 5.21 (R^2 = 0.93, n = XX)$  for precipitation water in the Baotou Station.

$\delta D = 6.86 \delta^{18}\text{O} - 2.22 (R^2 = 0.91, n = XX)$  for precipitation water in spring in the Tianjin Station.

$\delta D = 6.68 \delta^{18}\text{O} - 0.98 (R^2 = 0.92, n = XX)$  for precipitation water in summer in the Tianjin Station.

$\delta D = 5.51 \delta^{18}\text{O} - 4.12 (R^2 = 0.55, n = XX)$  for precipitation water in autumn in the Tianjin Station.

$\delta D = 7.44 \delta^{18}\text{O} + 13.57 (R^2 = 0.94, n = XX)$  for precipitation water in winter in the Tianjin Station.

$\delta D = 6.66 \delta^{18}\text{O} + 0.30 (R^2 = 0.93, n = XX)$  for precipitation water in spring in the Baotou Station.

The stable isotopes of  $\delta^2\text{H}$  and  $\delta^{18}\text{O}$  were analyzed for all the water samples collected in this study, as shown in Table 3 and Fig. 6. The radioactive isotope of tritium ( $^3\text{H}$ ) was analyzed for a part of the groundwater samples.

The  $\delta D^2\text{H}$  values of the groundwater samples collected in this study varied from -63.42‰ to -75.92‰ (Table 3), with an average -69.53‰. The  $\delta^{18}\text{O}$  values ranged between -8.64‰ and -11.26‰ (Table 3), with an average -10.17‰.

The spring water samples, which directly drain into rivers, were relatively concentrated in  $\delta^2\text{H}$  and  $\delta^{18}\text{O}$  and were greatly similar to those of the groundwater samples (Fig. 76). The  $\delta^2\text{H}$  and  $\delta^{18}\text{O}$  values in the spring samples varied from -70.83‰ to -72.60‰ (mean value -71.72‰) and from -10.34‰ to -10.47‰ (mean value -10.40‰), respectively (Table 3).

The  $\delta D^2\text{H}$  and  $\delta^{18}\text{O}$  values in the river water samples were slightly more variable and were also similar to those of the groundwater (Fig. 76), with a range of between -65.00‰ and -85.16‰ (mean value -73.02‰) in  $\delta D^2\text{H}$  values and a range of between -9.55‰ and -11.78‰ (mean value -10.51‰) in  $\delta^{18}\text{O}$  (Table 3).

The lake water samples in this study were enriched in  $\delta^2\text{H}$  and  $\delta^{18}\text{O}$  by comparison to the groundwater samples (Fig. 6), with a variable range of between -34.16‰ and -53.13‰ (mean value -46.47‰) in  $\delta^2\text{H}$  values and a range of between 0.38‰ and -6.55‰ (mean value -4.65‰) in  $\delta^{18}\text{O}$  (Table 3).

The precipitation sample p1 was also enriched in  $\delta D$  and  $\delta^{18}\text{O}$  by comparison to the groundwater samples (Fig. 76), showed the  $\delta^2\text{H}$  value of -47.4‰ and the  $\delta^{18}\text{O}$  value of -7.14‰, respectively (Table 3). The content of radioactive isotope of tritium ( $^3\text{H}$ ) measured in seven well groundwater samples with 6-60 m depth ranged from 1.86 to 24.35 TU (Table 3), with an average 14.95 TU, higher than the mean tritium concentration (9.8 TU) of groundwater in the Vienna Basin, Austria (Stolp et al., 2010), the seat of the International Atomic Energy Agency (IAEA).

If we plot the relationships between oxygen and hydrogen isotopes of groundwater, spring, river water and lake water samples, we observed that the regression line that fits all data points can be

described by the equation:  $\delta D = 4.09\delta^{18}\text{O} - 28.31$  ( $R^2=0.93$ ,  $n=24$ ) between two straight lines with a gradient of 4.4, but with different y intercepts (EL1 in Fig. 76 XX), as shown in. This local groundwater line (LGWL) is different from the Global Meteoric Water Line (GMWL,  $\delta D = 88\delta^{18}\text{O} + 10$ ) and the Mediterranean Meteoric Water Line (MMWL,  $\delta D = 88\delta^{18}\text{O} + 20$ ) estimated by Craig (1961), but it is similar to the local groundwater lines established for other deserts in northern China and central Asia with a same slope but different Y-intercepts, such as  $\delta D = 4.17\delta^{18}\text{O} - 31.3$  for the Badanjilin Desert (Jin et al., 2018),  $\delta D = 4.8\delta^{18}\text{O} - 15.2$  for the Ejina Desert in China (Wang et al., 2013), and  $\delta D = 4.26\delta^{18}\text{O} + 9.23$  for the Rub Al Khal Desert in the United Arab Emirates (Rizk and El-Etr, 1997). The scatter of stable isotope data points for the lake water samples (Fig. 76) in the Otindag suggests that the lake waters are affected by evaporation, but the other waters in the desert are not so, the following equations:

$$\delta D = 4.38\delta^{18}\text{O} - 24.97$$
 ( $R^2=0.87$ ,  $n=11$ ) for groundwater samples.

$$\delta D = 4.44\delta^{18}\text{O} - 24.56$$
 ( $R^2=0.86$ ,  $n=13$ ) for groundwater and spring water samples.

$\delta D = 4.09\delta^{18}\text{O} - 28.31$  ( $R^2=0.93$ ,  $n=24$ ) for groundwater, springer water, river water and lake water samples.

$$\delta D = 7.95\delta^{18}\text{O} + 10.52$$
 ( $R^2=0.77$ ,  $n=5$ ) for river water samples.

$$\delta D = 2.69\delta^{18}\text{O} - 33.94$$
 ( $R^2=0.92$ ,  $n=6$ ) for lake water samples.

$$\delta D = 6.57\delta^{18}\text{O} + 0.31$$
 ( $R^2=0.88$ ,  $n=XX$ ) for precipitation water in the Tianjin Station.

$$\delta D = 6.36\delta^{18}\text{O} - 5.21$$
 ( $R^2=0.93$ ,  $n=XX$ ) for precipitation water in the Baotou Station.

$$\delta D = 6.86\delta^{18}\text{O} - 2.23$$
 ( $R^2=0.91$ ,  $n=XX$ ) for precipitation water in spring in the Tianjin Station.

$$\delta D = 6.68\delta^{18}\text{O} - 0.98$$
 ( $R^2=0.93$ ,  $n=XX$ ) for precipitation water in summer in the Tianjin Station.

$$\delta D = 5.51\delta^{18}\text{O} - 4.13$$
 ( $R^2=0.55$ ,  $n=XX$ ) for precipitation water in autumn in the Tianjin Station.

$$\delta D = 7.44\delta^{18}\text{O} + 13.57$$
 ( $R^2=0.94$ ,  $n=XX$ ) for precipitation water in winter the Tianjin Station.

$$\delta D = 6.66\delta^{18}\text{O} + 0.30$$
 ( $R^2=0.93$ ,  $n=XX$ ) for precipitation water in spring in the Baotou Station.

$$\delta D = 5.07\delta^{18}\text{O} - 15.1$$
 ( $R^2=0.80$ ,  $n=XX$ ) for precipitation water in summer in the Baotou Station.

$$\delta D = 6.98\delta^{18}\text{O} + 0.85$$
 ( $R^2=0.95$ ,  $n=XX$ ) for precipitation water in autumn in the Baotou Station.

$$\delta D = 6.86\delta^{18}\text{O} - 0.72$$
 ( $R^2=0.98$ ,  $n=XX$ ) for precipitation water in winter in the Baotou Station.

The isotopic regression equation of the Otindag evaporation line (EL1) (Fig. 6), which was calculated based on the  $\delta^2\text{H}$  and  $\delta^{18}\text{O}$  data of the groundwater, lake, river and spring water samples in this study, was  $\delta^2\text{H} = 4.09\delta^{18}\text{O} - 28.31$  ( $R^2=0.93$ ,  $n=24$ ).

The content of radioactive isotope of tritium ( ${}^3\text{H}$ ) was measured in seven well groundwater samples with 6-60 m depth in this study. The tritium concentrations ranged from 1.86 to 24.35 TU (Table 3), with an average 14.95 TU, higher than the mean tritium concentration (9.8 TU) of groundwater in the Vienna Basin, Austria (Stolp et al., 2010), the seat of the International Atomic Energy Agency (IAEA).

域格式的: 无下划线, 字体颜色: 自动设置

**5 Discussion**

**4.5.2.1 Evaluation of Local precipitation recharge on as a recharge source of groundwater in the**

**Otindag**

**Comparison of the isotopic signals between the modern regional precipitation and natural waters**

**in the Otindag**

To incorporate the isotopic analysis of precipitation with similar areas in the studied area, local
data (p1) was plotted with those of Baotou (Fig. 76). The isotopic composition of rainfall in Baotou,
the nearest long-term station to the Otindag Desert, was monitored for the period 1986-2001 within the
scope of the International Atomic Energy Agency/World Meteorological Organization (IAEA/WMO)
global survey. The stable isotope data available from this station was used to provide basic
characteristics of the stable isotopic composition of the present-day meteoric water, especially in the
westward inland areas of the Otindag Desert (Fig. 1). Stable isotope data of the Tianjin station was also
used to characterize precipitation of the eastern coastal areas of the Otindag Desert (Fig. 1).

At present, the extensive record of stable isotope measurements from atmospheric precipitation
areas still lacking from absent in the Otindag. Thus in this study, we used the decadal isotope data of
atmospheric precipitation around the Otindag were collected in this study to determine the isotopic
relationship between the local groundwater and the regional precipitation that are available from. A
global database, the IAEA Global Network of Isotopes in Precipitation (GNIP) database, is available to
use in this study. Taking into account the boundary between the northern hemispheric westerly and the
Asian summer monsoon (Chen et al., 2010), which are the two major climate systems controlling the
Otindag (Yang et al., 2013), we chose two GNIP meteorological stations as the representations of the
atmospheric precipitation derived from the northern hemispheric westerly and the Asian summer
monsoon, respectively. One is the Baotou station, located to the southwest of the Otindag as
representative of (the westerly system), and another is the Tianjin station, located to the southeast of the
Otindag, as representative of (the Asian summer monsoon system) (Fig. 1a). The historical isotopic data
( $\delta^2\text{H}$ ,  $\delta^3\text{H}$  and  $\delta^{18}\text{O}$ , ‰ VSMOW) over the last four decades from the two stations, as well as other data
including the daily precipitation amount (mm) and air temperature (°C) in the same period, were taken
as the references of the stable isotopic signals in precipitation in the Otindag.

The annual weighted mean values of  $\delta^2\text{H}$  and  $\delta^{18}\text{O}$  at the Baotou station varied were variable from
-64.32‰ to -48.44‰ and from -9.40‰ to -6.50‰ during the period of 1986 to 1992, respectively. The
annual weighted mean values of  $\delta^2\text{H}$  and  $\delta^{18}\text{O}$  at the Tianjin station varied from -56.30‰ to -43.72‰
and from -8.35‰ to -6.86‰ during the period of 1988 to 1992 and of 2000 to 2001, respectively. The
long term weighted mean values of  $\delta^2\text{H}$  and  $\delta^{18}\text{O}$  at the Baotou station (LWMB) were -55.27‰ and
-7.78‰, respectively, and were -49.97‰ and -7.70‰ at the Tianjin station (LWMT), respectively. The
radioactive isotope of  ${}^3\text{H}$  (TU) in precipitation was not stable at the GNIP Baotou station. The annual
weighted mean values were higher than 30 TU in this station and tended to be decreased from 1986 to
1991 (72.06, 57.81, 59.97, 52.79, 55.89, 34.35 TU, respectively). The annual weighted mean values of
${}^3\text{H}$  at the GNIP Tianjin station were lower than those of the Baotou station. The mean values were
21.99, 21.65, 18.55, 25.72, 18.80 TU from 1988 to 1992, and 7.01 and 15.48 TU from 2000 to 2001.

As the sample p1, the only one precipitation sample collected in this study (during the 2011
summer rainfall event) of the Otindag, the sample p1 fell onto the Global Meteoric Water Line (GMWL,
$\delta^2\text{H} = 8\delta^{18}\text{O} + 10$ ) estimated by Craig (1961). It showed similar  $\delta^2\text{H}$  and  $\delta^{18}\text{O}$  values to those of the
precipitation collected in the GNIP stations of Baotou and Tianjin (Fig. 6).

Compared to the precipitation data from the GNIP Baotou and Tianjin stations and from the local
precipitation (p1) in the Otindag, the groundwater samples were evidently depleted in heavy stable
isotopes in the Otindag HSKDSL (Fig. 6).

In contrast to the precipitation data, the water samples from springs and rivers in the study area
also showed a depletion characteristics in the stable isotopes of  $\delta^2\text{H}$  and  $\delta^{18}\text{O}$  (Fig. 6).

Based on the isotopic data from the Baotou station, the local regional meteoric water lines, i.e.,
the regional Craig lines, can be statistically described as the isotopic regression equation of
$\delta^2\text{H} = 6.36\delta^{18}\text{O} - 5.21$  (line-LMWL-B). It can also, based on the isotopic data from the Baotou
station, and can be described as  $\delta^2\text{H} = 6.57\delta^{18}\text{O} + 0.31$  (line-LWML-T), based on the data
from the Tianjin station (Fig. 7). The precipitation sample p1 collected in this study fell onto the
GMWL (Fig. 7). It also showed similar  $\delta\text{D}$  and  $\delta^{18}\text{O}$  values to those of the precipitation collected in the
GNIP stations of Baotou and Tianjin (Fig. 7).

Compared to the precipitation data from the GNIP stations and from the local precipitation (p1),
the groundwater, spring, and river water samples were evidently depleted in heavy stable isotopes in
the Otindag (Fig. 7). Except for the lake water samples, most of the groundwater, river water and
spring water samples in the Otindag fell on or lay between the LMWL-B and the LMWL-T lines,
and were located at the lower left area of the precipitation points (Fig. 7). This indicates indicated
that no strong deep evaporation process was experienced by these ground and surface waters (except for
lake waters) compared with than the precipitation.

For the Otindag evaporation line (EL1), its equation slope and intercept were significantly lower
than that of the GMWL, LMWL B and LMWL T (Fig. 6). The points of intersection between the EL1
and LMWL B were atwas 69.93% for  $\delta^2\text{H}$  and 10.18% for  $\delta^2\text{H}$  and for  $\delta^{18}\text{O}$ , respectively, while the
intersection points between the EL1 and LMWL T werewas 75.51% for  $\delta^2\text{H}$  and 11.54% for  $\delta^2\text{H}$
and  $\delta^{18}\text{O}$ , respectively.

**664 5.2. The direct recharge of groundwater in the eastern Otindag**

Water infiltration of atmospheric precipitation through the unsaturated zone to groundwater is
hydrologically defined as the direct recharge. The deuterium and oxygen isotopes are the composition
of water molecules and are sensitive to physical processes such as mixing and evaporation, hence they
are ideal tracers of the origin of groundwater (Coplen, 1993; Seanlon et al., 2006). We used them to
identify the contribution of precipitation recharge on groundwater in this study.

Because the annual mean precipitation amount in the semi-arid regions of northern China is
between 200–400 mm, it seems that the direct recharge on groundwater cannot be neglected in the
eastern Otindag under a semi-arid climate. However, when we checked the stable isotopic data from the
GNIP stations both at the Baotou and Tianjin, we observed that almost all the annual weighted mean
values of the stable isotope contents in precipitation were enriched in  $\delta^2\text{H}$  and  $\delta^{18}\text{O}$  compared with than

those values measured for the groundwater, spring water and river watersamples in this study (Fig.
6). Because the isotopic evolution of  $\delta D^2H$  and  $\delta^{18}O$  in water illustrated in the Craig line represents a
one-way and irreversible process, thus the water bodies distributed at the upper right area of the Craig
line can not be recharge sources for the water bodies distributed at the lower left area of the line. Such
results indicated that the groundwater, river water and spring water in the Otindag are not
recharged by the regional precipitation, namely no significant modern direct recharge has taken place
for groundwater in the Otindag.

Dogramaci et al. (2012) documented that only the intense and remarkable rainfall events of >20 mm could remarkably recharge groundwater in the semi-arid Hamersley Basin of northwest Australia,
while the rainfall events <20 mm had limited influences on groundwater recharge. Chen et al. (2014) described that rainfall events  $\leq 5$  mm in the arid and semi-arid region of northern China would be evaporated into the atmosphere rapidly before it is infiltrated into the groundwater system. Based on the analysis on the data records from two meteorological stations around the Otindag, i.e., the Duolun station and the Xilinhaote stations (see Fig. 1a), we observed that the average times of rainfall events being >20 mm on average in amount were only occur 2.5-3.4 times per year (Table 4). In some years (e.g. from 2005 to 2007 at the Xilinhaote Station), no Even none of the rainfall events of >20 mm even occurred during the year from 2005 to 2007 at the Xilinhaote Station. It further indicated confirmed that the small amounts of intensive rainfall events had limited the contribution of regional precipitation on groundwater recharge in the Otindag.

In addition to groundwater, the river water and spring water samples from the the Otindag
had the similar isotopic signals with those of groundwaters, and were also deviated from the local
modern regional precipitation in the Craig diagram (Fig. 76). These water samples came from the Xilamulun, Shepi and Tuligen rivers. They shared the same evaporation line (EL1) with the groundwater and lake water samples (Fig. 76). Generally speaking, natural waters that have a same recharge source are can be distributed on a same line of evaporation in the  $\delta^2$  and  $\delta^{18}O$  diagram (Chen et al., 2012b) (Chen et al., 2012b). This indicates that the recharge sources of groundwater, river water, spring water and lake water in the Otindag are were genetically associated each other and were differ from ential to the local regional precipitation. During the field investigation, we observed that the elevation of spring outflow was lower than that of the groundwater table in some areas. This implies yes that the spring water can be originateds from the local phreatic water (groundwater). The same isotopic signals between the two kinds of water confirmed their close relationship in origin.

**707 45.3.3. Winter precipitation and palaeowater recharge on groundwater in the Otindag**

Potential sources of groundwater other than summer precipitation in the Otindag: three
hypotheses

Since the groundwater samples in the Otindag are were depleted in their  $\delta D^2H$  and  $\delta^{18}O$  values even more than those of the modern local rainfall (Fig. 76), they must be sourced from other waters characterized by similar with same or more depleted signals in their stable isotopes compositions. Due to the temperature effect (such as evaporation) on isotopic fractionation, only the waters issued from colder environments can be more depleted in their  $\delta D$  and  $\delta^{18}O$  values even more than those of the

local rainfall.

Because the Otindag Desert is under the control of the EASM climate (Fig. 1), the local rainfall\*
in the desert is mainly sourced from summer precipitation. This can also be illustrated by the seasonal
distributions in annual mean precipitation (Fig. 87a), in annual mean air temperature (Fig. 87b) and in
annual mean water vapor pressure (Fig. 87c) over the last forty years at the two surrounding GNIP
weather stations in Baotou and Tianjin.

Given the hypothesis (1) “the modern winter precipitation”, we can get clues from the isotopic
records of winter precipitation in the Baotou and Tianjin stations. It is shown that the annual mean
values of  $\delta D^2H$  and  $\delta^{18}O$  over the last forty years are more depleted in the winter precipitation
than in the summer precipitation at the Baotou and Tianjin stations (Fig. 8d-ea-b). This isotopic signal
qualifies the suggested that the regional winter precipitation to be was qualified to be a potential source of
groundwaters in the Otindag. However, the limited water amount of the winter precipitation in these
regions seemed to be a question towards its importance as an efficient source of
groundwater because the precipitation amounts and the water vapor pressures (effective moisture) in the
winter months are much lower than those in the summer months at both the Baotou and Tianjin
stations (Fig. 87a and 87c). It indicates that the winter seasons in these regions are relatively
colder and drier but not colder and wetter. A colder-wetter pattern of winter season precipitation is a
necessary condition for winter precipitation to be as a water source for the formation of groundwater
under a summer monsoon climate. This is, because the bigger amounts of summer precipitation will
easily remove or weaken the depleted isotopic signals of winter precipitation in groundwater. In this
regard, view of this consideration, the modern winter precipitation is unlikely to might not be an
important source of groundwater in the Otindag. The hypothesis (1) can be neglected.

As to the hypothesis (2) “the palaeowaters2 formed in colder and wetter periods such as the last
glacial2, it has been proposed to be a potential water source for groundwaters in the wide arid lands of
the world. In fact, The depleted signals of stable isotopes ( $\delta D^2H$  and  $\delta^{18}O$ ) in groundwater have been

recognized in global arid and semi-arid regions, such as the Sinai Desert in Egypt (Gat and Issar, 1974)
(Gat and Issar, 1974), Israel (Gat, 1983) (Gat, 1983), South Australia (Love et al., 1994, 2000) (Love et
al., 1994, 2000), northern China (Ma et al., 2010) (Ma et al., 2010), Saudi Arabia (Bazuhair and
Wood, 1996) and North Africa (Moser et al., 1983; Guendouz et al., 2003). These signals are very often
explained as palaeo-groundwater that recharged by precipitation during past wetter and colder periods
(Love et al., 1994, 2000; Herczeg and Leaney, 2011) (Love et al., 1994, 2000; Herczeg and Leaney,
2011). Gat and Issar (1974) reported that palaeowaters played a central role in the deep aquifers of the
Sinai Desert, with the evidence that groundwater stable isotope compositions ( $\delta^{18}\text{O}$  and  $\delta\text{D}$ ) were
more negative than those of weighted mean contemporary rainfall. Ma et al. (2010) presented data from
groundwater in the aquifer of Jinchang city and the adjacent Gobi desert areas in northern China, which
showed that palaeowaters were depleted in  $^{18}\text{O}$  and  $^2\text{H}$  relative to modern precipitation in the same
region.

In order to identify the role of palaeowater recharge on groundwater in the Otindag, Here we use
the tritium data as a environmental tracer to estimate the groundwater age in the Otindag. The half-life
of tritium is 12.43yr. Based on this decay time and the tritium concentrations in groundwater, the
exponential decay equation can be used to provide a qualitative age indication to interpretate the
regional groundwater flow system (Ma et al., 2010). Due to the lack of tritium data of local
precipitation in the Otindag, we still used the tritium data at the GNIP stations of the Baotou and
Tianjin are also referenced as the background values in precipitation of recent years.

A “piston model (flow)” was used to evaluate the residence time of groundwater in aquifer and
the residual tritium of a water body can be calculated by  $N = N_0 e^{-\lambda t}$  (Yang and Williams, 2003). Where
$N$  = content of residual tritium in water sample,  $\lambda = 0.0565$ , the radioactive decay constant,  $N_0$  =
content of tritium at the time of rainfall and  $t$  = years after precipitation. Based on this equation, the
residual tritium was theoretically calculated and the standard for tritium dating was established for this
study, the content of tritium was measured for seven groundwater samples in the Otindag Desert
(Table 3). As a result, all of which were taken from the wells in the Otindag dune field. To the extent
that the input function and piston model are reasonable approximations, ages of 0-60 years were
obtained for these groundwater samples (Table 5). This which indicates that recent recharge took
place several decades after the peak in global nuclear tests had been several decade years underway.
Based on the relatively high tritium contents and the calculated datings of the groundwater samples in
this study (Table 5), We thus concluded that groundwater is generally not older than 70 years in the
study area. It means The hypothesis (2) that the groundwater in the Otindag are not were palaeowater
recharged during glacial period in the Otindag is not valid.

Both the modern summer and winter precipitation recharge and the palaeowater recharge can be
the hypotheses (1) and (2)-refuted, were proved to be valid, indicating ing that the direct recharge is not
a major mechanism controlling the groundwater recharge in the Otindag.

45. 44. Remote waters recharge on groundwater in the Otindag: Dali Basin

The indirect recharge of groundwater in the eastern Otindag?

Through the above analysis, it seemed that the modern winter meteoric water was not a

volumetrically important source of groundwater in the Otindag, and the groundwater was not recharged
by palaeowaters. Thus, The third hypothesis that “remote/the mountains waters emanate under with
colder and wetter conditions” is further should be considered here as a key souce of groundwater in the
Otindag. In essence, it is an indirect recharge mechanism, as the indirect recharge is defined as as
water originates from remote areas (Healy, 2010; Herczeg and Leaney, 2011) (Healy, 2010) and it
generally occurs through rivers, canals, lakes and flash floodings (Herczeg and Leaney, 2011).

It  worth noting that the values of deuterium and oxygen-18
for groundwater in the north part of the study area
more depleted in  $\delta D^2H$  and  $\delta^{18}O$  than those in the south part (Table 3). It suggests that the
Otindag groundwater  might be potentially recharged by water resources coming from
the northern neighboring catchment , such as the Dali Basin.

In order to estimate the potential linkage between the eastern Otindag and the Dali Basin,
Recently published data of  $\delta D$  and  $\delta^{18}O$ -deuterium and oxygen-18 in groundwaters, lake waters, river
waters and spring water  from the Dali Basin (e.g., Chen et al., 2008; Zhen et al., 2014) were
compiled  in this study and were co-analyzed with the data from the Otindag.

In total, There were totally  About 70 natural water samples from the Dali and Otindag with  $\delta D^2H$
and  $\delta^{18}O$  values  shown in a Craig diagram (Fig. 9). As a result, All of these samples fell on
or lied near the evaporation line EL2 in the Craig diagram (Fig. 9), with a regression equation of  $\delta D$
$^2H = 4.81\delta^{18}O - 21.55$  and a high  correlation coefficient ( $R^2=0.98$ ,  $n=70$ )
) for the Otindag samples.

Compared to the groundwater samples in the Otindag, water samples from the groundwaters,
rivers and springs  the Dali Basin  more depleted in  $\delta^{18}O$  and  $\delta D^2H$  (Fig. 9). Such results
further indicate , in terms of  isotopic , the groundwater in the
Otindag has a close relationship with the natural waters in the Dali Basin,

**822 5.4.1. Linkage of the river water in the Dali and the groundwater in the Otindag**

The similar signals of  $\delta D$  and  $\delta^{18}O$ -deuterium and oxygen-18 between the groundwater in the
Otindag and the river water in the Dali (Fig. 9) point towards the idea that the
groundwater in the Otindag might be sourced from the river water in the Dali Basin, since the Dali has
more depleted isotopic signals in water than the Otindag (Fig. 9).

Considering  the topographical gradient of the elevations between the two regions,
however, river water in the Dali Basin  flow into the eastern Otindag, because the terrain
elevation of the Dali Basin is lower than that of the Otindag (Fig. 1). This is also the reason why the
huge Dali Lake  in the Dali Basin  in the Otindag (Fig. 1).
If there is a hydraulic linkage between the two regions, water should flow from the Otindag into the the
Dali, but not conversely.

A hypothesis that water flows from the Otindag into the Dali Lake has also been proposed by
Yang et al. (2015). They argued that a mega palaeolake in Dali, who was almost twice the size of the

自动设置, 图案: 清除

自动设置

自动设置, 图案: 15% (自动设置
前景, 白色 背景)

自动设置

自动设置

自动设置, 图案: 15% (自动设置
前景, 白色 背景)

自动设置

自动设置

自动设置, 图案: 15% (自动设置
前景, 白色 背景)

自动设置

自动设置, 图案: 15% (自动设置
前景, 白色 背景)

自动设置

自动设置, 图案: 15% (自动设置
前景, 白色 背景)

图案: 15% (自动设置
前景, 白色 背景)

自动设置

自动设置

自动设置

自动设置, 图案: 清除

present Dali Lake in area, was recharged by river systems to its south in the Otindag ca. 4,200 years
ago. After that, due to the catastrophic decrease in precipitation that occurred in monsoonal regions
being experienced catastrophic precipitation decreasing and the groundwater in Otindag being
sapping and captured of the Otindag groundwater by the Xilamulun River flowing eastward, the
Otindag's water was no longer recharging the megalake Dali and left a palaeo channel between the two
regions (Fig. 2). Since then the connection between surface waters in the two regions has been
halted was broken.

In view of the hydraulic gradient, river water in the Dali Basin could not be a recharge source for
groundwater in the Otindag. However, in view of the isotopic gradients, groundwater in the Otindag
could not conversely be the source of river water in the Dali. at present, due to the more depleted
values of deuterium and oxygen-18 in Dali than in Otindag (Fig. 9). Thus, the similar isotopic signals
between the river water in Dali and the groundwater in Otindag indicated that these waters might be
recharged from a common source.

**849 5.4.2 Linkage of groundwaters between the Otindag and the Dali**

Similar isotopic signals also occurred in the groundwaters between the Otindag and the Dali Basin
(Fig. 9). The linkage of groundwaters between the two regions is still unknown at present. In order to
understand the linkage of groundwaters between the two regions, answer this question, we need to
know the potential movement of groundwater in the transition zone of the two regions need to be
known.

Due to the inherent difficulties to directly observe groundwater movement along its hydraulic
gradient under ground, inert isotopic and hydrochemical tracers are often used to identify groundwater
movement (Nakaya et al., 2007), such as chloride, TDS and H-O isotopes, which were used as
environmental fingerprints to indicate groundwater movement in arid lands (Yang and Williams, 2003).
In a theoretical line of groundwater evolution, the chloride in water is readily removed from matrix
materials rather than being precipitated due to its high solubility, thus chloride concentrations tend to
be increased with the increasing of the flow path's length and residence time of groundwater (Lloyd
and Heathcote, 1985). The TDS has a similar trend with chloride in groundwater evolution, but its
tendency might be disturbed due to potential precipitation of certain ions when reaching their saturation
conditions. According to the salination classification of water, all the groundwater samples collected in
this study were fresh water in type (TDS < 1000 mg/L). Thus evident precipitation of major ions can
be considered as could be weak in the Otindag groundwaters.

In this study, a groundwater-sampling project was designed in the field along a N-S section of a
palaeo-channel located at the transition zone between the Dali and Otindag (Figs. 1, 2). The channel is
located near the south distal reach of the Xilamulun River and was named "PCSX" in this study, with its
The north part of the channel, named "as-NPCSX" and, is located at the riverhead of the Xilamulun
River and the south part named "as-SPCSX", is close to the eastern margin of the Yinshan Mountains
(Figs. 1, 2).

Regarding to the topographical gradient in the Otindag, the GPS elevation of the northernmost
sampling site in the NPCSX (g11, about 1317 m a.s.l.) was much lower than that of the southernmost

site in the SPCSX (g1, 1396 m.a.s.l.) (Fig. 2 and Table 1). Regarding to the topographical gradient in
the channel, there is a drop of 11 m is about 80 m meter drop between the NPCSX and the SPCSX.
Under such slope, the underground hydraulic gradient for groundwater flow can be roughly parallel
with that of the surface water flow, namely that the groundwaterflow should move downwards from the
SPCSX area into the NPCSX area. Thus we can speculate that groundwater in the NPCSX would have
higher salinity concentration values of chloride and TDS in concentration than those in the SPCSX
under such flowing direction, if the groundwater was flowing from the SPCSX to the NPCSX.

In order to verify check up this speculation, the actual variations of water thesalinity
environmental tracers(chloride and TDS) were detected along the PCSX section. The sampling site g1
was defined as the initial point and the distances between g1 and other sampling sites along the PCSX
section were calculated, based on their GPS geographical coordinates records measured in the field.
The results are shown in Fig. 10a-b. It is was very clear that the variations of chloride and TDS
concentrations in groundwater did not increase along the palaeo-channel from south to north (Fig.
10a-b). On the contrary, both the values of chloride and TDS are lower in the NPCSX area than
those in the SPCSX area. Such kind of spatial variations in the chloride and TDS values was contradict
to the speculated patterns abovementioned, suggesting that the hydraulic gradient of groundwater
flowing path in this region is not controlled by the topographical gradient between the NPCSX and
SPCSX areas.

a complicated movement of groundwater in the study area. It also indicatesd that the hydraulic
linkage wasweak in the groundwaters between the NPCSX and SPCSX areas.

SThe stable and radioactive isotopic data were also used here as tracers to differentiate the
groundwaters between the two regions. Before we use the stable isotopic signals, however, it is
necessary to think about the effect of evaporation process on the fractionation of stable isotopes.
During the evaporation process, dissolved chloride, the conservative ion, will be enriched along with
the heavy isotopes, which is manifested as a correlation between the chloride concentration and the
deuterium content in groundwater (Sklash and Mwangi, 1991; Taylor and Howard, 1996). Based on this
consideration, a bivariate diagram can be was built using the chloride and deuterium data of the
groundwater samples in this study, as shown in Fig. 11. The Groundwater samples from the PCSX
section showed a very weak correlation between the chloride and deuterium (Fig. 11). This indicates
that the groundwaters studied are werenot strongly affected by evaporation process in a deep degree.

Compared between t

---

## Author Comment (AC2) · 30 Jun 2018

Comments from Referee #2 Interactive comment on "Direct or indirect recharge on groundwater in the middle-latitude desert of Otindag, China?" by Bing-Qi Zhu and Xiao-Zong Ren, Anonymous Referee #2, Groundwater availability in arid and semi-arid regions is one of the key issues in hydrogeology and is becoming even more important because of the expected climate changes. Within this context, the contribution by Zhu and Ren provides an interesting analysis on the possible recharge supporting the availability of significant groundwater resources in the Otindag desert, north-eastern China. The analyses have been carried out using hy-

drogeochemical tracers and isotopic measurements on water samples collected from groundwater, surficial (river, lake, and spring) waters, and precipitation water, as well as in-situ records of temperature, pH, conductivity, and TDS concentration. The various steps implemented by the authors to reject possible hypotheses on the groundwater origin (e.g., water flowing from another nearby arid area, precipitation, paleo-water resources) are presented in detail and discussed. Zhu and Ren concludes that, based on the available evidences, the groundwater resources in this region are recharged by the leakage through the bed on incise rivers bounding the desert to the east and conveying downward the waters originated from the precipitation on Daxinganling Ranges. Hence, an "indirect" recharge is the main mechanism supporting the water availability in the study arid lands. Two are the main weaknesses of this ms: 1) the chemical/isotopic investigations seem not supported by a (at least minimum) knowledge of the hydrogeological setting. This is likely one of the reasons why the analyses carried out by the authors are mainly able to exclude recharge mechanisms, but not definitely explain from where this water is originated. The last part of Section 5.5 provides a list of speculative mechanisms (lines 614-652): how the Xilamulun river can recharge the Dali lake when Fig. 15 shows that the bed of the former is less elevated than that of the latter? What support the "speculation" about the "flash floods" in the southern portion of the desert? How you only "theoretically estimate" the isotopic firm of the precipitation on the Yinshan Ranges? 2) the contribution is over-long. The introduction addresses the topic with a too-wide perspective, concepts are repeated, with verbose descriptions. There are also too many figures that can be fruitfully combined. The English form must be improved too. Moreover, the location of the study area is unclear: Fig 1a is obscure, the various portions of the desert are not provided in the maps shown in Figs. 1b and 2, a large part of the toponymy cited in the text is not added to the maps. Because of this, the ms need a major revision. Interactive comment on Hydrol. Earth Syst. Sci. Discuss., https://doi.org/10.5194/hess-2018-71, 2018.

The authors' responses to the comments from Referee #2

Dear Dr/Professor Referee #2: On behalf of my co-authors, we thank you very much for giving us an opportunity to revise our manuscript. We appreciate you very much for your positive and constructive comments and suggestions on our manuscript (hess-2018-71). We have read your comments carefully and have made revision which marked in red in the revised manuscript. We tried our best to revise our manuscript according to your comments and suggestions one by one. Attached please find the revised version, which we would like to submit for your kind consideration. Thank you and best regards.

1) The chemical/isotopic investigations seem not supported by a (at least minimum) knowledge of the hydrogeological setting. This is likely one of the reasons why the analyses carried out by the authors are mainly able to exclude recharge mechanisms, but not definitely explain from where this water is originated. The last part of Section 5.5 provides a list of speculative mechanisms (lines 614-652): how the Xilamulun river can recharge the Dali lake when Fig. 15 shows that the bed of the former is less elevated than that of the latter? What support the "speculation" about the "flash floods" in the southern portion of the desert? How you only "theoretically estimate" the isotopic firm of the precipitation on the Yinshan Ranges? Our response: AGREE AND CHANGES MADE. We thank you very much for this comment. Yes, any chemical and isotopic investigations need to be supported by knowledge of the regional- and local-scale hydrogeological settings. According to this comment, we have added the specific information about the hydrogeological, geological (tectonic, lithological, sedimentological and structural), geomorphological, stratigraphical settings of the study area in the revised manuscript. Detailed changes and the added information can be seen from the section "2. Regional settings" and the section "4.5 remote water recharge on groundwater in the Otindag: mountains waters" in the revised manuscript (pages 2-4 lines 103-188 and pages 8-9 lines 441-483). Besides, two newly-built figures about the geological and hydrogeological maps of the study area are also provided as auxiliary instructions to illustrate the hydrogeological characteristics of the Otindag Desert in the revised manuscript. These figures are Figs. 2 and 3 in the revised manuscript. With

the help of these newly-added materials we believe that we can definitely and logically explain from where the groundwater in the Otindag is originated. About the Fig. 15 in the original manuscript (at present it is Fig. 11 in the revised manuscript) and the question "how the Xilamulun river can recharge the Dali lake when Fig. 15 shows that the bed of the former is less elevated than that of the latter?", our explanation is that: actually, the elevation of the Xilamulun river channel is not lower than the Dali lake. The recent elevation of the Dali Lake is 1,226 m above sea level (Xiao et al., 2008, J Paleolimnol, 40, 519-528). The elevations of the river samples collected from the Xilamulun River in this study ranges between 1360 and 1374 m (Table 1). The real elevation data (measured by handheld GPS in the field) for the river samples l1, l2, l3, l4, l5, l6 in this study are 1368 m, 1368m, 1365 m, 1366 m, 1360 m and 1374 m (Table 1), respectively. Thus, the elevation of the Xilamulun river channel is about 140 m higher than that of the Dali Lake. In Fig. 15 (Fig. 11 in the revised manuscript), it shows the variation of the topographical elevation along the section S1 (see Fig. 1b) from the upstream of the Dali Lake to the location site of the spring water samples s2. It does not show the elevations of the river samples from the Xilamulun River. Strictly speaking, however, this sketch map (Fig. 15) is likely to cause misunderstanding if we think about the river water but not the spring water. So we specially stated that "Note that no river water samples are shown in this figure" in the figure caption of Fig. 11 in the revised manuscript. About the question "What support the "speculation" about the "flash floods" in the southern portion of the desert?", we have added specific information about the hydrological settings of the flash foods derived from the Yinshan Piedmont in the section "2. Regional settings" in the revised manuscript (see pages 3-4 lines 157-188). About the question "How you only "theoretically estimate" the isotopic firm of the precipitation on the Yinshan Ranges?", we use the words "theoretically estimate" because we have not obtained the precipitation water samples from the Yinshan Mountains in this study. Thus the isotopic firm of the precipitation on the Yinshan Ranges is calculated based on the altitude effect of mountain temperature on stable isotopes fractionation in the original manuscript. It is thus a theoretical estimation. In

order to avoid ambiguity, we deleted the discussion of this "theoretically estimation" in the revised manuscript.

2) The contribution is over-long. The introduction addresses the topic with a too-wide perspective, concepts are repeated, with verbose descriptions. There are also too many figures that can be fruitfully combined. The English form must be improved too. Our response: AGREE AND CHANGES MADE. We thank you very much for this comment. Yes, according to the comment that "the contribution is over-long", we have rewritten the manuscript and made an intensive compression on the length of the paper. At present the number of text words in the revised manuscript has been greatly decreased compared with the original manuscript. According to the comment that "The introduction addresses the topic with a too-wide perspective, concepts are repeated, with verbose descriptions", we have rewritten the introduction section of the manuscript to make the topic being specific and not being too broad in its perspective. We tried our best to avid repeat and verbose descriptions in the revised manuscript whatever on the concept or the context of this section. The detailed changes can be seen in pages 1-2 lines 32-101 in the revised manuscript. According to the comment that "There are also too many figures that can be fruitfully combined", we reduced the number of figures in the revised manuscript by putting some figures together and deleting several figures. At last the revised manuscript has 11 figures compared with the original manuscript that including 15 figures. For example, the Figs. 5, 11, 13, 14a in the original manuscript are deleted in the revised manuscript, and the Figs. 7 and 8, the Figs. 10, 12 and 14a are combined, respectively. In addition, two newly-built figures are added into the revised manuscript according to the first comment from the you (the detailed content of this comment can be seen above). The specific changes and the final results of these figures can be seen in the newly submitted revised manuscript. About the comment that "The English form must be improved too", we are very sorry for our poor and incorrect English writing in the original manuscript. For the shortcomings of the English presentation and the grammatical edit in the first paper, we have checked and revised the whole manuscript carefully to avoid language errors, and finally we have got the

help of a native English speaking professional to check and improve the English quality of the revised manuscript. We believe that the language is now acceptable for the publishing purpose.

Moreover, the location of the study area is unclear: Fig 1a is obscure, the various portions of the desert are not provided in the maps shown in Figs. 1b and 2, a large part of the toponymy cited in the text is not added to the maps. Our response: AGREE AND CHANGES MADE. We thank you very much for this comment. According to this comment, we have revised the Fig. 1a and 1b and Fig. 2 (now it is Fig. 4 in the revised manuscript) to make them clear and make sure that the various portions of the Otindag Desert are provided in the corresponding maps. We tried our best to add each of the toponymy cited in the text to be included in these maps. The specific changes and the final results of these figures can be seen in the newly submitted revised manuscript (Figs. 1-4).

Finally, we want to say that special thanks to you for your good comments. We have tried our best to improve the manuscript and made specific changes in the revised manuscript according to the comments from you one by one. These changes will not influence the content and framework of the paper. And here we did not list the changes but marked in red in the revised paper. We hope that the correction will meet with approval. Once again, thank you very much for your comments and suggestions.

Please also note the supplement to this comment:
https://www.hydrol-earth-syst-sci-discuss.net/hess-2018-71/hess-2018-71-AC2-supplement.pdf

![Map figure showing the geographical location of the Otindag Desert in northern China, with panel (a) showing the study area at large scale within China and Inner Mongolia, and panel (b) showing the study area at smaller scale with elevation, rivers, faults, cities and sampling locations.]

**Fig. 1.** Fig. 1. The Geographical location of the Otindag Desert in northern China. (a) The study area shown at a large scale, and (b) the study area shown at a smaller scale, with detailed information about t

Fig. 2. (a) Tectonic framework of the north China-Mongolian segment of the Central Asian Orogenic Belt (modified after Jahn, 2004). (b) Geological sketch map of the northern China-Mongolia tract (modi

**Fig. 3.** Fig. 3. The hydrogeological division map of the Otindag Desert.

Elevation (m)

High : 2037

Low : 932

NPCSX

SPCSX

*Xilamulun River*

g11
g10
g9
g7, g8
g6
g3, g4, g5
l6
g1, g2
r5
s2
r4
r1 s1
l1, l2
l3, l4
l5
r3
p1
r2

0   12.5   25 km

116°40'E   117°0'E   117°20'E

43°0'N

42°40'N

42°20'N

● Groundwater ◆ Spring water — Section S1 ▨ Extent of megalake Dali
+ Lake water ▶ Precipitation -- Palaeo channel — The Xar Moron Fault
■ River water — River □ Desert boundary

**Fig. 4.** Fig. 4. The locations of the water sampling sites in this study.

[Figure]

**Fig. 5.** Fig. 5. The fingerprint diagram showing the variations of multiple ions' concentrations in the studied water samples in an equivalent unit. The HCO3+CO3 concentration in the sample p1 was not shown, d

**Fig. 6.** Fig. 6. The Piper diagram showing the relative abundances of major cations and anions in the studied water samples. Major water types are also shown in this diagram.

**Fig. 7.** Fig. 7. The bivariate diagram of $\delta$D and $\delta$18O, i.e. the Craig diagram, for the natural water samples in this study. Different relationships between the groundwaters, lake waters, river waters, spring w

[Figure]

**Fig. 8.** Fig. 8. The seasonal mean distributions of (a) precipitation, (b) surface air temperature and (c) water vapor pressure from the Baotou and Tianjin weather stations (station sites seen in Fig. 1a) in t

**Fig. 9.** Fig. 9. The bivariate diagram of $\delta$D and $\delta$18O, i.e. the Craig diagram, for the natural water samples collected in the Otindag (this study) and the Dali Basin. Different relationships between the ground

**Fig. 10.** Fig. 10. (a) Sketch map showing the relationship between the groundwaters in the NPCSX and SPCSX areas, based on variations of (a) the chloride concentrations, (b) the TDS concentrations, (c) the $\delta 18O$

[Figure]

**Fig. 11.** Fig. 11. Variation of the topographical elevation along the section S1 (see Fig. 1b) from the upstream of the Dali Lake to the location site of the spring water sample (s2) in the riverhead of the Xil

**Supplement:**

Groundwater availability in arid and semi-arid regions is one of the key issues in hydrogeology and is becoming even more important because of the expected climate changes. Within this context, the contribution by Zhu and Ren provides an interesting analysis on the possible recharge supporting the availability of significant groundwater resources in the Otindag desert, north-eastern China. The analyses have been carried out using hydrogeochemical tracers and isotopic measurements on water samples collected from groundwater, surficial (river, lake, and spring) waters, and precipitation water, as well as in-situ records of temperature, pH, conductivity, and TDS concentration. The various steps implemented by the authors to reject possible hypotheses on the groundwater origin (e.g., water flowing from another nearby arid area, precipitation, paleo-water resources) are presented in detail and discussed. Zhu and Ren concludes that, based on the available evidences, the groundwater resources in this region are recharged by the leakage through the bed on incise rivers bounding the desert to the east and conveying downward the waters originated from the precipitation on Daxinganling Ranges. Hence, an "indirect" recharge is the main mechanism supporting the water availability in the study arid lands.

Two are the main weaknesses of this ms: 1) the chemical/isotopic investigations seem not supported by a (at least minimum) knowledge of the hydrogeological setting. This is likely one of the reasons why the analyses carried out by the authors are mainly able to exclude recharge mechanisms, but not definitely explain from where this water is originated. The last part of Section 5.5 provides a list of speculative mechanisms (lines 614-652): how the Xilamulun river can recharge the Dali lake when Fig. 15 shows that the bed of the former is less elevated than that of the latter? What support the "speculation" about the "flash floods" in the southern portion of the desert? How you only "theoretically estimate" the isotopic firm of the precipitation on the Yinshan Ranges? 2) the contribution is over-long. The introduction addresses the topic with a too-wide perspective, concepts are repeated, with verbose descriptions. There are also too many figures that can be fruitfully combined. The English form must be improved too. Moreover, the location of the study area is unclear: Fig 1a is obscure, the various portions of the desert are not provided in the maps shown in Figs. 1b and 2, a large part of the toponymy cited in the text is not added to the maps. Because of this, the ms need a major revision.

**The authors' responses to the comments from Referee #2**

Dear Dr/Professor Referee #2:

On behalf of my co-authors, we thank you very much for giving us an opportunity to revise our manuscript. We appreciate you very much for your positive and constructive comments and suggestions on our manuscript (hess-2018-71). We have read your comments carefully and have made revision which marked in red in the revised manuscript. We tried our best to revise our manuscript according to your comments and suggestions one by one. Attached please find the revised version, which we would like to submit for your kind consideration. Thank you and best regards.

1) The chemical/isotopic investigations seem not supported by a (at least minimum) knowledge of the hydrogeological setting. This is likely one of the reasons why the analyses carried out by the authors are mainly able to exclude recharge mechanisms, but not definitely explain from where this water is originated. The last part of Section 5.5 provides a list of speculative mechanisms (lines 614-652): how the Xilamulun river can recharge the Dali lake when Fig. 15 shows that the bed of the former is less elevated than that of the latter? What support the "speculation" about the "flash floods" in the southern portion of the desert? How you only

"theoretically estimate" the isotopic firm of the precipitation on the Yinshan Ranges?

Our response: AGREE AND CHANGES MADE.

We thank you very much for this comment. Yes, any chemical and isotopic investigations need to be supported by knowledge of the regional- and local-scale hydrogeological settings. According to this comment, we have added the specific information about the hydrogeological, geological (tectonic, lithological, sedimentological and structural), geomorphological, stratigraphical settings of the study area in the revised manuscript. Detailed changes and the added information can be seen from the section "2. Regional settings" and the section "4.5 remote water recharge on groundwater in the Otindag: mountains waters" in the revised manuscript (pages 2-4 lines 103-188 and pages 8-9 lines 441-483). Besides, two newly-built figures about the geological and hydrogeological maps of the study area are also provided as auxiliary instructions to illustrate the hydrogeological characteristics of the Otindag Desert in the revised manuscript. These figures are Figs. 2 and 3 in the revised manuscript. With the help of these newly-added materials we believe that we can definitely and logically explain from where the groundwater in the Otindag is originated.

About the Fig. 15 in the original manuscript (at present it is Fig. 11 in the revised manuscript) and the question "how the Xilamulun river can recharge the Dali lake when Fig. 15 shows that the bed of the former is less elevated than that of the latter?", our explanation is that: actually, the elevation of the Xilamulun river channel is not lower than the Dali lake. The recent elevation of the Dali Lake is 1,226 m above sea level (Xiao et al., 2008, J Paleolimnol, 40, 519-528). The elevations of the river samples collected from the Xilamulun River in this study ranges between 1360 and 1374 m (Table 1). The real elevation data (measured by handheld GPS in the field) for the river samples l1, l2, l3, l4, l5, l6 in this study are 1368 m, 1368m, 1365 m, 1366 m, 1360 m and 1374 m (Table 1), respectively. Thus, the elevation of the Xilamulun river channel is about 140 m higher than that of the Dali Lake. In Fig. 15 (Fig. 11 in the revised manuscript), it shows the variation of the topographical elevation along the section S1 (see Fig. 1b) from the upstream of the Dali Lake to the location site of the spring water samples s2. It does not show the elevations of the river samples from the Xilamulun River. Strictly speaking, however, this sketch map (Fig. 15) is likely to cause misunderstanding if we think about the river water but not the spring water. So we specially stated that "Note that no river water samples are shown in this figure" in the figure caption of Fig. 11 in the revised manuscript.

About the question "What support the "speculation" about the "flash floods" in the southern portion of the desert?", we have added specific information about the hydrological settings of the flash foods derived from the Yinshan Piedmont in the section "2. Regional settings" in the revised manuscript (see pages 3-4 lines 157-188).

About the question "How you only "theoretically estimate" the isotopic firm of the precipitation on the Yinshan Ranges?", we use the words "theoretically estimate" because we have not obtained the precipitation water samples from the Yinshan Mountains in this study. Thus the isotopic firm of the precipitation on the Yinshan Ranges is calculated based on the altitude effect of mountain temperature on stable isotopes fractionation in the original manuscript. It is thus a theoretical estimation. In order to avoid ambiguity, we deleted the discussion of this "theoretically estimation" in the revised manuscript.

2) The contribution is over-long. The introduction addresses the topic with a too-wide perspective, concepts are repeated, with verbose descriptions. There are also too many figures that can be fruitfully combined. The English form must be improved too.

Our response: AGREE AND CHANGES MADE.

We thank you very much for this comment. Yes, according to the comment that "the contribution is over-long", we have rewritten the manuscript and made an intensive compression on the length of the paper. At present the number of text words in the revised manuscript has been greatly decreased compared with the original manuscript.

According to the comment that "The introduction addresses the topic with a too-wide perspective, concepts are repeated, with verbose descriptions", we have rewritten the introduction section of the manuscript to make the topic being specific and not being too broad

in its perspective. We tried our best to avid repeat and verbose descriptions in the revised manuscript whatever on the concept or the context of this section. The detailed changes can be seen in pages 1-2 lines 32-101 in the revised manuscript.

According to the comment that "There are also too many figures that can be fruitfully combined", we reduced the number of figures in the revised manuscript by putting some figures together and deleting several figures. At last the revised manuscript has 11 figures compared with the original manuscript that including 15 figures. For example, the Figs. 5, 11, 13, 14a in the original manuscript are deleted in the revised manuscript, and the Figs. 7 and 8, the Figs. 10, 12 and 14a are combined, respectively. In addition, two newly-built figures are added into the revised manuscript according to the first comment from the you (the detailed content of this comment can be seen above). The specific changes and the final results of these figures can be seen in the newly submitted revised manuscript.

About the comment that "The English form must be improved too", we are very sorry for our poor and incorrect English writing in the original manuscript. For the shortcomings of the English presentation and the grammatical edit in the first paper, we have checked and revised the whole manuscript carefully to avoid language errors, and finally we have got the help of a native English speaking professional to check and improve the English quality of the revised manuscript. We believe that the language is now acceptable for the publishing purpose.

Moreover, the location of the study area is unclear: Fig 1a is obscure, the various portions of the desert are not provided in the maps shown in Figs. 1b and 2, a large part of the toponymy cited in the text is not added to the maps.

Our response: AGREE AND CHANGES MADE.

We thank you very much for this comment. According to this comment, we have revised the Fig. 1a and 1b and Fig. 2 (now it is Fig. 4 in the revised manuscript) to make them clear and make sure that the various portions of the Otindag Desert are provided in the corresponding maps. We tried our best to add each of the toponymy cited in the text to be included in these maps. The specific changes and the final results of these figures can be seen in the newly submitted revised manuscript (Figs. 1-4).

Finally, we want to say that special thanks to you for your good comments. We have tried our best to improve the manuscript and made specific changes in the revised manuscript according to the comments from you one by one. These changes will not influence the content and framework of the paper. And here we did not list the changes but marked in red in the revised paper. We hope that the correction will meet with approval. Once again, thank you very much for your comments and suggestions.

**Direct or indirect recharge on groundwater in the middle-latitude desert of Otindag, China?**

Bing-Qi Zhu[1*], Xiao-Zong Ren[2], Patrick Rioual[3]

[1]KLWCRESP, IGSNRR, CAS, Beijing, China

[2]SGS, TYNU, Jinzhong, China

[3]KLCGE, IGGCAS, Beijing, China

*Correspondence to*: Bing-Qi Zhu (zhubingqi@sina.com)

**Abstract.** The Otindag Desert is essential to livestock-economy and ecoenvironment of northern China. Although surface water is the traditional source for China's socio-economy in arid areas, the groundwater resources underlying the desert are increasingly burdened by groundwater pumping, which increases interest in the status of the groundwater resources. Widespread fresh groundwater deep to 60 m was found at the eastern part of the Otindag Desert. The occurrence of this massive fresh groundwater raises doubts on the often-made assumption in the literature that regional atmospheric precipitation or palaeowater, namely the direct recharge, is the source of water in the middle-latitude desert aquifers of northern China and makes further investigation necessary. Knowledge on the origin and recharge of this fresh groundwater is key in assessing the possibility of groundwater exploitation and utilization. In this study we conducted hydrogeochemical and isotopical analyses to assess possible origin and recharge of these groundwaters. It is concluded that the fresh groundwater can neither originate from regional atmospheric precipitation derived from the Asian Summer Monsoon system, nor from palaeowater that formed during the last glacial period. Our results indicate that with groundwater dating it is possible to originate from remote mountain areas via the faults of the Solonker Suture zone, including the Daxing'Anlin and Yinshan Mountains. Furthermore, it is deduced that the hydrological connection between desert aquifers and mountain systems through the suturezone is crucial to the hydrogeological functioning of the Otindag aquifer.Thissuggests that the modern indirect recharge mechanism, instead of the direct recharge and the palaeo-water recharge,is the most significant for groundwaterrecharge in the Otindag Desert. This study provides a new perspective into the origin and evolution of groundwater resourcesin the middle-latitude desert zone of Asian continent.

**Keywords:** fresh groundwater recharge; atmospheric precipitation; direct recharge; indirect recharge; palaeowater recharge; fault hydrology; middle-latitude desert; Otindag Desert.

**1. Introduction**

In a semi-arid to arid region where rainfall is insufficient to supply the needs of a growing population and a higher standard of living, the deficit is normally made up by extracting groundwater. Many areas in the middle-latitude desert zone of northern China such as the Badanjilin Desert, the Mu US sandy Land and the Hobq Desert (Chen et al., 2012a; Chen et al., 2012b), are unexpectedly rich

with large groundwater resources although they have been under arid or hyper-arid climate for a long time (Sun et al., 2010). How these groundwaters originated and how they are recharged in these deserts are thus fundamental scientific questions. Until now, however, no consensus has been achieved in academic circles.

The Otindag Desert is one of the largest sandy lands located at the monsoon margin of northern China and is the geographical centre of the northeastern Asian Continent (Fig. 1), which can be regarded as a significant repository of information relating to the groundwater recharge in the arid Inner Asia. At present, the eastern Otindag is also a typical case for its unexpected groundwater resources, because there is abundant groundwater in this desert land and even rivers originate there due to the spillover of spring water, such as the tributaries of Xilamulun River in its north and the Shandian River in its south (Fig. 1). Climatically, the monsoon margin of northern China refers to a strip along the present East Asian Summer Monsoon (EASM) limits and is considered to be sensitive to climate change (Wang and Feng, 2013). Geologically, the Otindag Desert lies in a tectonic depression of the central Solonker suture zone with a few faults stretching east and west (Fig. 2), with its northern margin along a fault marked by a series of lake basins. Thus, the large-scale hydrogeological conditions of the Otindag Desert belong to a fault zone under the influence of the EASM climate.

Until now, however, whether the climate or other factors affected the groundwater recharge in the Otindag are still not known. Little data about the groundwater and its origin is available in the literature, and knowledge and reliable data on various hydrogeological characteristics of the desert such as the catchment extent, input/output, the hysteretic hydraulic functions, the transient hydraulic conditions, in-homogeneities, and on transfer functions to overcome scale problems are also missing. Under such conditions, conventional methods such as water balance and hydraulic methods sometimes fail in determining groundwater recharge, particularly in extreme environments (arid, semi-arid, or cold) (Drever, 1997). Because pristine aquatic conditions may significantly differ from managed conditions in arid environment, and thus groundwater recharge is not a fixed number, but may vary with the boundary conditions of the recharge system (Seiler and Gat, 2007).

Groundwater recharge can be broadly classified into two categories: the direct recharge by native water resources and the indirect recharge by external water resources (Herczeg and Leaney, 2011). Water infiltration of atmospheric precipitation through the unsaturated zone to the groundwater is hydrologically defined as the direct recharge, and the indirect recharge is defined as recharge from mappable features such as rivers, canals, lakes and originates from remote areas (Scanlon et al., 2006; Healy, 2010). It is well known that groundwater recharge can be influenced by environmental factors, including climate change, underlying soil and geology, land cover and the growth in human population that affects withdrawal and economic development (Zhu et al., 2015, 2017). Among these environmental factors, climate and land cover largely determine precipitation and evapotranspiration, whereas the underlying soil and geology dictate whether a water surplus (precipitation minus evapotranspiration) can be transmitted and stored in the subsurface (Doll, 2008, 2009; Giordano, 2009).

For some earth scientists, the direct recharge is thought to be very important for groundwaters in the wide desert lands of north China due to the lack of surface runoffs (Yang et al., 2010; Yang and Williams, 2003; Zhao et al., 2017). They argued that although the amount of atmospheric precipitation

77    is small, the vast catchment area in the desert region could concentrate the rainfall into large inland

78    basins, creating an aquifer with large storage capacity and great thickness. However, some hydrologists

79    estimated by the chloride mass balance method that the direct recharge was 1.4 mm/year, which

80    represents approximately only 1.7% of the mean annual precipitation in a cold large desert (Badanjilin)

81    in northern China (Gates et al., 2008). A similar estimation of 1 mm/year was given for Gobi deserts

82    from the Hexi Corridor to the Inner Mongolia Plateau in northwestern China (Ma et al., 2008).

83    Consequently, they thought that heavy potential evaporation and little precipitation make it difficult for

84    direct recharge to meet the supply of groundwater in these desert areas. Thus, the indirect recharge is

85    considered to be an important mechanism for groundwater recharge in these desert areas. For example,

86    Zhao et al. (2012) suggested that little precipitation had recharged into groundwaters in the Badain

87    Jaran Desert. Chen et al. (2004) argued that the groundwaters in the Badanjilin Desert were recharged

88    by palaeo-glacial melt water through faults and deep carbonate layers far away from the local desert.

89    Many studies also suggested that palaeowaters stored in an aquifer during wetter climate periods could

90    recharge to groundwater under certain conditions in arid lands (Edmunds et al., 2006; Ma and Edmunds,

91    2006). Other kinds of indirect recharge, such as mountain front recharge from adjacent mountain

92    blocks, are also proposed to offer an important inflow to aquifers within arid to semiarid catchments

93    (Blasch and Bryson, 2007).

94        In this paper, we focus to answer the question that whether groundwater recharge in Otindag is

95    mainly direct or indirect, using hydrochemical and isotopic indicators as tracers to offer a valuable

96    support for identifying the contributions of preicipitation recharge on groundwater, since these

97    indicators reflect the composition of water molecules and are sensitive to physical processes such as

98    mixing and evaporation (Sultan et al., 2000; Guendouz et al., 2003; Petrides et al., 2006; Scanlon et al.,

99    2006; Zhu et al., 2007, 2008; Jobbágy et al., 2011). The detailed objectives are: (1) to recognize the

100   major sources of groundwater in the area, and (2) to identify the key mechanism of groundwater

101   recharge in the desert.

103   **2.Regional settings**

104        Geographic setting. The Otindag Desert lies between latitudes 42° and 44°N and longitudes 112°

105   and 118° E (Fig. 1). It forms a part of the great middle-latitude desert belt in northern China which

106   stretches from the Taklamakan Desert of northwestern China to the Kelqin Desert of northeastern

107   China, near the west coast of the Pacific Ocean. The desert has an area of approximately 21,400 square

108   kilometers located in the eastern Inner Mongolia and at the monsoon margin of northern China (Fig. 1).

109   It is the fourth largest sandy lands in China (Yang et al., 2012) and is bordered by a flat steppe terrain

110   of Dali Basin to the north, the Yinshan Mountains and mountainous loess landscape to the south, and

111   the the Greater Khingan (Daxing'Anling) Mountains to the east (Fig. 1). The Otindag Desert is

112   essential to livestock-economy and ecoenvironment of northern China. Settlements in this desert are

113   restricted to areas to permanent springs, shallow groundwater and oases to areas where irrigation is

114   possible. Some nomads continue to eke out a precarious existence grazing livestock in the desert.

115        Topography and geomorphology. The Otindag Desert has a varied relief, combining extensive

116   dune fields with rugged mountains along the eastern, southern and southeastern rims. In the east, the

Daxing'Anling Mountains stretch from the Heilong River Valley into the upper reach valleys of the Xilumulun River from northeast to southwest, gradually increasing in height northwards from about 180 m near Huma to Huanggangliang, where the highest peaks reach 2,029 m with an average elevation range from 1,100 to 1,400 m. In the south and southeast, the Yinshan Mountains decline gradually near Duolun and Zhenglanqi, and in some areas leave wide alluvial plains. The terrain of the Otindag Desert is less rough and elevations decrease from ca. 1300 m in the southeast to ca. 1000 m in the northwest. Over the greater part of this desert the ground cover consists of fixed and semi-fixed sandy dunes, with a few mobile dunes in area of little vegetation. The dominated dune types are represented from parabolic to barchans, linear and grid-formed types, ranging from a few meters to over 40 m in height (Zhu et al., 1980; Yang et al., 2008).

Climate, vegetation and soil. The climate of the Otindag Desert was not uniform in geological period, with much sand movement, occasional rainy years, and several wetter intervals during the Holocene (Yang et al., 2015; Tian et al., 2017). At present the whole desert belongs to the arid and semi-arid temperate zone, with a meanannual temperature of 2 °C in the north and 4°Cin the south (Liu and Yang, 2013). At the regional scale, the climate of the desert is typically controlled by the East Asian Monsoon system, characterized by a warm summer, with precipitation transported by the EASM, and by a cold and dry winter under the influence of the East Asian Winter Monsoon (EAWM). The rainfall in the desert exhibits a wide variation in space and time. Influence of the EASM changes from southeast to northwest in the desert, and varies with latitude and distance from the Pacific Ocean, leading to the mean annual rainfall decreasing from ~450 mm in the southeast to ~150 mm in the northwest (Yang et al., 2013). This uneven distribution of precipitation has a major influence on the availability of near-surface moisture, consequently on the distribution of vegetation, soil and the animal husbandry potential of local communities. The basic soil cover consists of grey desert soil in the west and changes to sierozems and chernozem or chestnut soil in the east. Through the desert, vegetation is sparse in the west and relatively abundant in the east. The natural vegetation is characteristic of desert or semi-deserts, with scrub woodland in the east and steppe in the west. Due to the scarcity of surface water, the growing season is affected by temperature, rainfall and elevation, and hence cultivation is restricted mainly to flood plains.

Geology. The Otindag Desert is located in a tectonic depression of the Solonker Suture Zone (Jian et al., 2010) bounded by the Northern Early to Mid-Paleozoic Orogen Zone and the Hatug Uul Block to the north, the Southern Early to Mid-Paleozoic Orogen Zone and the North China Craton system to the south (Fig. 2). A few faults such as the Xar Moron Fault and Chifeng-Bayan Obo Fault stretch east and west, with its northern margin along the Solonker Suture Zone marked by a series of lake basins (Figs. 1 and 2). The tectonostratigraphic units and overall structural trends are mainly oriented NE–SW (Fig. 2), which may be interpreted as resulting from overall compressive stresses oriented principally in the NW–SE quadrants during orogenesis (Jian et al., 2010; Zhang et al., 2015). Diverse rock types from unlithified and lithified clastic sediments through to carbonate, crystalline, and volcanic rocks are distributed in and around the Otindag Desert (Zhang et al., 2015) (Figs. 2 and 3). Tertiary and Quaternary sandstones and mudstones are the common basement rocks under the dunes of the Otindag, and extensive volcanic basalts forming flat terrains are to the north (Zhu et al., 1980; Li et al., 1995).

157    Hydrology and hydrogeology. The Otindag Desert originated during the Late Quaternary (Yang
158  et al., 2015) and various alluvial fans formed at the margins of this desert during the early to middle
159  Holocene. These are composed of conglomerate and sand deposits, where major periodic steams or
160  wadis debouched into the Otindag. At present two rivers run through the eastern margin of the Otindag
161  Desert, i.e. the Xilamulun River in the north and the Shandian River and its two tributaries, the Shepi
162  River and Tuligen River in the south. Both stem from the eastern and southeastern parts of the Otindag
163  (Fig. 1). The Xilamulun River, 380 km in length and $32.54 \times 10^3$ km$^2$ in area, is a neighboring river both
164  to the northeastern Otindag and the southeastern Dali Basin, the northern catchment of the Otindag
165  Desert. The Xilamulun River flows to the east and finally goes into the Xiliao River, with an annual
166  mean runoff of $6.58 \times 10^8$ m$^3$  (Wu et al., 2014). The Shandian River is the upper reach of the Luan
167  River, with a length of 254 km and a catchment area of $4.11 \times 10^3$ km$^2$  (Yao et al., 2013). Along the low,
168  flat and sandy shorelines of some lakes in the Otindag, salt flats or sabkhas have formed in shallow
169  depressions. Due to the high rate of evaporation, salt crusts develop which have been locally exploited
170  where the salt is relatively free from sand. During rainy season, some rain and floodwaters (generally
171  coming from the Yinshan piedmonts) are retained in low-lying areas, which may temporarily recharge
172  shallow aquifers. Under storm conditions, occasional heavy, short rainstorms cause floods in soil-rich
173  wadi channels. Under other conditions, sand dunes and sand sheets bury the ground and sabkhas.

174    The Otindag Desert can depend on several water-bearing formations and units (aquifers) for their
175  groundwater resources (Fig. 3). Coarse- to fine-grained sedimentary rocks, magmatic rocks and
176  metamorphic rocks of the Inner Mongolia-Daxing'Anling Orogenic Belt (Zhang et al., 2015) form the
177  major regional aquifer unit (Fig. 3). They are composed mainly of alluvial sediments (mid-Permian
178  Zhesi Formation), melange (Solonker suture zone), A-type granite (early Permian), bimodal volcanic
179  rocks with sedimentary intercalations (early Permian Dashizhai Formation), diorite-quartz
180  diorite-granodiorite rocks (Carboniferous-Permian) and metamorphic complex (predominantly gneiss,
181  early Paleozoic) (Fig. 2). The aquifer is generally unconfined in dune fields of the Otindag Desert,
182  unconfined to semi-confined in the Yinshan Mountains' piedmont, and semi-confined to confined in the
183  Daxing'Anling uplands (Fig. 3). Water-level measurement in June 2010 indicated that the general depth
184  of unconfined groundwater level ranges between 10 to 70 m in the Otindag Desert (Fig. 3). Local
185  granular aquifers in the central desert are composed of coarse fluvial, lacustrine and aeolian sediments,
186  but their extent and thickness vary throughout the watershed (Zhu et al., 1980; Li et al., 1995). The
187  generally coarse-grained texture of the unconsolidated rock formations provides primary porosity in
188  terms of groundwater flow in the desert.

190  **3.Methods**

191    The hydrochemistry of natural water in the Otindag Desert, as related to the prevailing EASM
192  climate, as well as, the dominant topographical, geological (tectonic) and hydrogeological conditions,
193  are discussed here and interpreted, using chemical and isotope analyses of water samples from rain,
194  springs, shallow aquifers and deep aquifers, rivers and lakes, and are represented on relevant graphs
195  and diagrams. Fieldworks took place during the summer season of 2011 and the spring season of 2012.
196  Water samples were mainly retrieved from shallow and deep wells located over a wide area in dune

197   fields of the study regions. The detailed locations of the sampling sites are shown in Fig. 4.

198   Two groups of parameters are measured to characterize the chemistry of any water analysis:
199   field-measured parameters and lab-measured parameters. The filed-measured parameters include
200   temperature (°C), hydrogen-ion concentration (pH), electrical conductivity (EC in micro-Siemens per
201   centimeter or µS/cm) and total dissolved solid (TDS, mg/L). The values of these parameters change
202   when they are not directly measured in the field. The number lab-measured parameters depend on the
203   purpose of study. However, the measurement of major cations ($F^-$, $Cl^-$, $NO_2^-$, $NO_3^-$, $SO_4^{2-}$ $HCO_3^-$,
204   $CO_3^{2-}$ and $H_2PO_4^-$) and anions ($Li^+$, $Na^+$, $NH_4^+$, $K^+$, $Mg^{2+}$ and $Ca^{2+}$) are determined in most chemical
205   analyses. Analysis for stable ($^2H$ and $^{18}O$) and radioactive isotopes ($^3H$) in rain and groundwater are
206   also included. The analytical data of the physiochemical parameters and the stable and radioactive
207   isotopes of the water samples collected in this study are listed in Tables 1, 2 and 3, respectively.

209   **4.Results and Discussions**

211   **4.1.Hydrochemical characteristics of natural waters**
212   The natural water samples collected in this study are generally neutral to slightly alkaline, with the
213   pH values varying between 6.26 and 9.44 (except the precipitation sample p1, 4.61) (Table 1) and a
214   median value of 7.27. The TDS values range between 67 and 660 mg/L (average 211 mg/L) (Table 1),
215   all belonging to fresh water(TDS < 1000 mg/L) in the salination classification of natural water
216   (Meybeck, 2004). The variations in ion concentrations of the major cations and anions in the studied
217   water samples were displayed in a fingerprint diagram with a semi-logarithm y-axis (Fig. 5). The rain
218   water sample is the most depleted in ions among these samples. The groundwater samples have the
219   highest concentrations of cations and anions and the lake, river and spring waters had intermediate
220   values. The calcium concentration is the highest among cations in almost all of the water samples, and
221   the $HCO_3+CO_3$ concentration (bicarbonate + carbonate, alkalinity) is the highest among anions in most
222   of the water samples. For several groundwater samples (g3, g4, g5, g6 and g11), spring sample (s1) and
223   precipitation sample (p1), they have higher $SO_4$ concentrations than alkalinity (Fig. 5).

224   Two chemically distinct water types are recognized for the studied waters via a Piper diagram (Fig.
225   6), calcium bicarbonate and calcium sulphate. No Chloride-type and sodium-type waters occur in the
226   study area (Fig. 6). Based on more than 10,000 chemical analyses of groundwater samples from the
227   world, Chebotarev (1955) observed that the global groundwater tends to evolve chemically towards the
228   composition of seawater. He also observed that this evolution is associated with regional changes in
229   dominant anions but not cations, as the concentration of cations may exhibit a wide range of
230   fluctuations in groundwater and is not as steady as the changes in anion dominance. Freeze and Cherry
231   (1979) illustrated the Chebotarev's (1955) general evolution of groundwater as a anion evolution line:
232   $HCO_3^- \rightarrow HCO_3^- + SO_4^{2-} \rightarrow SO_4^{2-} + HCO_3^- \rightarrow SO_4^{2-} + Cl^- \rightarrow Cl^- + SO_4^{2-} \rightarrow Cl^-$, which travels
233   along the flow paths and increasing ages. On this evolution line, bicarbonate water is generally
234   characteristic of low salinity, renewable water resources and low residence time, while sulphate waters
235   predominate in groundwater passing through gypsum and anhydrite aquifers, and is usually associated
236   with intermediate salinity in unconfined aquifers (Clark, 2015). The distribution pattern of water

237 chemical types occurred in the studied area indicates a primary stage of groundwater evolution in the
238 Otindag Desert.

239     The δD values of the groundwater samples collected in this study varied from -63.42‰ to -75.92‰
240 (Table 3), with an average -69.53‰. The $\delta^{18}O$ values ranged between -8.64‰ and -11.26‰ (Table 3),
241 with an average -10.17‰. The spring water samples were relatively concentrated in δD and $\delta^{18}O$ and
242 were greatly similar to those of the groundwater samples (Fig. 7). The δD and $\delta^{18}O$ values in the river
243 water samples were slightly more variable and were also similar to those of the groundwater (Fig. 7).
244 The lake water samples were enriched in δD and $\delta^{18}O$ by comparison to the groundwater samples (Fig.
245 6). The precipitation sample p1 was also enriched in δD and $\delta^{18}O$ by comparison to the groundwater
246 samples (Fig. 7). The content of radioactive isotope of tritium ($^3H$) measured in seven well
247 groundwater samples with 6-60 m depth ranged from 1.86 to 24.35 TU (Table 3), with an average
248 14.95 TU, higher than the mean tritium concentration (9.8 TU) of groundwater in the Vienna Basin,
249 Austria (Stolp et al., 2010), the seat of the International Atomic Energy Agency (IAEA).

250     If we plot the relationships between oxygen and hydrogen isotopes of groundwater, spring, river
251 and lake water samples, we observed that the regression line that fits all data points can be described by
252 the equation: $\delta D = 4.09\delta^{18}O - 28.31$ ($R^2$=0.93, n=24) (EL1 in Fig. 7). This local groundwater line
253 (LGWL) is different from the Global Meteoric Water Line (GMWL, $\delta D = 8\delta^{18}O + 10$) and the
254 Mediterranean Meteoric Water Line (MMWL, $\delta D = 8\delta^{18}O + 20$) estimated by Craig (1961), but it is
255 similar to the local groundwater lines established for other deserts in northern China and central Asia
256 with a same slope but different Y-intercepts, such as $\delta D = 4.17\delta^{18}O - 31.3$ for the Badanjilin Desert (Jin
257 et al., 2018), $\delta D = 4.8\delta^{18}O - 15.2$ for the Ejina Desert in China (Wang et al., 2013), and $\delta D = 4.26\delta^{18}O$
258 + 9.23 for the Rub Al Khal Desert in the United Arab Emirates (Rizk and El-Etr, 1997). The scatter of
259 stable isotope data points for the lake water samples (Fig. 7) in the Otindag suggests that the lake
260 waters are affected by evaporation, but the other waters in the desert are not so.

261

262 **4.2. Local precipitation recharge on groundwater in the Otindag**

263     To incorporate the isotopic analysis of precipitation with similar areas in the studied area, local
264 data (p1) was plotted with those of Baotou (Fig. 7). The isotopic composition of rainfall in Baotou, the
265 nearest long-term station to the Otindag Desert, was monitored for the period 1986-2001 within the
266 scope of the International Atomic Energy Agency/World Meteorological Organization (IAEA/WMO)
267 global survey. The stable isotope data available from this station was used to provide basic
268 characteristics of the stable isotopic composition of the present-day meteoric water, especially in the
269 westward inland areas of the Otindag Desert (Fig. 1). Stable isotope data of the Tianjin station was also
270 used to characterize precipitation of the eastern coastal areas of the Otindag Desert (Fig. 1).

271     Based on the isotopic data from the Baotou station, the local meteoric water lines can be
272 statistically expressed as the isotopic regression equation of $\delta D = 6.36\delta^{18}O - 5.21$ (LMWL-B). It can
273 also be expressed as $\delta D = 6.57\delta^{18}O + 0.31$ (LWML-T), based on the data from the Tianjin station (Fig.
274 7). The precipitation sample p1 collected in this study fell onto the GMWL (Fig. 7). It also showed
275 similar δD and $\delta^{18}O$ values to those of the precipitation collected in the GNIP stations of Baotou and
276 Tianjin (Fig. 7).

277    Compared to the precipitation data from the GNIP stations and from the local precipitation (p1),
278    the groundwater, spring, and river water samples were evidently depleted in heavy stable isotopes in
279    the Otindag (Fig. 7). Except for the lake water samples, most of the groundwater, river water and
280    spring water samples in the Otindag fall on or lay between the LMWL-B and the LMWL-T lines, and
281    are located at the lower left area of the precipitation points (Fig. 7).

282    Because the isotopic evolution of $\delta D$ and $\delta^{18}O$ in water illustrated in the Craig line represents a
283    one-way and irreversible process, the water bodies distributed at the upper right area of the Craig line
284    can not be recharge sources for the water bodies distributed at the lower left area of the line. Such
285    results indicate that the groundwater, river water and spring water in the Otindag are not recharged by
286    the regional precipitation, namely no significant modern direct recharge has taken place for
287    groundwater in the Otindag.

288    Dogramaci et al. (2012) documented that only intense and remarkable rainfall events >20 mm
289    could recharge groundwater in the semi-arid Hamersley Basin of northwest Australia, while the rainfall
290    events <20 mm had limited influences on groundwater recharge. Chen et al. (2014) described that
291    rainfall events ≤5 mm in the arid and semi-arid region of northern China would be evaporated into
292    the atmosphere rapidly before it is infiltrated into the groundwater system. Based on the analysis on the
293    data records from two meteorological stations around the Otindag, i.e.the Duolun and Xilinhaote
294    stations (see Fig. 1a), we observed that rainfall events >20 mm on average only occur 2.5-3.4 times per
295    year (Table 4). In some years (e.g. from 2005 to 2007 at the Xilinhaote Station), no rainfall events >20
296    mm even occurred. It further indicated the limited contribution of regional precipitation on
297    groundwater recharge in the Otindag.

298    In addition to groundwater, the river and spring water samples from the Otindag also deviated
299    from the local precipitation in the Craig diagram (Fig. 7).These water samples came from the
300    Xilamulun, Shepi and Tuligen rivers. They shared the same evaporation line (EL1) with the
301    groundwater and lake water samples (Fig. 7). Generally speaking, natural waters that have a same
302    recharge source are distributed on a same line of evaporation in the $\delta^2$ and $\delta^{18}O$ diagram (Chen et al.,
303    2012b). This indicates that the recharge sources of groundwater, river water, spring water and lake
304    water in the Otindag are genetically associated each other and differ from the local precipitation.

305

306    **4.3. Winter precipitation and palaeowater recharge on groundwater in the Otindag**

307    Since the groundwater samples in the Otindag are depleted in their $\delta D$ and $\delta^{18}O$ values even more
308    than those of the local rainfall (Fig. 7), they must be sourced from other waters characterized by similar
309    or more depleted signals in their stable isotopes compositions. Due to the temperature effect (such as
310    evaporation) on isotopic fractionation, only the waters issued from colder environments can be more
311    depleted in their $\delta D$ and $\delta^{18}O$ values even more than those of the local rainfall.

312    Because the Otindag Desert is under the control of the EASM climate (Fig. 1), the local rainfall in
313    the desert is maily sourced from summer precipitation. This can also be illustrated by the seasonal
314    distributions in annual mean precipitation (Fig.8a), in annual mean air temperature (Fig. 8b) and in
315    annual mean water vapor pressure (Fig. 8c) over the last forty years at the two surrounding GNIP
316    weather stations in Baotou and Tianjin. The seaonal distributions of stable isotopes in the two stations

(Fig. 8d-e) show that the summer rainfall is evidently positive in its signals of δD and δ[18]O by comparison with those of the winter rainfall, further suggesting that the waters issued from cold environments can be more depleted in their δD and δ[18]O values than those of the summer rainfall. Thus we speculate that groundwater in the Otindag can be potentially derived from (1) modern precipitation in winter, (2) palaeowater formed in the past glacial period, or (3) remote/mountains waters that emanate in colder and wetter conditions.

The annual mean values of δD and δ[18]O over the last forty years are more depleted in winter precipitation than in summer precipitation at the Baotou and Tianjin stations (Fig. 8d-e). This isotopic signal qualifies the regional winter precipitation to be a potential souce of groundwaters in the Otindag. However, the precipitation amounts and the water vapor pressures (effective moisture) in winter months are much lower than those in the summer months at both the Baotou and Tianjin stations (Fig. 8a and 8c). It indicates that the winter seasons in these regions are relatively colder and drier but not colder and wetter. A colder-wetter winter season is a necessary condition for winter precipitation to be a water source for the formation of groundwater under a summer monsoon climate. This is because the bigger amounts of summer precipitation will easily remove or weaken the depleted isotopic signals of winter prepicitation in groundwater. In this regard, modern winter precipitation is unlikely to be an important source of groundwater in the Otindag.

As to the palaeowaters formed in colder and wetter periods such as the last glacial, it has been proposed to be a potential water source for groundwaters in the wide arid lands of the world. The depleted signals of stable isotopes (δD and δ[18]O) in groundwater have been recognized in global arid and semi-arid regions, such as the Sinai Desert in Egypt (Gat and Issar, 1974), Israel (Gat, 1983), South Australia (Love et al., 1994, 2000), northern China (Ma et al., 2010), Saudi Arabia (Bazuhair and Wood, 1996) and North Africa (Guendouz et al., 2003). These signals are very often explained as palaeo-groundwater that recharged by precipitation during past wetter and colder periods (Love et al., 1994, 2000; Herczeg and Leaney, 2011).

Here we use the tritium data as a environmental tracer to estimate the groundwater age in the Otindag. The tritium data at the GNIP stations of the Baotou and Tianjin are also referenced as the background values in precipitation of recent years. The residence time of groundwater in aquifer and the residual tritium of a water body can be calculated by $N = N_0 e^{-\lambda t}$ (Yang and Williams, 2003). Where N = content of residual tritium in water sample, $\lambda = 0.0565$, the radioactive decay constant, $N_0$ = content of tritium at the time of rainfall and t = years after precipitation. Based on this equation, the residual tritium was theoretically calculated and the standard for tritium dating was established for seven groundwater samples in the Otindag Desert (Table 3). As a result, ages of 0-60 years were obtained for these groundwater samples (Table 5). This indicates that recent recharge took place several decades after the peak in global nuclear tests. We thus conclude that groundwater is generally not older than 70 years in the study area. It means that groundwater in the Otindag are not palaeowater recharged.

Both the modern summer and winter precipitation recharge and the palaeowater recharge can be refuted, indicating that direct recharge is not a major mechanism controlling the groundwater recharge in the Otindag.

357

**4. 4. Remote water recharge on groundwater in the Otindag: Dali Basin**

The third hypothesis that "remote/mountains waters emanate under colder and wetter conditions" is further considered here. In essence, it is an indirect recharge mechanismas as water originates from remote areas (Healy, 2010; Herczeg and Leaney, 2011).

It is worth noting that the values of deuterium and oxygen-18 for groundwater in the north part of the study area are more depleted in $\delta D$ and $\delta^{18}O$ than those in the south part (Table 3). It suggests that the Otindag groundwater might be potentially recharged by water resouces coming from the northern neighboring catchment, such as the Dali Basin.

Recently published data of $\delta D$ and $\delta^{18}O$ in groundwater, lake water, river water and spring water sampled from the Dali Basin (e.g., Chen et al., 2008; Zhen et al., 2014) were compiled in this study and were co-analyzed with the data from the Otindag. About 70 natural water samples from the Dali and Otindag with $\delta D$ and $\delta^{18}O$ values are shown in a Craig diagram (Fig. 9). All of these samples fell on or lied near the evaporation line EL2 in the Craig diragram (Fig. 9), with a regression equation of $\delta D = 4.81\delta^{18}O - 21.55$ and a high correlation coefficient ($R^2$=0.98, n=70). Compared to the groundwater samples in the Otindag, water samples from the groundwaters, rivers and springs from the Dali Basin are more depleted in $\delta^{18}O$ and $\delta D$ (Fig. 9). Such results further indicate that, in terms of itsisotopic signature, the groundwater in the Otindag has a close relationship with the natural waters in the Dali Basin.

The similar signals of $\delta D$ and $\delta^{18}O$ between the groundwater in the Otindag and the river water in the Dali (Fig. 9) point towards the idea that the groundwater in the Otindag might be sourced from the river water in the Dali Basin, since the Dali has more depleted isotopic signals in water than the Otindag (Fig. 9). Considering the topographical gradient of elevations between the two regions, however, river water in the Dali Basin cannot flow into the eastern Otindag, because the terrain elevation of the Dali Basin is lower than that of the Otindag (Fig. 1). This is also the reason why the huge Dali Lake that lies in the Dali Basin has no equivalent in the Otindag (Fig. 1). If there is a hydraulic linkage between the two regions, water should flow from the Otindag into the the Dali, but not conversely.

In view of the hydraulic gradient, river water in the Dali Basin could not be a recharge source for groundwater in the Otindag. However, in view of the isotopic gradients, groundwater in the Otindag could not conversely be the source of river water in the Dali (Fig. 9). Thus, the similar isotopic signals between the river water in Dali and the groundwater in Otindag indicate that these waters might be recharged from a common source.

Similar isotopic signals also occurred in the groundwaters between the Otindag and the Dali Basin (Fig. 9). In order to understand the linkage of groundwaters between the two regions, the potential movement of groundwater in the transition zone of the two regions need to be known. In this study, a groundwater-sampling project was designed in the field along a N-S section of a palaeo-channel located at the transition zone between the Dali and Otindag (Figs. 1, 2). The channel was named "PCSX" in this study, with its north part named "NPCSX" and the south part named "SPCSX".

The GPS elevation of the northernmost sampling site in the NPCSX (g11, about 1317 m a.s.l.) was

397  much lower than that of the southernmost site in the SPCSX (g1, 1396 ma.s.l.) (Fig. 2 and Table 1).

398  Regarding to the topographical gradient in the channel, there is a drop of about 80 m between the

399  NPCSX and the SPCSX. Under such slope, the underground hydraulic gradient for groundwater flow

400  can be roughly parallel with that of the surface water flow, namely that the groundwaterflow should

401  move downwards from the SPCSX area into the NPCSX area. Thus we can speculate that groundwater

402  in the NPCSX would have higher salinity than those in the SPCSX under such flowing direction. In

403  order to verify this speculation, actual variations of water salinity (chloride and TDS) were detected

404  along the PCSX section. The sampling site g1 was defined as the initial point and the distances between

405  g1 and other sampling sites along the PCSX section were calculated, based on their GPS geographical

406  coordinates measured in the field. The results are shown in Fig. 10a-b. It is clear that the variations of

407  chloride and TDS concentrations in groundwater do not increase along the palaeo-channel from south

408  to north (Fig. 10a-b). On the contrary, both the values of chloride and TDS are lower in the NPCSX

409  area than those in the SPCSX area. Such kind of spatial variations in the chloride and TDS values

410  contradict the speculated patterns abovementioned, suggesting that the hydraulic gradient of

411  groundwater flowing path in this region is not controlled by the topographical gradient between the

412  NPCSX and SPCSX areas.

413      Compared between the NPCSX and SPCSX regions, the stable isotopic values ($\delta^{18}$O and $\delta$D) of

414  groundwaters in the SPCSX region vary greatly with a large amplitude, while those in the NPCSX are

415  relatively constant (Fig. 10c-d). The constant variations indicate that the recharge source of

416  groundwater in the NPCSX is relatively unitary. The isotopic values in the SPCSX are much lighter

417  than those in the NPCSX along the distance section from south to north (Fig. 10c-d). The heaviest

418  values occurred in the sample g11 collected from the NPCSX (Fig. 10c-d), indicating a water being

419  earlier recharged. The spring water sample s2, a representation of discharge water, is characterized by

420  medium values of $\delta$D and $\delta^{18}$O. These results indicate that the groundwaters in the SPCSX area, with

421  relatively enriched isotopic signals in $\delta$D and $\delta^{18}$O by comparison with those in the NPCSX area, are

422  composed of a mixture of the groundwaters in the NPCSX with other waters.

423      The tritium contents were broadly and positively related to the values of deuterium excess in the

424  groundwater samples in the PCSX (Fig. 10e). For water that experiences an evaporation process, the

425  d-excess value will increase in the evaporated water vapor, but will decrease in the residual water body

426  (Dansgaard, 1964; Merlivat and Jouzel, 1979). In this study, except for sample g11 (a sample very

427  close to the riverhead area), the positive relationship between the tritium and the deuterium excess

428  generally shows that the d-excess values are higher in the groundwaters collected from the NPCSX, but

429  are lower in those from the SPCSX (Fig. 10e). This distribution pattern indicates that the groundwaters

430  in the NPCSX are relatively younger and experienced a lower degree of evaporation than those in the

431  SPCSX. The d-excess gradient, increasing from south to north in the PCSX, further suggests that

432  groundwater does not flow from the SPCSX area to the NPCSX area, namely out of the topographical

433  control.

434      Many studies (e.g., Boronina et al., 2005; Kazemi et al., 2006) have demonstrated that

435  groundwater flows in the direction in which it gets older. In view of this point, groundwaters in the

436  PCSX region should flow from the NPCSX area to the SPCSX area, in opposition to the S-N

437  topographical gradient between the Otindag and Dali regions. Thus groundwater in the Dali are not the
438  source of groundwater in the Otindag. The similar isotopic signals between groundwaters in the two
439  regions indicate that these waters might be recharged from a common source in other place.
440

441  **4. 5. Remote water recharge on groundwater in the Otindag: mountains waters**
442  The discussions above revealed that both the groundwaters in the Otindag and DaliBasin might be
443  recharged from a common source derived from another place. Considering the third hypothesis
444  abovementioned that "remote/mountains waters emanate under colder and wetter conditions", we
445  propose that this "common source" of the two regions are from mountians areas surrounding the
446  Otindag and Dali Basin.
447  There are two large permanent rivers and lots of small intermittent streams entering the Dali Basin
448  (Xiao et al., 2008), including the Xilamulun River to the south and the Gongger River to the north, both
449  of which are stemming from the Greater Khingan Mountains (Daxing'Anling Mountains in Chinese
450  pinyin, 1,100-1,400 m above seal level) (Fig. 1). The Xilamulun River carries a large amount of water
451  (about $6.58\times10^8$ m$^3$/y) from the Daxing'Anling Mountains flowing through the east margins of the Dali
452  and Otindag (Wu et al., 2014). This is an important clue linking natural waters between the Otindag
453  and Dali Basin.
454  Variation in the elevation from the Dali Lake to the riverhead of the Xilamulun River can be
455  clearly found along a land surface topographical section (Fig. 11). The channel of the Xilamulun River
456  is located in the Xar Moron Fault (Fig. 1), which is a part of the Solonker Suture Zone (Eizenhöfer et
457  al., 2014) or the Xilamulun-Changchun-Yanji plate suture zone (Sun et al., 2004) in the regional
458  tectonical settings (Fig. 2). Outcrop observations indicate that fault zones commonly have a
459  permeability structure suggesting they should act as complex conduit–barrier systems in which
460  along-fault flow is encouraged and across-fault flow is impeded (Bense et al., 2013). Thus the
461  hydraulic grediant of groundwater flow in the Eastern margins of the Otindag and Dali Basin must be
462  controlled by the fault zone hydrogeology. This may be the reason why the hydraulic gradient of
463  groundwater represented by the isotopic and hydrogeochemical gradients of groundwater samples in
464  this study is not consistent with the local topographical gradient in the Otindag Desert. On the other
465  hand, the regional aquifer is generally unconfined in dune fields of the Otindag Desert but
466  semi-confined to confined in the Daxing'Anling uplands (Fig. 3), thus the thick unconsolidated
467  aquifers in the study area (Figs. 3 and 11) will be favourable conditions for groundwater storage and
468  transportation along the Solonker Suture Zone. When rivers stem from the Daxing'Anling
469  Mountainsand flow downward to the marginal areas of the Dali and Otindag, leakage water from these
470  rivers can recharge the desert land through thick unconsolidated aquifers. A strong isotopic evidence is
471  that the lake and river waters in the Dali Basin share the same evaporation line (EL2) with the
472  groundwaters in the PCSX area.
473  Although groundwaters in the SPCSX area are different from those in the NPCSX area, their
474  isotopic data points still fell onto the EL2 (Fig. 9), which further indicates that the groundwaters in the
475  SPCSX are a mixture of waters from the Daxing'Anling Mountain and other sources. Another source
476  for groundwater recharge in the SPCSX could be represented by remote water such as flash floods

coming from the north Yinshan Mountains, because it can be clearly observed from digitial maps that many transient rivers or streams originated from the Yinshan Mountrains flow into the south and southeastern Otindag (Fig. 1). Supportive evidence for this idea can also be observed in the summer rainy season. During rainy days or under storm conditions, occasional heavy, short rainstorms cause floods in soil-rich wadi channels and low-lying depressions in the unconfined to semi-confined areas of the Yinshan Mountains' piedmont. These waters may temporarily recharge shallow aquifers in the SPCSX area.

**5.Conclusions**

In the middle-latitude desert zone of northern China, many deserts such as the Otindag and Badanjilin Deserts, are unexpectedly rich in groundwater resources, although they have no surface runoff and have been under an arid or hyper-arid climate for a long period of time. How groundwaters originated and recharged in these deserts are thus key questions that are still under debate. For some earth scientists, the direct recharge is thought to be very important for groundwaters in the wide desert lands of northern China, due to the lack of surface runoffs. However, groundwater availability is very much a function of the local- and regional-scale geological and climatic settings. To achieve an integrated understanding of the groundwater recharge and its controlling mechanisms is of great significance. In this study, groundwater recharge was explored using multiple environmental tracers in the Otindag Desert of northern China, a region that is under the influence of the East Asian Summer Monsoon (EASM) climate. Compared to modern summer precipitation, the groundwaters, river waters and spring waters are depleted in $\delta D$ and $\delta^{18}O$. All these waters shared a same Craig line, indicating a genetic relationship on their recharge sources. The stable isotopic signals of the groundwaters is more depleted than those of the modern summer precipitation and this suggests that the groundwaters studied could only be sourced from cold water different from the EASM precipitation. In general, the analyses revealed that the highland remote water resources from the Daxing'Anling and Yinshan Mountains were isotopically and geochemically traced to be a major source for the groundwater in the Otindag. It suggests that the modern indirect recharge mechanism, instead of the direct recharge and the palaeo-water recharge, is the most significant for groundwater recharge in the eastern Otindag. This study provides a new perspective into the origin and evolution of groundwater resources in the middle-latitude desert zone of northern China.

**Acknowledgements**

This study was financially supported by the National Natural Science Foundation of China (41771014 and 41602196) and the National Key Research and Development Program of China (2016YFA0601900). We thank the China Meteorological Data Sharing Service system for providing the weather data. Sincere thanks are also extended to Profs. Xiaoping Yang, Xunming Wang, Jule Xiao and other workmates, e.g., Ziting Liu, Hongwei Li, and DeguoZhangfor their generous help in the research work.

[revised manuscript text omitted]

827 **Fig. 8.** The seasonal mean distributions of (a) precipitation, (b) surface air temperature and (c) water
828 vapor pressure from the Baotou and Tianjin weather stations (station sites seen in **Fig. 1a**) in the
829 surrounding areas of the Otindag for the period 1981-2010. The seasonal mean distributions of (d) δ[18]O
830 and (e) δD values in precipitation from the Baotou and Tianjin weather stations in the surrounding
831 areas of the Otindag for the period 1986-2001.

[Figure]

**Fig. 9.** The bivariate diagram of δD and δ¹⁸O, i.e. the Craig diagram, for the natural water samples collected in the Otindag (this study) and the Dali Basin. Different relationships between the groundwaters, lake waters, river waters, spring waters and the precipitation waters are clearly illustrated. AWMB, AWMT, LWMB, LWMT, GMWL, LMWL-B, LWML-T, and EL1 are the same as in Fig. 7. EL2, the evaporation line calculated based on the data from the groundwater, lake water, river water and spring water samples collected from the Otindag and Dali Basin. The data for the Dali were taken from Chen et al. (2008) and Zhen et al. (2014).

[Figure]

**Fig. 10.** (a) Sketch map showing the relationship between the groundwaters in the NPCSX and SPCSX areas, based on variations of (a) the chloride concentrations, (b) the TDS concentrations, (c) the $\delta^{18}O$ values and (d) the $\delta D$ values of these water samples versus their distances away from the water sample g1 along the palaeo river channel (PCSX) from south to north. The dashed line in (c) and (d) represents the corresponding values of the spring water sample s2, and divides samples into the NPCSX and SPCSX parts. (e) Variations of tritium contents vs. deuterium excess for the groundwater samples in the study area. The sample g6 was omitted due to its potential contamination.

[revised manuscript text omitted]

Bing-Qi Zhu[1*], Xiao-Zong Ren[2], Patrick Rioual[3]

[1]KLWCRESP, IGSNRR, CAS, Beijing, China

[2]SGS, TYNU, Jinzhong, China

[3]KLCGE, IGGCAS, Beijing, China

*Correspondence to*: Bing-Qi Zhu (zhubingqi@sina.com)

**Abstract.** Although rainfall is scarce in most desert lands of the world, the Otindag Desert in the middle latitude desert zone of northern China in Northern Hemisphere (NH)had is abundant of water resources, mainly groundwater. To gain an insight into the origin of groundwater origin in this desert, stable and radioactive isotopes and major ion hydrochemistry of groundwater, as well as other natural waters including river water, spring water, lake water and precipitation water, were investigated in the eastern part of the Otindag. The results showed that the groundwaters in the Otindag arewere freshwater (TDS < 700 mg/L) and were depleted in $\delta^2H$ and $\delta^{18}O$, when compared with the modern precipitation. The major water types arewere the Ca HCO$_3$ and Ca/Mg SO$_4$ typeswaters. No Cl-type and Na-type waters occurred in the study area. The ionic and depleted stable isotopic signals in groundwater, as well as the high values contents in of tritium contents (5-25 TU), indicated that thesegroundwatersstudiedwereareyoung but not of meteoric origin, i.e., out of control by the modern and palaeo direct recharge. Clear differencesin the isotopic signals werer observed between the groundwaters in the north (NPCSX) and south (SPCSX) parts of the study area, but the signals were silimar inbetweenthe groundwaters betweeninthe NPCSX and itsneighbouring catchment, the Dali Basin. The topographical elevation is decreasesing from the SPCSX (1396 m a.s.l.) to the NPCSX (1317 m a.s.l.) and the Dali (1226 m a.s.l.).Groundwaters in the NPCSX arewere characterized by lower elevations, the lower chloride and TDS concentrations, higher tritium contents, higher deuterium excess, and more depleted values of $\delta^2H$ and $\delta^{18}O$ than those in the SPCSX. Thespatial distribution pattern of these environmental parametersThisindicatesda discrepancydisunitybewtweenin the one hand,the hydraulic gradient of groundwater andin the other hand,the isotopic and hydrochemicalgradients of groundwater in the deserteastern Otindag. It also, suggestsingthat the groundwaters havedifferent waterrecharge sources between the two areasparts in the study area. However, the groundwaters in the two areas shared a common evaporation line (EL2) in the Craig diagram of $\delta^2H$ and $\delta^{18}O$, indicating a genetic relationship in their recharge sources.Combined analysis was further performedusing the isotopic and physio-chemicaldata of natural waters collected fromthe Dali Basin and the surrounding mountains.It indicated that the major rechargesources of the groundwaters in the NPCSX, as well as the river waters and groundwatersinthe Dali Basin,were mainly derived from the Daxin'Anling Mountains, by leaking of the Xilamulan River water through a thick aquifer in the eastern margins of the Otindag. By contrast, While the groundwaters in the

SPCSXweremainly recharged from two sources, . One was the flash floodsderived from the Yinshan Mountains and the river waters other was the Xilamulun River waters derived from the Daxin'Anlin Mountains. It indicatesthat the modern indirect recharge mechanism, instead of the direct recharge and the palaeo-water recharge,isthe most significant for groundnwaterrecharge in the eastern Otindag. This suggests thatthetectonic settings at a regional scale, but not the climate,is at the origin of was responsible forthe groundwater origin in the Otindag. This study providesd a new perspectivesight into the origin and evolution of groundwater resourcesin the middle-latitude desert zone of NH.The Otindag Desert is essential to livestock-economy and ecoenvironment of northern China. Although surface water is the traditional source for China's socio-economy in arid areas, the groundwater resources underlying the desert are increasingly burdened by groundwater pumping, which increases interest in the status of the groundwater resources. Widespread fresh groundwater deep to 60 m was found at the eastern part of the Otindag Desert. The occurrence of this massive fresh groundwater raises doubts on the often-made assumption in the literature that regional atmospheric precipitation or palaeowater, namely the direct recharge, is the source of water in the middle-latitude desert aquifers of northern China and makes further investigation necessary. Knowledge on the origin and recharge of this fresh groundwater is key in assessing the possibility of groundwater exploitation and utilization. In this study we conducted hydrogeochemical and isotopical analyses to assess possible origin and recharge of these groundwaters. It is concluded that the fresh groundwater can neither originate from regional atmospheric precipitation derived from the Asian Summer Monsoon system, nor from palaeowater that formed during the last glacial period. Our results indicate that with groundwater dating it is possible to originate from remote mountain areas via the faults of the Solonker Suture zone, including the Daxing'Anlin and Yinshan Mountains. Furthermore, it is deduced that the hydrological connection between desert aquifers and mountain systems through the suturezone is crucial to the hydrogeological functioning of the Otindag aquifer.Thissuggests that the modern indirect recharge mechanism, instead of the direct recharge and the palaeo-water recharge,is the most significant for groundwaterrecharge in the Otindag Desert. This study provides a new perspective into the origin and evolution of groundwater resourcesin the middle-latitude desert zone of Asian continent.

**Keywords:** fresh groundwater recharge origin; atmospheric precipitation; direct recharge; indirect recharge; palaeowater recharge; fault hydrology; middle-latitude desert; direct and indirect recharge; stable and radioactive isotope; ion hydrochemistry;climate control; tectonic control;Otindag Desert.

**1. Introduction**

Water Resources. In a semi-arid to arid region where rainfall is insufficient to supply the needs of a growing population and a higher standard of living, the deficit is normally made up by extracting groundwater. [Alsharhan, 2001, Hydrogeology of an Arid Region The Arabian Gulf and Adjoining Areas] As rainfall events are infrequent in arid and semi-arid regions of the world, surface runoff andrelated water resources are globally scarce and ephemeral.These areas thus rely heavily on groundwater asthe primary water resource to support local ecosystems(Herezeg and Leaney, 2011; Scanlon et al., 2006).It has been widely proved that the origin, quality and quantity of groundwater in

77 arid lands can be deeply influenced by environmental factors/processes,whichcontrollingthe

78 groundwater recharge and evolution, such as in the arid lands of northwestern China and Central Asia

79 (Zhu et al., 2015, 2016, 2017).For this reasonthese factors/processes become an essential component in

80 the understanding of regional hydrological systems and the management of water resources

81 (Dogramaci et al., 2012). For example, groundwater recharged by modern precipitation can refill

82 quickly but is vulnerable to contamination by the surface wastes, inversely, groundwater containing

83 mostly ancient water may not recharge to a useful extent over human timescalesand cannot be affected

84 by surface waters(Bethke and Johnson, 2008). Therefore, different strategies on groundwater

85 resourcesmanagement should be adopted when the different recharge mechanisms of groundwater

86 occurring.

87 In general, groundwater recharge can be broadly classified into two ways,the direct

88 recharge,namelydiffuse recharge by native water resources, and the indirect recharge,namely focus

89 recharge by external water resources. The direct recharge is replenished by precipitation infiltration

90 through the unsaturated zone and the indirect recharge is defined as recharge from mappable features

91 such as rivers, canals, and lakes originated from remote areas(Healy, 2010). It is well known that

92 groundwater recharge can be influenced by environmental factors, including climate change,

93 underlying soil and geology, land cover and population growth,over withdrawal and economic

94 development (Zhu et al., 2015, 2017), thus the amount of groundwater in arid and semi-arid regions

95 decrease rapidly while human demands on the limited water resources increase rather than decrease

96 (Ma et al., 2013). Between environment and groundwater recharge, climate and land cover largely

97 determine precipitation and evapotranspiration, whereas the underlying soil and geology dictate

98 whether a water surplus (precipitation minus evapotranspiration) can be transmitted and stored in the

99 subsurface (Giordano, 2009; Doll, 2009). Modelled estimates of diffuse recharge globally (Doll and

100 Fiedler, 2008; Wada et al., 2010) range from 13,000 to 15,000 km$^3$/yr, equivalent to ~30% of the

101 world's renewable freshwater resources (Doll, 2009) or a mean per capita groundwater recharge of

102 2100 to 2500 m$^3$/yr. These estimates represent potential recharge fluxes as they are based on a water

103 surplus rather than measured contributions to aquifers. Furthermore, these modelled global recharge

104 fluxes do not include focused recharge, which, in semi-arid and arid environments, can be substantial

105 (Scanlon et al., 2006; Favreau et al., 2009). For keeping sustainable management of water resources, it

106 requires urgently tounderstand both diffuse and focused recharge and meet both human and ecosystem

107 needs in arid areas of the world, particularly in Central Asia and Northern China.

108 Many areas iIn the middle-latitude desert zone of northern China such as, many areas of these

109 lands the Badanjilin Desert, the Mu US sandy Land and the Hobq Desert (Chen et al., 2012a; Chen et

110 al., 2012b), are unexpectedly rich in incommensuratewith large groundwater resources , such as the

111 Badanjilin Desert, the Mu Us Sandy Land and the Hobq Desert (Chen et al., 2012a; Chen et al., 2012b),

112 although they have been under arid or hyper-arid climate for a long time (Sun et al., 2010). How these

113 groudndwaters are originated and how they are recharged in these deserts are thus fundamental

114 scientific becaming a keyquestions. Until now, however, no consensus has been achieved ithas long

115 been altereated in the acadeamic circles.

116 For some of the earth scientists, the direct recharge is thought to be very important for

groundwaters in the wide desertlands of northwestern China due to lack of surface runoffs(Yang et al., 2010; Yang and Williams, 2003; Zhao et al., 2017). They argued that although the amount of atmospheric precipitation is small, the vast catchment area in the desert region couldconcentrate the rainfall into large inland basins, creating an aquifer with large storage capacity and great thickness.However, some of hydrologists suggested that the estimate of direct recharge usedby the chloride mass balancemethodwas 1.4 mm/year, approximately only 1.7% of the mean annual precipitation in a cold large desert (Badanjilin) in northern China (Gates et al., 2008). A similar estimation wasonly 1 mm/year for Gobi deserts from the Hexi Corridor to the Inner Mongolia Plateau in northwestern China(Ma et al., 2008). Consequently, they thought that heavy potentialevaporation and little precipitation make it difficult for direct recharge to meet the supply of groundwater in these desert areas.Thus, the indirect recharge isconsidered to be an important mechanism for groundwater recharge in these desert areas. For example, based on isotopic compositions of natural waters,Zhao et al. (2012)suggested that little precipitation had rechargedinto groundwaters in the Badain Jaran Desert.Chen et al. (2004)argued that the groundwatersin the Badanjilin Desert were recharged by palaeo-glacial melt water through faults and deep carbonate layers far away from the local desert.Many studies also suggested that palaeowaters stored in aquifer during wetter climate periods could recharge to groundwater under certain conditions in arid lands (Edmunds et al., 2006; Ma and Edmunds, 2006).Other kinds of indirect recharge, such as mountain front recharge from adjacent mountain blocks,are also proposed to offer an important inflow to aquifers within arid to semiarid catchments(Blasch and Bryson, 2007).

The Otindag Desert is one of the largest sandydesert lands located at inthe monsoon margin of northern China and is the geographical centre of the northeastern Asian Continent (Fig. 1), which can be regarded as a significant repository of information relating to the groundwater recharge in the arid Inner Asia. At present, the eastern Otindag is also a typical case for its unexpected incommensurate groundwater resources, because t.There is abundant of groundwater in this desert land and even rivers originate there due to the spillover of spring water, such as the tributaries of Xilamulun River in its north and the Shandian River in its south (Fig. 1). Climatically, the monsoon margin of northern China refers to a strip along the present East Asian Summer Monsoon (EASM) limits and is considered to be sensitive to climate change (Wang and Feng, 2013). Geologically, the Otindag Desert lies in a tectonic depression of the central Solonker suture zone with a few faults stretching east and west (Fig. 2), with its northern margin along a fault marked by a series of lake basins. Thus, the large-scale hydrogeological conditions of the Otindag Desert belong to a fault zone under the influence of the EASM climate.

Until now, hHowever, whether the climate or other factorsthe tectonic faults affected the origin of groundwater recharge andthefluid flow patterns in groundwateraquifersin the Otindag are still not known. Because at present Until now, however, lLittle data and documents about the groundwater and its origin is available in the literature inOtindag, and knowledge and reliable data on various hydrogeological characteristics of the desert such as the catchment extent, input/output, the hysteretic hydraulic functions, the transient hydraulic conditions, in-homogeneities, and on transfer functions to

overcome scale problems are also missing. Under such conditions, conventional methods such as water balance and hydraulic methods sometimes fail in determining groundwater recharge, particularly in extreme environments (arid, semi-arid, or cold) (Drever, 1997). Because pristine aquatic conditions may significantly differ from managed conditions in arid environment, and thus groundwater recharge is not a fixed number, but may vary with the boundary conditions of the recharge system (Seiler and Gat, 2007).

, can be obtained in literature. Whether the direct or indirect recharge is the major mechanism for groundwater recharge in Otindag In general, Groundwater recharge can be broadly classified into two categories: the direct recharge by native water resources and the indirect recharge by external water resources (Herczeg and Leaney, 2011). Water infiltration of atmospheric precipitation through the unsaturated zone to the groundwater is hydrologically defined as the direct recharge, and t. The indirect recharge is defined as recharge from mappable features such as rivers, canals, lakes and originates from remote areas (Scanlon et al., 2006; Healy, 2010)(Healy, 2010). It is well known that groundwater recharge can be influenced by environmental factors, including climate change, underlying soil and geology, land cover and the growth in human population that affects withdrawal and economic development (Zhu et al., 2015, 2017). Among these environmental factors, climate and land cover largely determine precipitation and evapotranspiration, whereas the underlying soil and geology dictate whether a water surplus (precipitation minus evapotranspiration) can be transmitted and stored in the subsurface (Giordano, 2009; Doll, 2008, 2009; Giordano, 2009).

For some earth scientists, the direct recharge is thought to be very important for groundwaters in the wide desert lands of north China due to the lack of surface runoffs (Yang et al., 2010; Yang and Williams, 2003; Zhao et al., 2017)(Yang et al., 2010; Yang and Williams, 2003; Zhao et al., 2017). They argued that although the amount of atmospheric precipitation is small, the vast catchment area in the desert region could concentrate the rainfall into large inland basins, creating an aquifer with large storage capacity and great thickness. However, some hydrologists estimated by the chloride mass balance method that the direct recharge was 1.4 mm/year, which represents approximately only 1.7% of the mean annual precipitation in a cold large desert (Badanjilin) in northern China (Gates et al., 2008)(Gates et al., 2008). A similar estimation of 1 mm/year was given for Gobi deserts from the Hexi Corridor to the Inner Mongolia Plateau in northwestern China (Ma et al., 2008)(Ma et al., 2008). Consequently, they thought that heavy potential evaporation and little precipitation make it difficult for direct recharge to meet the supply of groundwater in these desert areas. Thus, the indirect recharge is considered to be an important mechanism for groundwater recharge in these desert areas. For example, based on isotopic compositions of natural waters, Zhao et al. (2012) suggested that little precipitation had recharged into groundwaters in the Badain Jaran Desert. Chen et al. (2004) argued that the groundwaters in the Badanjilin Desert were recharged by palaeo-glacial melt water through faults and deep carbonate layers far away from the local desert. Many studies also suggested that palaeowaters stored in an aquifer during wetter climate periods could recharge to groundwater under certain conditions in arid lands (Edmunds et al., 2006; Ma and Edmunds, 2006). Other kinds of indirect recharge, such as mountain front recharge from adjacent mountain blocks, are also proposed to offer an important inflow to aquifers within arid to semiarid catchments (Blasch and Bryson, 2007)(Blasch and

197 Bryson, 2007).

198 In this paper, we focus to answer the a question that whether groundwater recharge in Otindag is
199 mainly direct or indirect, using hydrochemical and isotopic indicators as tracers to offer a valuable
200 support for identifying the contributions of preicipitation recharge on groundwater, since these
201 indicators reflect the composition of water molecules and are sensitive to physical processes such as
202 mixing and evaporation (Lawrence et al., 1976;Coplen, 1993; Sultan et al., 2000; Guendouz et al., 2003;
203 Petrides et al., 2006; Scanlon et al., 2006; Zhu et al., 2007, 2008; Jobbágy et al., 2011; Zhai et al., 2013;
204 Eissa et al., 2014). T, as the abovementioned hot question for other deserts in China,is alsounknown.

205 It should be kept in mind that virgin aquatic conditions may significantly differ from managed
206 conditions in arid environment, because groundwater recharge is not a fixed number, but may vary with
207 the boundary conditions of the recharge system (Seiler and Gat, 2007). Conventional methods such as
208 water balance and hydraulic methods sometimes fail in determining groundwater recharge in extreme
209 environments (arid, semi-arid, or cold) (Drever, 1997), because of missing knowledge and the lack of
210 reliable data on various characteristics such as the catchment extent, input/output, the hysteretic
211 hydraulic functions, the transient hydraulic conditions, in homogeneities, and on transfer functions to
212 overcome scale problems (Seiler and Gat, 2007). Under such conditions, tracer methods offer a
213 valuable support for natural water studies.

214 Geochemical elements and environmental isotopes have been widely used as effective tracers to
215 determine the sources of groundwater recharge, which could be attributed to infiltration by rainfall,
216 surface waters or both of them(Zhu et al., 2007, 2008; Zhu et al., 2017). For example, by comparing the
217 composition ofstable isotopics of hydrogen and oxygen in local meteoric waterswith these in
218 groundwaters, many studies successfully applied in identifyingwhether the rainfall play a vital role in
219 recharging groundwater or not(Zhu et al., 2007; Petrides et al., 2006; Jobbágy et al., 2011; Zhai et al.,
220 2013). Also, investigating the spatial distribution of groundwater age represented by the concentration
221 of tritium or radioactive carbon ($^{14}C$) can provide a way to understand the recharge relationship
222 between the modern rainfall and the groundwater(Sultan et al., 2000; Zhu et al., 2008). For the indirect
223 recharge, the groundwater flow regimes or its movement pathway deduced from hydrochemical and
224 isotopical tracers can indicate its origin and recharge processes. For example, the groundwater
225 mineralisation will increaseas a result of dissolution of evaporite minerals along flow lines that begin
226 with the recharge area (Guendouz et al., 2003). While, the geochemical and isotopic composition of
227 groundwaterswill be much complexat interface zones between groundwaters with different
228 hydrochemistry or ages, they will show distinct physiochemical characteristics indicating how they
229 mixed(Lawrence et al., 1976; Eissa et al., 2014).

230 he detailed The objectives of this study are: (1) to examine the distribution patterns of
231 environmental signals in the stable and radioactive isotopes and the major ionic hydrochemistry of
232 groundwater in the eastern Otindag drainage system, and (2) to recognize the major sources of
233 groundwater in the area, and (32) to identify the key mechanism of groundwater recharge in the desert.
234 land, particularly to discriminate whether the direct recharge or the indirect recharge being the major
235 control on groundwater rechage in the desert land.

236

237 **2.Regional settings**

238 Geographic setting. The Otindag Desert lies between latitudes 42° and 44°N and longitudes 112°

239 and 118° E (Fig. 1). It forms a part of the great middle-latitude desert belt in northern China which

240 stretches from the Taklamakan Desert of northwestern China to the Kelqin Desert of northeastern

241 China, near the west coast of the Pacific Ocean. The desert has an area of approximately

242  21,400 square kilometers , a middle-latitude  located the eastern

243  Inner Mongolia and  the monsoon margin of northern China

244  (Fig. 1). It ss in

245 China (Yang et al., 2012) and is bordered by a flat steppe terrain of Dali Basin to the

246 north, the Yinshan Mountains  and mountainous loess landscape to the south, and  the Greater

247 Khingan (Daxing'Anling) Mountains  to the east (Fig. 1). The Otindag Desert is essential to

248 livestock economy and ecoenvironment of northern China. Settlements in this desert are restricted to

249 areas to permanent springs, shallow groundwater and oases to areas where irrigation is possible. Some

250 nomads continue to eke out a precarious existence grazing livestock in the desert.

251 Topography and geomorphology. The Otindag Desert has a varied relief, combining extensive

252 dune fields with rugged mountains along the eastern, southern and southeastern rims. In the east, the

253 Daxing'Anling Mountains stretch from the Heilong River Valley into the upper reach valleys of the

254 Xilumulun River from northeast to southwest, gradually increasing in height northwards from about

255 180 m near Huma to Huanggangliang, where the highest peaks reach 2,029 m with an average

256 elevation range from 1,100 to 1,400 m. In the south and southeast, the Yinshan Mountains decline

257 gradually near Duolun and Zhenglanqi, and in some areas leave wide alluvial plains. The terrain of the

258 Otindag Desert is less rough and elevations decrease from ca. 1300 m in the southeast to ca. 1000 m in

259 the northwest. Over the greater part of this desert the ground cover consists of fixed and semi-fixed

260 sandy dunes, with a few mobile dunes in area of little vegetation. The dominated dune types are

261 represented from parabolic to barchans, linear and grid-formed types, ranging from a few meters to

262 over 40 m in height (Zhu et al., 1980; Yang et al., 2008).

263 Climate, vegetation and soil.

264 The climate of the Otindag Desert was not uniform in geological period, with much sand

265 movement, occasional rainy years, and several wetter intervals during the Holocene (Yang et al., 2015;

266 Tian et al., 2017). At present

267

268

269 ~~volcanic basalts forming flat terrains are to the north (Zhu et al., 1980;Li et al., 1995).__[Yang et al.,__

270 __2007, Catena]__

271

272 he whole desert belongs to the temperate  temperate zone

273 , with a mean temperature of 2 °C in the north and 4°Cin the south (Liu and Yang,

274 2013). At the regional scale, the climate of the desert is typically

275 controlled by the East Asian monsoon system, characterized by a warm summer, with precipitation

276 transported by the EASM, and by a cold and dry winter under the influence of the East Asian Winter

Monsoon (EAWM). The rainfall in the desert exhibits a wide variation in space and time. Iwhose influence of the EASM is changesing from southeast to northwest in the desert, and varies with latitude and distance from the Pacific Ocean, leading to the mean annual rainfall decreasing from ~450 mm in the southeast to ~150 mm in the northwest (Yang et al., 2013)(Yang et al., 2013). This uneven distribution of precipitation has a major influence on the availability of near-surface moisture, consequently on the distribution of vegetation, soil and the animal husbandry potential of local communities. The basic soil cover consists of grey desert soil in the west and changes to sierozems and chernozem or chestnut soil in the east. Through the desert, vegetation is sparse in the west and relatively abundant in the east. The natural vegetation is characteristic of desert or semi-deserts, with scrub woodland in the east and steppe in the west. Due to the scarcity of surface water, the growing season is affected by temperature, rainfall and elevation, and hence cultivation is restricted mainly to flood plains.Fixed and semi-fixed sandy dunes are dominate the landscaped in the desert land, with a few mobile dunes in area of little vegetation. Several dDune types are represented various from parabolic to barchans, linear and grid-formed types, ranging from a few meters to over 40 m in height (Yang et al., 2008; Zhu et al., 1980).

Geology. The Ontindag Desert is located in a tectonic depression of the Solonker Suture Zone (Jian et al., 2010) bounded by the Northern Early to Mid-Paleozoic Orogen Zone and the Hatug Uul Block to the north, the Southern Early to Mid-Paleozoic Orogen Zone and the North China Craton system to the south (Fig. 2). A few faults such as the Xar Moron Fault and Chifeng-Bayan Obo Fault stretch east and west, with its northern margin along the Solonker Suture Zone marked by a series of lake basins (Figs. 1 and 2). The tectonostratigraphic units and overall structural trends are mainly oriented NE–SW (Fig. 2), which may be interpreted as resulting from overall compressive stresses oriented principally in the NW–SE quadrants during orogenesis (Jian et al., 2010; Zhang et al., 2015). Diverse rock types from unlithified and lithified clastic sediments through to carbonate, crystalline, and volcanic rocks are distributed in and around the Otindag Desert (Zhang et al., 2015) (Figs. 2 and 3XX). [Bense et al., 2013, ESR]

Tertiary and Quaternary sandstones and mudstones are the common basement rocks under the dunes of the Otindag, and extensive volcanic basalts forming flat terrains are to the north (Zhu et al., 1980; Li et al., 1995).

Hydrology and hydrogeology.

The Otindag Desert originated during the Late Quaternary (Yang et al., 2015) and various alluvial fans formed at the margins of this desert during the early to middle Holocene. These are composed of conglomerate and sand deposits, where major periodic steams or wadis debouched into the Otindag. At present two rivers run through the eastern margin of the Otindag Desert, i.e. the Xilamulun River in the north and the Shandian River and its two tributaries, the Shepi River and Tuligen River in the south. Both stem from the eastern and southeastern parts of the Otindag (Fig. 1). The Xilamulun River, 380 km in length and $32.54\times10^3$ km$^2$ in area, is a neighboring river both to the northeastern Otindag and the southeastern Dali Basin, the northern catchment of the Otindag Desert. The Xilamulun River flows to the east and finally goes into the Xiliao River, with an annual mean runoff of $6.58\times10^8$ m$^3$ (Wu et al., 2014). The Shandian River is the upper reach of the Luan River,

with a length of 254 km and a catchment area of $4.11\times10^3 \text{km}^2$ (Yao et al., 2013). Along the low, flat and sandy shorelines of some lakes in the Otindag, salt flats or sabkhas have formed in shallow depressions. Due to the high rate of evaporation, salt crusts develop which have been locally exploited where the salt is relatively free from sand. During rainy season, some rain and floodwaters (generally coming from the Yinshan piedmonts) are retained in low-lying areas, which may temporarily recharge shallow aquifers. Under storm conditions, occasional heavy, short rainstorms cause floods in soil-rich wadi channels. Under other conditions, sand dunes and sand sheets bury the ground and sabkhas.

The Otindag Desert can depend on several water-bearing formations and units (aquifers) for their groundwater resources (Fig. 3). Coarse- to f

Fine-grained sedimentary rocks, magmatic rocks and metamorphic rocks of the Inner Mongolia-Daxing'Anling Orogenic Belt (Zhang et al., 2015) form the major regional aquifer unit (Fig. 3). They are composed mainly of alluvial sediments (mid-Permian Zhesi Formation), melange (Solonker suture zone), A-type granite (early Permian), bimodal volcanic rocks with sedimentary intercalations (early Permian Dashizhai Formation), diorite-quartz diorite-granodiorite rocks (Carboniferous-Permian) and metamorphic complex (predominantly gneiss, early Paleozoic) (Fig. 2). The aquifer is generally unconfined in dune fields of the Otindag Desert, unconfined to semi-confined in the Yinshan Mountains' piedmont, and semi-confined to confined in the Daxing'Anling uplands (Fig. 3). Water-level measurement in June 2010 indicated that the general depth of unconfined groundwater level ranges between 10 to 70 m in the Otindag Desert (Fig. 3). Local granular aquifers in the central desert are composed of coarse fluvial, lacustrine and aeolian sediments, but their extent and thickness vary throughout the watershed (Zhu et al., 1980; Li et al., 1995). The generally coarse-grained texture of the unconsolidated rock formations provides primary porosity in terms of groundwater flow in the desert.

~~Most of the tectonic fabric of the Appalachians was generated by compression or low angle thrusting; in those areas where major faults are strike-slip in nature, deformation is largely limited to rocks adjacent to the faults. The tectonostratigraphic units and overall structural trends are mainly oriented NE-SW, which may be interpreted as resulting from overall compressive stresses oriented principally in the NW-SE quadrants during orogenesis (Faure et al. 2004, 2006). The generally fine-grained texture of the rock formations provides negligible primary porosity in terms of~~

groundwater flow (Benoît et al. 2008).[Benoît et al., 2014, CWRJ]

Deformation along faults in the shallow crust (<1 km) introduces permeability heterogeneity anisotropy, which has an important impact on processes such as regional groundwater flow. Fault zones have the capacity to be hydraulic conduits connecting shallowand deep geological environments, but simultaneously the fault cores of many faults often form effective barriers to flow.The direct evaluation of the impact of faults to fluid flow patterns remains a challenge and requires a multidisciplinaryresearch effort of structural geologists and hydrogeologists.[Bense et al., 2013, ESR]

The Otindag Desert depend on several water bearing formations and units (aquifers) for their groundwater resources. Theyare composed mainly of sandstone and limestone. [Alsharhan, 2001, Hydrogeology of an Arid Region The Arabian Gulf and Adjoining Areas]

During rainy season, some rain and flood waters are retained behind dune dams and recharge shallow aquifers.

Two rivers inrun through the Otindag, i.e. the Xilamulun River in the north and the ShandianRiverand its withtwo tributaries of theShepi River and the Tuligen River in the south. B,bothstem from the eastern and southeastern parts of the Otindag(Fig. 1). The Xilamulun River flows to the east and finally goes into the Xiliao River, with a catchment area of $32.54 \times 10^3$ km$^2$ and an annual mean runoff of $6.58 \times 10^8$ m$^3$(Wu et al., 2014).The Shandian River is the upper reach of the Luan River, with a length of 254 km and a catchment area of $4.11 \times 10^3$ km$^2$ (Yao et al., 2013).

**3.Methods**

The hydrochemistry of natural water in the Otindag Desert, as related to the prevailing EASM climate, as well as, the dominant topographical, geological (tectonic) and hydrogeological conditions, are discussed here and interpreted, using chemical and isotope analyses of water samples from rain, springs, shallow aquifers and deep aquifers, rivers and lakes, and are represented on relevant graphs and diagrams. Fieldworks took place during the summer season of 2011 and the spring season of 2012. The water samples selected in this study were all collected from natural water, including the groundwater, river water, lake water, spring water and precipitation water in types.A tTotal of twenty fivewater samples were analyzed collected for ion chemical,stable and radioactive isotopic analyseisforin this study.Water Groundwatersamples is the major type among these waters, whichwere mainly retrieved taken from shallow and deep wells widelylocated over a wide area in dune fields of the study regionsarea. The detailed locations of the sampling sites are shown in Fig. 43.

Two groups of parameters are measured to characterize the chemistry of any water analysis: field-measured parameters and lab-measured parameters. The filed-measured parameters include temperature (°C), hydrogen-ion concentration (pH), electrical conductivity (EC in micro-Siemens per centimeter or μS/cm) and total dissolved solid (TDS, mg/L). The values of these parameters change when they are not directly measured in the field. The number lab-measured parameters depend on the purpose of study. However, the measurement of major cations (F$^-$, Cl$^-$, NO$_2^-$, NO$_3^-$, SO$_4^{2-}$HCO$_3^-$, CO$_3^{2-}$and H$_2$PO$_4^-$) and anions (Li$^+$, Na$^+$, NH$_4^+$, K$^+$, Mg$^{2+}$and Ca$^{2+}$) are determined in most chemical analyses. Analysis for stable ($^2$H and $^{18}$O) and radioactive isotopes ($^3$H) in rain and groundwater are

also included. The analytical data of the physiochemical parameters and the stable and radioactive isotopes of the water samples collected in this study are listed in Tables 1, 2 and 3, respectively.

Thesurface waters were mainly sampledfrom rivers and lakes in the Otindag, and the spring waters were collected from the riverhead of the Xilamulun River,the Shepi River and the Tuligen River. One rainfallsample of the local atmospheric precipitation (p1)was also collected at the southeastern margin of the Otindag in the 2011 summer season.Water samples were filtered using 0.45μm membrane filters for cation and anion analysis, and were acidified with 1% $HNO_3$ for cation analysis.Water samples for stable and radioactive isotope analysis were collected in the field with a polyethylene bottles of 0.5L in volume, respectively. Variables Some kinds of analysis were measured on site with a portable instrument (Eijkelkamp). These determinations included temperature, pH, oxidation-reduction potential (Eh), electrical conductivity (EC), and total dissolved solid (TDS).The measurement errors barswere<±0.1 °C for temperature, <±1% for pH, <±5% for Eh, <±5% for EC, and <±0.5% for TDS, respectively.

The concentrations of major anions ($F^-$, $Cl^-$, $NO_2^-$, $NO_3^-$, $SO_4^{2-}$ and $H_2PO_4^-$) and cations($Li^+$, $Na^+$, $NH_4^+$, $K^+$, $Mg^{2+}$and $Ca^{2+}$) were determined by electrochemical detectors of an ion chromatography (Dionex 600) in the Institute of Geology and Geophysics, Chinese Academy of Sciences, with measurement errorsbars<±3% for anions and <±2% for cations. The concentrations of carbonate (alkaline) ions of $HCO_3^-$ and $CO_3^{2-}$were measured by titration with HCl (0.1 M)following theaGran Method (Gran, 1952), with an error bar<±5%. The hardness (HD, German standards) of these water samples wascalculated based on the equation HD = ([$Mg^{2+}$]×100/24.305+[$Ca^{2+}$]×100/40.08)/17.847, [$Mg^{2+}$] and [$Ca^{2+}$] referring to the concentration of $Mg^{2+}$and $Ca^{2+}$with unit of mg/L.

Two stable isotopes of $^2H$ and $^{18}O$,as being expressed in δ-notation ($\delta^2H$=$^2H/^1H$, $\delta^{18}O$=$^{18}O/^{16}O$) relative to the Vienna standard mean water (VSMOW), were measured for all of the water samples collected in this study,usingby MAT 252 in the Laboratory for Stable Isotope Geochemistry, Institute of Geology and Geophysics, Chinese Academy of Sciences, with σ< ±0.374‰ for $\delta^2H$ and <±0.062‰ for $\delta^2H$ and $\delta^{18}O$, respectively.

Several groundwater samples (500 ml each), collected from wells (6-60 m deep) in the study area, were prepared for the analysis of radioactive isotope (tritium) analysis. 300 ml of water sample, added with addition of 1 g $KMnO_4$, were distilled to remove any impurities. In order to increase the tritium concentration to an easily measurable level, electrolytic enrichment was applied (Kaufman, 1954; Baeza et al., 1999). A volume of 250 ml of previously distilled sample with 2.5 g NaOH was then put to the electrolysis apparatus containing electrolytic cells with co-axial stainless steel electrodes. Electrolysis was carried out until the volume of electrolyte was reduced to 8 ml and all runs were performed at a temperature of 2-5 °C to prevent the loss of tritiated water molecules by evaporation. After electrolysis $CO_2$ was bubbled through the cell to neutralize the water because the medium in which the electrolysis took place earlier is alkaline. The water sample was separated from the electrolyte by distilling. The pretreated samples were measured by a low level background liquid scintillation counter (Quantulus 1220-003) according to the manufacturer's guidelines. The error bar of the measurement errors are should be<±3%. The tritium data of several groundwater samples collected in this study had been partially mentioned by Yang et al. (2015) as one of the supplementary materials.

436

437

438 **4.Results and Discussions**

439

440
441
442

443

444 **4.1.Hydrochemical characteristicsstry of natural  waters **

445
446
447
448
449  The natural water samples collected in this study are generally neutral to slightly
450 alkaline, with the pH values varying between 6.26 and 9.44 (except the precipitation sample p1, 4.61)
451 (Table 1) and a median value of 7.27. The TDS values range between 67 and 660 mg/L (average 211
452 mg/L) (Table 1), all belonging to fresh water(TDS < 1000 mg/L) in the salination classification of
453 natural water (Meybeck, 2004).

454 The variations in ion concentrations of the major cations and anions in the studied water samples
455 were displayed in  a fingerprint diagram with a semi-logarithm
456 y-axis (Fig. 35). The rain water sample is the most depleted in ions among these samples. T
457 The groundwater samples have the highest concentrations of cations and anions
458 and the lake, river and spring waters had
459 intermediate  values. The calcium concentration is the highest among in in
460 almost all of the water samples, and the $HCO_3+CO_3$ concentration (bicarbonate + carbonate, alkalinity)
461 is the highest among in in most of the water samples. For several groundwater
462 samples (g3, g4, g5, g6 and g11), spring sample (s1) and  precipitation sample (p1),
463 they have  higher $SO_4$ concentrations than alkalinity (Fig. 53).

464 Two chemically distinct water types are recognized for the studied waters in a Piper diagram
465 (Fig. 6), calcium bicarbonate and calcium sulphate .
466 revealed
467
468 , type I, the $Ca-HCO_3$
469 affected by surface mineral weathering,
470 dominated by alkaline earth metals (Zhu et al., 2011, 2012; Clark, 2015).
471
472
473
474 No  Chloride-type and
475 sodium-type waters occur in the study area (Fig. 64). Based on more than 10,000 chemical

analyses of groundwater samples from the world, Chebotarev (1955) observed that the global groundwater tends to evolve chemically towards the composition of seawater. He also observed that this evolution is associated with regional changes in dominant anions but not cations, as the concentration of cations may exhibit a wide range of fluctuations in groundwater and is not as steady as the changes in anion dominance. Freeze and Cherry (1979) illustrated the Chebotarev's (1955) general evolution of groundwater as a anion evolution line: $HCO_3^- \rightarrow HCO_3^- + SO_4^{2-} \rightarrow SO_4^{2-} + HCO_3^- \rightarrow SO_4^{2-} + Cl^- \rightarrow Cl^- + SO_4^{2-} \rightarrow Cl^-$, which travels along the flow paths and increasing ages. On this evolution line, bicarbonate water is generally characteristic of low salinity, renewable water resources and low residence time, while sulphate waters predominate in groundwater passing through gypsum and anhydrite aquifers, and is usually associated with intermediate salinity in unconfined aquifers (Clark, 2015). The distribution pattern of water chemical types occurred in the studied area, indicatesing a primary stage of groundwaterwater evolution for natural watersin the Otindag Desert., in terms of the hydrogeochemical perspective.

The hydrochemical facies of the studied water samplescan be further illustrated by anDurov diagram (Durov, 1948) and its expanded models (Lioyd and Heathcote, 1985; Al Bassam et al., 1997; Chadha, 1999; Al Bassam and Khalil, 2012). All the groundwater and spring water samplesin this study fell into the Durov fields 1, 4 and 5 of the expanded Durov diagram (Fig. 5).The water samples in the Durov field 1were actually thesame than to those classified into the Piper water type 1 (Fig. 4), while samples in the Durov fields 4 and 5 were the same thantothose of the Piper water type II (Fig. 4). Based on the graphic decipherment of Lioyd and Heathcote (1985), water samples in field 1 represent the presence of $HCO_3^-$ and $Ca^{2+}$-dominant water type, while samples in field 4 indicate the $SO_4^{2-}$ dominant (or anions indiscriminate) and $Ca^{2+}$ dominant water type, and samples in field 5 represent the water type without any dominant anion or cation. All the groundwater and spring water samples in this study were distributed close to the line of simple dissolution or mixing process. However, almost all the river and lake water samples were located in the Durov field 2 and were close to the line of ion exchange process (Fig. 5). These distribution patternsindicated that the ground waters and the surface waters had experienced different geochemical processes in the formation and evolution of natural waters in the Otindag.

**4.2.The sStable and radioactive isotopic compositions esof natural waters in the Otindag**

If we plot the relationships between oxygen and hydrogen isotopes of groundwater, springer, river water and lake water, we find that they are distributed between two straight lines with a gradient of 4.4, but with different y intercepts (Fig. XX), as shown in the following equations:

$\delta D = 4.38\delta^{18}O - 24.97$ ($R^2=0.87$, n=11) for groundwater samples.

$\delta D = 4.44\delta^{18}O - 24.56$ ($R^2=0.86$, n=13) for groundwater and spring water samples.

$\delta D = 4.09\delta^{18}O - 28.31$ ($R^2=0.93$, n=24) for groundwater, springer water, river water and lake water samples.

$\delta D = 7.95\delta^{18}O + 10.52$ ($R^2=0.77$, n=5) for river water samples.

$\delta D = 2.69\delta^{18}O - 33.94$ ($R^2=0.92$, n=6) for lake water samples.

$\delta D = 6.57\delta^{18}O + 0.31$ ($R^2=0.88$, n=XX) for precipitation water in the Tianjin Station.

$\delta D = 6.36\ \delta^{18}O - 5.21\ (R^2=0.93, n=XX)$ for precipitation water in the Baotou Station.

$\delta D = 6.86\delta^{18}O - 2.23\ (R^2=0.91, n=XX)$ for precipitation water in spring in the Tianjin Station.

$\delta D = 6.68\delta^{18}O - 0.98\ (R^2=0.93, n=XX)$ for precipitation water in summer in the Tianjin Station.

$\delta D = 5.51\delta^{18}O - 4.13\ (R^2=0.55, n=XX)$ for precipitation water in autumn in the Tianjin Station.

$\delta D = 7.44\delta^{18}O + 13.57\ (R^2=0.94, n=XX)$ for precipitation water in winter the Tianjin Station.

$\delta D = 6.66\ \delta^{18}O + 0.30\ (R^2=0.93, n=XX)$ for precipitation water in spring in the Baotou Station.

$\delta D = 5.07\delta^{18}O - 15.1\ (R^2=0.80, n=XX)$ for precipitation water in summer in the Baotou Station.

$\delta D = 6.99\delta^{18}O + 0.85\ (R^2=0.95, n=XX)$ for precipitation water in autumn in the Baotou Station.

$\delta D = 6.86\delta^{18}O - 0.72\ (R^2=0.98, n=XX)$ for precipitation water in winter in the Baotou Station.

The stable isotopes of $\delta^2H$ and $\delta^{18}O$ were analyzed for all the water samples collected in this study, as shown in Table 3 and Fig. 6. The radioactive isotope of tritium ($^3H$) was analyzed for a part of the groundwater samples.

The $\delta D$ values of the groundwater samples collected in this study varied from -63.42‰ to -75.92‰ (Table 3), with an average -69.53‰. The $\delta^{18}O$ values ranged between -8.64‰ and -11.26‰ (Table 3), with an average -10.17‰.

The spring water samples which directly drain into rivers were relatively concentrated in values of $\delta^2H$ $\delta D$ and $\delta^{18}O$ and were greatly similar to those of the groundwater samples (Fig. 7 6). The $\delta^2H$ and $\delta^{18}O$ values in the spring samples varied from -70.83‰ to -72.60‰ (mean value -71.72‰)and from -10.34‰ to -10.47‰(mean value -10.40‰), respectively(Table 3).

The $\delta D$ and $\delta^{18}O$ values in the river water samples were slightly more variable ed and were also similar to those of the groundwater (Fig. 7 6), with a range of between -65.00‰ and -85.16‰(mean value -73.02‰) in $\delta^2H$ values and a range of between -9.55‰ and -11.78‰(mean value -10.51‰)in $\delta^{18}O$(Table 3).

The lake water samples in this study were enriched in $\delta^2H$ $\delta D$ and $\delta^{18}O$ by comparison ng to the groundwater samples (Fig. 6), with a variable range of between -34.16‰and -53.13‰(mean value -46.47‰)in $\delta^2H$ values and a range of between 0.38‰and -6.55‰(mean value -4.65‰)in $\delta^{18}O$(Table 3).

The precipitation sample p1 was also enriched in $\delta D$ and $\delta^{18}O$ by comparison to the groundwater samples (Fig. 7 6), showed the a $\delta^2H$ value of -47.4‰ and a the $\delta^{18}O$ value of -7.14‰, respectively(Table 3).The content of radioactive isotope of tritium ($^3H$) measured in seven well groundwater samples with 6-60 m depth ranged from 1.86 to 24.35 TU (Table 3), with an average 14.95 TU, higher than the mean tritium concentration (9.8 TU) of groundwater in the Vienna Basin, Austria (Stolp et al., 2010), the seat of the International Atomic Energy Agency (IAEA).

If we plot the relationships between oxygen and hydrogen isotopes of groundwater, spring er, river water and lake water samples, we observed find that the regression line that fits all data points can be

556 described by the equation:  $\delta D = 4.09\delta^{18}O - 28.31$ ($R^2$=0.93, n=24)
557 (EL1 in Fig. 7XX), .
558 This local groundwater line (LGWL) is different from the Global Meteoric Water Line (GMWL, $\delta D =$
559 $8\delta^{18}O + 10$) and the Mediterranean Meteoric Water Line (MMWL, $\delta D = 8\delta^{18}O + 20$) estimated by Craig
560 (1961), but it is similar to the local groundwater lines established for other deserts in northern China
561 and central Asia with a same slope but different Y-intercepts, such as $\delta D = 4.17\delta^{18}O - 31.3$ for the
562 Badanjilin Desert (Jin et al., 2018), $\delta D = 4.8\delta^{18}O - 15.2$ for the Ejina Desert in China (Wang et al.,
563 2013), and $\delta D = 4.26\delta^{18}O + 9.23$ for the Rub Al Khal Desert in the United Arab Emirates (Rizk and
564 El-Etr, 1997). The scatter of stable isotope data points for the lake water samples (Fig. 7) in the
565 Otindag suggests that the lake waters are affected by evaporation, but the other waters in the desert are
566 not so.

**45.21 . Evaluation of Ll Local precipitation recharge on as a recharge source of groundwater in the Otindag**

Comparison of the isotopic signals between the modern regional precipitation and natural waters in the Otindag

To incorporate the isotopic analysis of precipitation with similar areas in the studied area, local data (p1) was plotted with those of Baotou (Fig. 76). The isotopic composition of rainfall in Baotou, the nearest long-term station to the Otindag Desert, was monitored for the period 1986-2001 within the scope of the International Atomic Energy Agency/World Meteorological Organization (IAEA/WMO) global survey. The stable isotope data available from this station was used to provide basic characteristics of the stable isotopic composition of the present-day meteoric water, especially in the westward inland areas of the Otindag Desert (Fig. 1). Stable isotope data of the Tianjin station was also used to characterize precipitation of the eastern coastal areas of the Otindag Desert (Fig. 1).

At present, the extensive record of stable isotope measurements from in atmospheric precipitation are is still lacking from absent in the Otindag. T, thus in this study, we used the decadal isotope data of atmospheric precipitation around the Otindag were collected in this study to determine the isotopic relationship between the local groundwater and the regional precipitation that are available from. A global database, the IAEA Global Network of Isotopes in Precipitation (GNIP) database, is available to use in this study. Taking into account the boundary between the northern hemispheric westerly and the Asian summer monsoon (Chen et al., 2010), which are the two major climate systems controlling the Otindag(Yang et al., 2013), we chose two GNIP meteorological stations:as the representations of the atmospheric precipitation derived from the northern hemispheric westerly and the Asian summer monsoon, respectively. One is the Baotou station, located to the southwest of the Otindag as representative of (the westerly system), and another is the Tianjin station, located to the southeast of the Otindag, as representativeof (the Asian summer monsoon system) (Fig. 1a). The historical isotopic data ($^3$H, $\delta^2$H and $\delta^{18}$O, ‰VSMOW) over the last four decades from the two stations, as well as other data including the daily precipitation amount (mm) and air temperature (°C) in the same period, were taken as the references of the stable isotopic signals in precipitation in the Otindag.

The annual weighted mean values of $\delta^2$H and $\delta^{18}$O at the Baotou station varied were variable from -64.32‰ to -48.44‰ and from -9.40‰ to -6.50‰ during the period of 1986 to 1992, respectively. The annual weighted mean values of $\delta^2$H and $\delta^{18}$O at the Tianjin station varied from -56.30‰ to -43.72‰ and from -8.35‰ to -6.86‰ during the periodsof 1988 to 1992 and of 2000 to 2001, respectively. The long-term weighted mean values of $\delta^2$H and $\delta^{18}$O at the Baotou station (LWMB) were -55.27‰ and -7.78‰, respectively, and were -49.97‰ and -7.70‰ at the Tianjin station (LWMT), respectively. The radioactive isotope of $^3$H (TU) in precipitation was not stable at the GNIP Baotou station. The annual weighted mean values were higher than 30 TU in this station and tended to be decreased from 1986 to 1991 (72.06, 57.81, 59.97, 52.79, 55.89, 34.35 TU, respectively). The annual weighted mean values of $^3$H at the GNIP Tianjin station were lower than those of the Baotou station. The mean values were 21.99, 21.65, 18.55, 25.72, 18.80 TU from 1988 to 1992, and 7.01 and 15.48 TU from 2000 to 2001.

As theThe sample p1,the only one precipitation sample collected in this study (during the 2011 summer rainfall event) of the Otindag, the sample p1 fell onto the Global Meteoric Water Line (GMWL: $\delta^2H = 8\delta^{18}O + 10$) estimated by Craig (1961). It showed similar $\delta^2H$ and $\delta^{18}O$ values to those of the precipitation collected in the GNIP stations of Baotou and Tianjin (Fig. 6).

Compared to the precipitation data from the GNIP Baotou and Tianjin stations and from the local precipitation (p1) in the Otindag, the groundwater samples were evidently depleted in heavy stable isotopes in the OtindagHSKDSL(Fig. 6).

In contrast to the precipitation data, the water samples from springs and rivers in the study area also showed a depletion characteristics in the stable isotopes of $\delta^2H$ and $\delta^{18}O$(Fig. 6).

Based on the isotopic data from the Baotou station, tThe local regional meteoric water lines i.e. the regional Craig lines,can be statistically describedexpressed as the isotopic regression equation of $\delta\underline{D}^2H = 6.36\delta^{18}O - 5.21$ (line LMWL-B). It can also ,based on the isotopic data fromat the Baotou station,and can be destribedexpressed as $\delta\underline{D}^2H = 6.57\delta^{18}O + 0.31$ (line LWML-T), based on the data fromat the Tianjin station (Fig. 76). The precipitation sample p1 collected in this study fell onto the GMWL (Fig. 7). It also showed similar $\delta D$ and $\delta^{18}O$ values to those of the precipitation collected in the GNIP stations of Baotou and Tianjin (Fig. 76).

Compared to the precipitation data from the GNIP stations and from the local precipitation (p1), the groundwater, spring, and river water samples were evidently depleted in heavy stable isotopes in the Otindag (Fig. 76). Exceapt for the lake water samples, most of the groundwater, river water and spring water samples in the Otindag fallfell on or lay between the LMWL-B and the LMWL-T lines, and are werelocated at the lower left area of the precipitation points (Fig. 76). This indicates indicated that no strongdeep evaporation process was experienced by these ground and surface waters (except for lake waters) compared with than the precipitation.

For the Otindag evaporation line (EL1), its equation slope and intercept were significantly lower than that of the GMWL, LMWL-B and LMWL-T (Fig. 6). The points of intersection between the EL1 and LMWL-B were atwas -69.93‰ for $\delta^2H$ and -10.18‰for $\delta^2H$and for$\delta^{18}O$, respectively, while the intersection points between the EL1 and LMWL-T werewas -75.51‰ for $\delta^2H$ and -11.54‰ for$\delta^2H$ and$\delta^{18}O$, respectively.

5.2.The direct recharge of groundwater in the eastern Otindag

Water infiltration of atmospheric precipitation through the unsaturated zone to groundwater is hydrologically defined as the direct recahrge. The deuterium and oxygen isotopesare the composition of water molecules and are sensitive to physical processes such as mixing and evaporation, hence they are ideal tracers of the origin of groundwater(Coplen, 1993; Scanlon et al., 2006). Weusedthem to identify the contribution of preicipitation recharge on groundwater in this study.

Because the annual mean precipitation amount in the semi-arid regions of northern China is between 200- 400 mm, it seems that the direct recharge on groundwater cannot be neglected in the eastern Otindag under a semi-arid climate. However, when we checked the stable isotopic data fromthe GNIP stations both at the Baotou and Tianjin, we observed that almostall the annualweighted mean values of the stable isotope contentsinprecipitationwere enriched in$\delta^2H$and $\delta^{18}O$ compared with than

those values measured for the groundwater, spring water and river watersamples in this study (Fig. 6).Because the isotopic evolution of δD$^2$H and δ$^{18}$O in water illustrated in the Craig line represents a one-way and irreversible process, thus the water bodies distributed at the upper right area of the Craig line can not be recharge sources for the water bodies distributed at the lower left area of the line. Such results indicated that the groundwater, river water and spring water in the Otindag arewere not recharged by the regional precipitation, namely no significant modern direct recharge has taken place for groundwater in the Otindag.

Dogramaci et al. (2012) documented that only theintense and remarkable rainfall events of>20 mm could remarkably recharge groundwater in the semi-arid Hamersley Basin of northwest Australia, while the rainfall events <20 mm had limited influences on groundwater recharge. Chen et al. (2014) described that rainfall events ≤5 mm in the arid and semi-arid region of northern China would be evaporated into the atmosphere rapidly before it is infiltrated into the groundwater system. Based on the analysis on the data records from two meteorological stations around the Otindag, i.e.,the Duolun stationand the Xilinhaote stations (see Fig. 1a), we observed that the average times of rainfall events being >20 mm on averagein amount were only occur 2.5-3.4 times per year (Table 4). In some years (e.g. from 2005 to 2007 at the Xilinhaote Station), no Even none of the rainfall events of>20 mm even occurredduring the year from2005 to 2007 at the XilinhaoteStation. It further indicated confirmed that the small amounts of intensive rainfall eventshadlimited the contribution of regional precipitation on groundwater recharge in the Otindag.

In addition to groundwater, the river water and spring water samples fromin the the Otindag hadthesimilar isotopic signals with those of groudndwaters, and were also deviated from the local modern regional precipitation in the Craig diagram (Fig. 76).These water samples came from the Xilamulun, Shepi and Tuligen rivers. They shared the same evaporation line (EL1) with the groundwater and lake water samples (Fig. 76). Generally speaking, natural waters that have a same recharge source are can bedistributed on a same line of evaporation in the δ$^2$ and δ$^{18}$O diagram (Chen et al., 2012b)(Chen et al., 2012b) This indicates dthat the recharge sources of groundwater, river water, spring water and lake water in the Otindag are weregenetically associated each other and werediffer from ential to the localregional precipitation. During the fieldinvestigation, we observed that the elevation of spring outflow waslower thanthat of the groundwater table in some areas. Thisimplies yed that the spring water can be originateds fromthe local phreatic water (groundwater). The same isotopic signals between the two kinds of water confirmed their close relationship in origin.

**45.33. Winter precipitation and palaeowater recharge on groundwater in the Otindag Potential sources of groundwater other than summer precipitation in the Otindag: three hypotheses**

Since the groundwater samples in the Otindag arewere depleted in their δD $^2$Hand δ$^{18}$O values even more than those of the modernlocal rainfall (Fig. 76), they must be sourced from other waters characterized by similar with same or more depleted signals in their stable isotopes compositions. Due to the temperature effect (such as evaporation) on isotopic fractionation, only the waters issued from colder environments can be more depleted in their δD and δ$^{18}$O values even more than those of the

716 local rainfall.

717 Because the Otindag Desert is under the control of the EASM climate (Fig. 1), the local rainfall

718 in the desert is maily sourced from summer precipitation. This can also be illustrated by the seasonal

719 distributions in annual mean precipitation (Fig.8a), in annual mean air temperature (Fig. 8b) and in

720 annual mean water vapor pressure (Fig. 8c) over the last forty years at the two surrounding GNIP

721 weather stations in Baotou and Tianjin.

722 Because the Otindag Desert is under the control of the East Asian Summer Monsoon

723  modern rainfall in the desert is mainly sourced from the

724

725

726

727 The seanonal

728 distributions of stable isotopes in the two stations (Fig. 8d-e) show that the

729 summer rainfall is  evidently  positive in its signals of δD$^2$ and δ$^{18}$O by

730 comparison with those of the winter rainfall, further suggesting that the waters issued from cold

731 environments can be more depleted in their δD and δ$^{18}$O values than those of the summer rainfall

732

733 Thus we  speculated that  groundwater in the Otindag can be

734 potentially derived from (1)  modern

735 precipitation in winter, (2)  palaeowater formed in the past glacial period, or (3) remote/

736 mountains waters that emanate in  colder and wetter conditions.

737

738  The annual mean

739 values of δD$^2$ and δ$^{18}$O over the last forty years are more depleted in  winter precipitation

740 than in  summer precipitation at the Baotou and Tianjin stations (Fig. 8d-e). This isotopic signal

741 qualifies the regional winter precipitation to be  a potential souce of

742 groundwaters in the Otindag. However,

743

744 the precipitation amounts and the water vapor pressures (effective moisture) in

745 winter months are much lower than those in the summer months at both the Baotou and Tianjin

746 stations (Fig. 8a and 8c). It indicates that the winter seasons in these regions are relatively

747 colder and drier but not colder and wetter. A colder-wetter  winter season is a

748 necessary condition for winter precipitation to be  a water source for the formation of groundwater

749 under a summer monsoon climate. This is because the bigger amounts of summer precipitation will

750 easily remove or weaken the depleted isotopic signals of winter prepicitation in groundwater. In this

751 regard,  modern winter precipitation is unlikely to  be an

752 important source of groundwater in the Otindag.

753 As to the palaeowaters" formed in colder and wetter periods such as the last

754 glacial", it has been proposed to be a potential water source for groundwaters in the wide arid lands of

755 the world. the depleted signals of stable isotopes (δD $^2$and δ$^{18}$O) in groundwater have been

recognized in global arid and semi-arid regions, such as the Sinai Desert in Egypt (Gat and Issar, 1974) , Israel (Gat, 1983) , South Australia (Love et al., 1994, 2000)  northern China (Ma et al., 2010)  Saudi Arabia (Bazuhair and Wood, 1996) and North Africa ( Guendouz et al., 2003). The signals are very often explained as palaeo-groundwater that recharged by precipitation during past wetter and colder periods (Love et al., 1994, 2000; Herczeg and Leaney, 2011)~~( Love et al., 1994, 2000; Herczeg and Leaney, 2011) Gat and Issar (1974)reported that palaeowaters played a central role in the deep aquifers of the Sinai Desert, with the evidentsthat groundwater stable isotope compositions ($\delta^{18}$O and $\delta D^2$H) were more negative than those of weighted mean contemporary rainfall.Ma et al. (2010)presented data from groundwater in the aquifer of Jinchang city and the adjacent Gobi desert areas in northern China, which showed that palaeowaters were depleted in $^{18}$O and $^{2}$H relative to modern precipitation in the same region.~~

 Here we use the tritium data as a environmental tracer to estimate the groundwater age in the Otindag. the tritium data at the GNIP stations of the Baotou and Tianjin are also referenced as the background values in precipitation of recent years.

the residence time of groundwater in aquifer and the residual tritium of a water body can be calculated by $N = N_0 e^{-\lambda t}$ (Yang and Williams, 2003). Where N = content of residual tritium in water sample, $\lambda = 0.0565$, the radioactive decay constant, $N_0$ = content of tritium at the time of rainfall and t = years after precipitation. Based on this equation, the residual tritium was theoretically calculated and the standard for tritium dating was established for seven groundwater samples in the Otindag Desert (Table 3). As a result,  ages of 0-60 years were obtained for these groundwater samples (Table 5). This  indicates that recent recharge took place several decades after the peak in global nuclear tests . we thus conclude that groundwater is generally not older than 70 years in the study area. It means that  groundwater in the Otindag are not palaeowater recharged

Both the modern summer and winter precipitation recharge and the palaeowater recharge can be refuted, indicating that direct recharge is not a major mechanism controlling the groundwater recharge in the Otindag.

**44. Remote water recharge on groundwater in the Otindag: Dali Basin**

796 volumetrically important source of groundwater in the Otindag, and the groundwater was not recharged
797 by palaeowaters.Thus, Tthe third hypothesis that : "remote/the mountains waters emanate under with
798 colder and wetter conditions" is further ,should be considered here as a key souce of groundwaterin the
799 Otindag. In essence, it is an indirect recharge meachanism, as the indirect recharge is defined as as
800 water originatesd from remote areas (Healy, 2010; Herczeg and Leaney, 2011).(Healy, 2010)and it
801 generally occurs through rivers, canals, lakes and flash floodings(Herczeg and Leaney, 2011).

802 It iswas worth noting that the values of deuterium and oxygen-18 in the groundwater samples of
803 the eastern Otindagare were variable. These valuesfor groundwater in the north part of the study area
804 arewere more depleted in $\delta D^2H$ and $\delta^{18}O$ than those in the south part (Table 3). It suggests that the
805 Otindag groundwater in the study area might be potentially recharged by water resouces coming from
806 the northern neighboring catchment of the eastern Otindag, such as the Dali Basin.

807 In order to estimate the potential linkage between the eastern Otindag and the Dali Basin,
808 RRecently published data of $\delta D$ and $\delta^{18}O$ deuterium and oxygen-18 in groundwaters, lake waters, river
809 waters and spring water ssampled from the Dali Basin (e.g., Chen et al., 2008; Zhen et al., 2014) were
810 compiled collectedin this study and were co-analyzed with the data from the Otindag.

811 In total, There were totally aAbout 70 natural water samples from the Dali and Otindag with $\delta D^2H$
812 and $\delta^{18}O$ values arebeing shown in a Craig diagram (Fig. 9). As a result, Aall of these samples fell on
813 or lied near the evaporation line EL2 in the Craig diragram (Fig. 9), with a regression equation of $\delta D$
814 $^2H$= 4.81$\delta^{18}O$ - 21.55 and a high ercorrelation coefficient ($R^2$=0.98, n=70), than that of EL1 ($R^2$=0.93,
815 n=24) for the Otindag samples.

816 Compared to the groundwater samples in the Otindag, water samples from the groundwaters,
817 rivers and springs fromin the Dali Basin arewere more depleted in $\delta^{18}O$ and $\delta D^2H$ (Fig. 9). Such results
818 further indicate dthat, in terms of itsthe isotopic signatureperspective, the groundwater in the eastern
819 Otindag has a close relationship with the natural waters in the Dali Basin, except for the lake water in
820 Dali.It seems that the Dali wateris a potential source for groundwater in the Otindag, or both of them
821 are recharged by a common souce derived from surrounding mountains.

823 **5. 4. 1. Linkage of the river water in the Dali and the groundwater in the Otindag**
824 The similar signals of $\delta D$ and $\delta^{18}O$ deuterium and oxygen 18 between the groundwater in the
825 Otindag and the river water in the Dali (Fig. 9) point towards thegave us a possible idea that the
826 groundwater in the Otindag might be sourced from the river water in the Dali Basin, since the Dali has
827 more depleted isotopic signals in water than the Otindag (Fig. 9).

828 Considering Regarding to the topographical gradient of the elevations between the two regions,
829 however, river water in the Dali Basin cannot't flow into the eastern Otindag, because the terrain
830 elevation of the Dali Basin is lower than that of the Otindag (Fig. 1). This is also the reason why the
831 huge Dali Lake is formedthat lies in the Dali Basin has no equivalent but not in the Otindag (Fig. 1).
832 If there is a hydraulic linkage between the two regions, water should flow from the Otindag into the the
833 Dali, but not conversely.

834 IA hypothesis that water flows from the Otindag into the Dali Lake has also been proposed by
835 Yang et al. (2015). They argued that a mega-palaeolake in Dali, who was almost twice the size of the

~~present Dali Lake in area, was recharged by river systems to its south in the Otindag ca. 4,200 years ago. After that, due to the catastrophic decrease in precipitation that occurred inmonsoonal regions being experienced catastrophic precipitation decreasing and the groundwater in Otindag being sappinged and captured of the Otindag groundwater by the Xilamulun River flowing eastward, the Otindag's water was no longer recharging the megalake Dali and left a palaeo-channel between the two regions(Fig. 2). Since then the connection between surface waters in the two regions has been halted.was broken.~~

In view of the hydraulic gradient, river water in the Dali Basin could not be a recharge source for groundwater in the Otindag. However, in view of the isotopic gradients, groundwater in the Otindag could not conversely be the source of river water in the Dali (Fig. 9). Thus, the similar isotopic signals between the river water in Dali and the groundwater in Otindag indicate that these waters might be recharged from a common source.

Similar isotopic signals also occurred in the groundwaters between the Otindag and the Dali Basin (Fig. 9). In order to understand the linkage of groundwaters between the two regions, the potential movement of groundwater in the transition zone of the two regions need to be known.

~~Due to the inherent difficulties to directly observe groundwater movement along its hydraulic gradient under ground, inert isotopic and hydrochemical tracers are often used to identify groundwater movement (Nakaya et al., 2007), such as chloride, TDS and H-O isotopes, which arewere used as environmental fingerprints to indicate groundwater movement in arid lands (Yang and Williams, 2003). In a theoretical line of groundwater evolution, the chloride in water is readily removed from matrix materials rather than being precipitated due to its high solubility, thus chloride concentrations tend to be increased with the increasing of the flow path's length and residence time of groundwater (Lioyd and Heathcote, 1985). The TDS has a similar trend with chloride in groundwater evolution, but its tendency might be disturbed due to potential precipitation of certain ions when reaching their saturation conditions. According to the salination classification of water, all the groundwater samples collected in this study arewere fresh water in type (TDS < 1000 mg/L). Thus evident precipitation of major ions can be considered as could be weak in theOtindag groundwaters.~~

In this study, a groundwater-sampling project was designed in the field along a N-S section of a palaeo-channel located at the transition zone between the Dali and Otindag (Figs. 1, 2). The channel was named "PCSX" in this study, with its  north part named " NPCSX" and the south part named " (SPCSX"

Regarding to the topographical gradient in the Otindag, he GPS elevation of the northernmost sampling site in the NPCSX (g11, about 1317 m a.s.l.) was much lower than that of the southernmost

876 site in the SPCSX (g1, 1396 ma.s.l.) (Fig. 2 and Table 1). Regarding to the topographical gradient in
877 the channel, tThere is a drop of It is about an 80 m meter drop between the NPCSX and the SPCSX.
878 Under such slope, the underground hydraulic gradient for groundwater flow can be roughly parallel
879 with that of the surface water flow, namely that the groundwaterflow should move downwards from the
880 SPCSX area into the NPCSX area. Thus we can speculate that groundwater in the NPCSX would have
881 higher salinity concentration values of chloride and TDSin concentration than those in the SPCSX
882 under such flowing direction, if the groundwater was flowing from the SPCSX to the NPCSX.

In order to verify check up this speculation, the actual variations of water thesalinity environmental tracers(chloride and TDS) were detected along the PCSX section. The sampling site g1 was defined as the initial point and the distances between g1 and other sampling sites along the PCSX section were calculated, based on their GPS geographical coordinates records measured in the field. The results arewere shown in Fig. 10a-b. It is was veryclear that the variations of chloride and TDS concentrations in groundwater dodid not increase along the palaeo-channel from south to north (Fig. 10a-b). On the contrary, both the values of chloride and TDS arewere lower in the NPCSX area than those in the SPCSX area. Such kind of spatial variations in the chloride and TDS values was contradicted to the speculated patterns abovementioned, suggesting that the hydraulic gradient of groundwater flowing path in this region is not controlled by the topographical gradient between the NPCSX and SPCSX areas.

a complicated movement of groundwater in the study area. It also indicatesd that the hydraulic linkage wasweak in the groundwaters between the NPCSX and SPCSX areas.

SThe stable and radioactive isotopic data were also used here as tracers to differentiate the groundwaters between the two regions.Before we use the stable isotopic signals, however,it is necessary to think about the effect of evaporation process on the fractionation of stable isotopes. During the evaporation process, dissolved chloride, the conservative ion, will be enriched along with the heavy isotopes, which is manifested as a correlation between the chloride concentration and the deuterium content in groundwater (Sklash and Mwangi, 1991; Taylor and Howard, 1996).Based on this consideration, a bivariate diagram can be was built using the chloride and deuterium data of the groundwater samples in this study, as shown in Fig. 11. The gGroundwater samples from the PCSX section showed a very weak correlation between the chloride and deuterium (Fig. 11).This indicatesd that the groundwaters studied are werenot strongly affected by evapotation process in a deep degree.

Compared between the NPCSX and SPCSX regions, the stable isotopic values ($\delta^{18}$O and $\delta D^2H$) of groundwaters in the SPCSX region varyied greatly with a large amplitude, while those in the NPCSX arewere relatively constant (Fig. 10c-d12). This indicatesd that the recharge sources of groundwater in the SPCSX are more diverse were diversity than those in the NPCSX.The constant variations indicated that the recharge source of groundwater in the NPCSX is relatively unitary. The isotopic values in the SPCSX arewere much lighter than those in the NPCSX along the distance section from south to north (Fig. 10c-d12). The heaviest valuses occurred in the sample g11 collected from the NPCSX (Fig. 10c-d12), indicating a water being earlier firsthand recharged. The spring water sample s2, a representation of discharge water, iswas characterized by medium values of $\delta D^2H$ and $\delta^{18}$O. Similarly, the deuterium excess values of these groundwaters also showed such spatial patterns in the two regions

(Fig. 10e f13), These results indicated that the groundwaters in the SPCSX area, with relatively enriched isotopic signals in $\delta^2$HD and $\delta^{18}$O by comparison with than those in the NPCSX area, arewere composed of an mixture of the groundwaters in the NPCSX withand other waters. In consequence, thus resulting in the spring water sample s2in the discharge zoneisbeingcharacterized by an intermediate isotopic signal (Figs. 12, 13). A similar case was also observed by Abdalla (2009), who reported a thatprogressive decrease of the isotopic compositions had decreased progressively along a regional-scale flow path of groundwater in the semi-arid central Sudan, because of the mixture of groundwaterswithbetween the heavier/lighter isotope rechargesd and the lighter isotope recharged.

In addition to stable isotopes, tThe tritium contents were broadly and positively related to the values of deuterium excess in the groundwater samples in the PCSX section (Fig. 10eg14a). The deuterium excess or d-excess, computed from the equation d = $\delta^2$H – 8$\delta^{18}$O (Dansgaard, 1964),is controlled primarily by themean relative humidity of the air masses formed above the water surface (Merlivat and Jouzel, 1979) and generally reflects the rate of evaporation process experienced during the flowing paths(Dansgaard, 1964).For a water that experiences dan evaporation process, the d-excess value will increase in the evaporated water vapor, but will decrease in the residual water body (Dansgaard, 1964; Merlivat and Jouzel, 1979). In this study, except for sample g11 (a sample very close to the riverhead area), the positive relationship between the tritium and the deuterium excess generally shows edthat the d-excess values arewere higher in the groundwaters collected from the NPCSX, but are werelower in those from the SPCSX (Fig. 10eg14a). This edistribution pattern indicatesd that the groundwaters in the NPCSX arewere relatively younger and had experienced a lower lessdegree of evaporation than those in the SPCSX. The d-excess gradient, increasing from the south to north in the PCSX, further suggestseonfirmsed that groundwater does did not flow from the SPCSX area to the NPCSX area, namely out of the topographical control.

In Fig. 14b, the tritium contents of groundwater increased while the TDS decreasesdfrom the south to north in the PCSX (Fig.14b). This distribution pattern of the two environmental tracers further proved that the groundwaters in the NPCSXarewere younger and fresher than those in the SPCSX. The reason why the older groundwater hasa higher TDS value can be attributed to the fact that most minerals dissolve slowly in an aquifer and the older groundwater stay have morein contacting with the surrounding rocks for a longer time allowing more time to act between water solution and soluble minerals to pass in solution into the water, leading to a higher TDS (Fitts, 2002). Many studies (e.g., Boronina et al., 2005; Kazemi et al., 2006) have demonstrated that groundwater will flows in the direction in which it gets older. In view of this point, groundwaters in the PCSX region should theritically flow from the NPCSX area to the SPCSX area, in opposition to evidently being paradoxical with the S-N topographical gradient between the Otindag and Daliin the PCSX regions. Thus groundwater in the Dali are not the source of groundwater in the Otindag. The similar isotopic signals between groundwaters in the two regions indicate that these waters might be recharged from a common source in other place.

Overall, it impliesdthat the hydraulic gradient of groundwater derived from thein topography is not consistent with the isotopic and hydrogeochemical gradients of groundwaterthat is observed in the

956
957
958

959 **5. 5. Remote water recharge on groundwater in the Otindag: mountains waters**
960 ****
961 ****

962 The discussions above revealed
963  isotopic perspective, and that
964 both the  groundwaters in the Otindag and DaliBasin might be recharged
965 from a common source derived from another place. Considering the third hypothesis abovementioned
966 that "remote/mountains waters emanate under colder and wetter conditions", we propose that this
967 "common source" of the two regions are from mountians areas surrounding the Otindag and Dali
968 Basin.
969
970 Otindag
971

972
973  There are two large permanent rivers and lots of small intermittent streams
974 entering the Dali Basin (Xiao et al., 2008), including the Xilamulun River to the south and the
975 Gongger River to the north, both of which are stemming from the Greater Khingan Mountains
976 (Daxing'Aanling Mountains in Chinese pinyin, 1,100-1,400 m above seal level) (Fig. 1).
977
978  The Xilamulun River carries a large
979 amount of water (about $6.58×10^8$ m$^3$/y) from the Daxing'Aanling Mountains flowing through the east
980 margins of the Dali and Otindag (Wu et al., 2014). This is an important clue linking natural waters
981 between  the  Otindag and  the
982 Dali Basin.

983 Variation in the elevation from the Dali Lake to the riverhead of the Xilamulun River can be
984 clearly found along a land surface topographical section (Fig. 11). The channel of the Xilamulun
985 River is located in River Fault or the Xar Moron  Fault (Fig. 1), which
986 is a part of the Solonker Suture Zone (Eizenhöfer et al., 2014) or the Xilamulun-Changchun-Yanji plate
987 suture zone (Sun et al., 2004) the in the regional
988 tectonical settings (Figs. -3). Outcrop observations indicate that fault zones commonly
989 have a permeability structure suggesting they should act as complex conduit–barrier systems in which
990 along-fault flow is encouraged and across-fault flow is impeded (Bense et al., 2013). Thus the
991 hydraulic grediant of groundwater flow in the Eastern margins of the Otindag and Dali Basin must be
992 controlled by the fault zone hydrogeology. This may be the reason why the hydraulic gradient of
993 groundwater represented by the isotopic and hydrogeochemical gradients of groundwater samples in
994 this study is not consistent with the local topographical gradient in the Otindag Desert. On the other
995 hand, the regional aquifer is generally unconfined in dune fields of the Otindag Desert but

semi-confined to confined in the Daxing'Anling uplands (Fig. 3), thus the thick unconsolidated aquifers in the study area (Figs. 32 and 115) will be favourable conditions for groundwater storage and transportation along the Solonker Suture Zone. When rivers stem from the Daxing'Aanling Mountainsand flow downward to the marginal areas of the Dali and Otindag, leakage water from these rivers can recharge the desert land through thick unconsolidated aquifers (Fig. 15). A strong isotopic evidence is that the lake and river waters in the Dali Basin share the same evaporation line (EL2) with the groundwaters in the PCSX area.

Although groundwaters in the SPCSX area are weredifferent from those in the NPCSX area, their isotopic data points still fell onto the EL2 (Fig. 9XX), which further indicatesd that the groundwaters in the SPCSX arewere a mixture of waters from the Daxing'Aanling Mountain and other sources. Another

Another source for groundwater recharge in the SPCSX could can be speculatedrepresented by toremote water such as flash floods derived coming from the north Yinshan Mountains (Fig. 1), because it can be clearly observed from digitial maps that many transient rivers or streams originated from the Yinshan Mountrains flow into the south and southeastern Otindag (Fig. 1). Supportive evidence for this idea can be derived A keycluefor this view can also be obtained from the isotopic signals of local precipitation and groundwater samples collectedfromthe areas near to the Yinshan Mountains in this study. Supportive evidence for this idea can also be observed in the summer rainy season. During rainy days or under storm conditions, occasional heavy, short rainstorms cause floods in soil-rich wadi channels and low-lying depressions in the unconfined to semi-confined areas of the Yinshan Mountains' piedmont. These waters may temporarily recharge shallow aquifers in the SPCSX area.

It has been reported that temperature and altitude can deeply affect theδD and δ$^{18}$Ocompositions of precipitationthe isotope depleted signals ofδ$^{2}$H and δ$^{18}$O in watersfrom mountain areasto can be passed into the groundwater in aplain areas (Harrington et al., 2002; Vanderzalm et al., 2011; Liu and Yamanaka, 2012; Rattray, 2015; Khalid and Hamid, 2017). Rattray (2015)attributed this isotopic signature to the altitude effect on precipitation, because temperature and altitude can deeply affect the deuterium and oxygen-18 compositions in precipitation. The values of δ$^{2}$H and δ$^{18}$O in precipitation from the mountain areaswill be depleted when compared with those in precipitation from the piedmont areas (Rattray, 2015). Rattray (2015)attributed this isotopic signature to the altitude effect on precipitation. For the Yinshan Mountain Range, there is lack of the data ofstable isotopes datain precipitation are lackingfrom the mountains in this study. However, based on the altitude effect of temperature on isotopic signals,wecan theoreticallyestimate the values usingthe precipitation sample (p1), which wascollected fromthe piedmont area of the Yinshan Mountains in this study. For example, tThe GPS elevation of the sample location of p1 is about 1260 m a.s.l. and that of the top of the Yinshan mountain range is around 1700-1800 m a.s.l., thus the elevation drop is approximately 500 m between the two sites. Based on this difference in elevation dropand thea potentialeffect of elevation change on temperature ( that elevation arises will lead to a decrease of temperature by0.65°C per 100 m), thetemperature difference between the two sites is about 3.25 °C. According to an empirical estimation for precipitation in NW China that the δ$^{18}$O-temperature gradient is 0.37 ‰/°Candthe

1036 $\delta^{18}$O elevation gradient is -0.13‰/100 m (Liu et al., 2014), the $\delta^{18}$O value in precipitation at the
1037 Yinshan Mountains shall be 1.85 ‰ lower than that in the sample p1, namely -8.99‰ in$\delta^{18}$O for the
1038 Yinshanmountain precipitation. This value is very similar to that of the groundwater (-9‰) in the
1039 SPCSX area. It indicates that the Yinshan Mountains area is area potential source area for the
1040 groundwater recharge inthe SPCSX area.

1041 In general, the above analyses revealed that the highland water resources from the Daxing'Anling
1042 and Yinshan Mountains were isotopically and geochemically traced to be a major source for the
1043 groundwater in the Otindag. It suggests means that the modern indirect recharge mechanism, instead of
1044 the direct recharge and the palaeowater recharge, isresponsible for groudnwater recharge in thisthe
1045 desert land in northern China. It also This implies that the tectonic settings (such as the Solonker suture
1046 zone), but not the climatic and topographical e control, iswassignificant for the groundwater origin in
1047 the Otindag.

1048

**56. Conclusions**

1050 Water resources in arid lands of the world are generally scarce and highly uncertain. In the
1051 middle-latitude desert zone of northern China, however, many deserts such as the Otindag and
1052 Badanjilin Deserts, are unexpectedly rich in incommensurate groundwater resources, such as the
1053 Otindag and the Badanjilin Deserts, although they have no surface runoff and have been under an arid
1054 or hyper-arid climate for a long geological period of time. How the groundwaters are originated and
1055 recharged in thesea deserts environment are thus becaming a key questions that are longtime ago, but
1056 it is still under an endless debate at present in the acadamic circle. For some of the earth scientists, the
1057 direct recharge is thought to be very important for groundwaters in the wide desert lands of northern
1058 China, due to the lack of surface runoffs. However, the groundwater availability is very much as
1059 function of the local- and regional-scale geological and climatic settingscomponents. To achieve an i
1060 Integrated understanding of the groundwater recharge and its their controlling mechanisms is of great
1061 significance. In this study, an effort to explore the groundwater recharge was explored carried out using
1062 multiple environmental tracers in the eastern Otindag Desert of northern China, a region that where is
1063 under the influence control of the East Asian Summer Monsoon (EASM) climate. The results showed
1064 that (1), the natural waters in the study area arewere fresh water (TDS < 1000 mg/L) with and awere
1065 neutral to slightly alkalinepH. The major water types arewere the Ca-HCO₃ and Ca/Mg-SO₄ types.
1066 There arewere no Cl-type and Na-type waters occurring in the study area, indicating a primary stage of
1067 water evolution in terms of the hydrogeochemicalperspectiveterms. (2) Compared to the modern
1068 summer precipitation, the groundwaters, river waters and spring waters are weredepleted in $\delta D^2H$ and
1069 $\delta^{18}$O, while the lake waters arewere enriched in $\delta^2$H and $\delta^{18}$O. All these waters, however, shared a same
1070 line of evaporation in the Craig linediagram, indicating a genetic relationship on their recharge sources.
1071 The more depleted stable isotopic signals ofin the groundwaters is more depleted than those ofin the
1072 modern summer precipitation and this suggestsed that the groundwaters studied here could only be
1073 sourced from a colder water different from other than the EASM precipitation. In general, the analyses
1074 revealed that the highland remote water resources from the Daxing'Anling and Yinshan Mountains
1075 were isotopically and geochemically traced to be a major source for the groundwater in the Otindag.

The contribution from local winter precipitation iswas very small due to its weak rainfall effect. The high contents (5-25 TU) of tritium in these groundwaters indicated that they arewere young and are could not be recharged by palaeowaters formed during the past glacial periods. (3) There are cClear difference in the isotopic signals occurred between the groundwaters in the north (NPCSX) and south (SPCSX) parts of the study area, but the signals of were silimar between the groundwaters in the NPCSX are similar to that of and its neighbouring catchment, the Dali Basin. (4) Combined analysis was further performed using the isotopic and physiochemical data of natural waters collected from the Dali Basin and the surrounding mountains. The resulsts indicated that the major sources of the groundwaters in the NPCSX, as well as the river waters and groundwaters in the Dali Basin, arewere mainly derived from the Daxin'Anling Mountains, by leaking the Xilamulan River water through a thick aquifer in the eastern margins of the Otindag. By contrast, While the groundwaters in the SPCSX arewere mainly recharged from two sources, the flash floods from the Yinshan Mountains and the river waters from the Daxin'Anling Mountains. (5)It suggests that tThe modern indirect recharge mechanism, instead of the direct recharge and the palaeo-water recharge, iswas the most significant for groudndwater recharge in the eastern Otindag. It indicates that the tectonic settings at a regional scale, but not the climate and topography, iswasat the origin of responsible for the groundwater origin in the Otindag. This study providesd a new perspective sight into the origin and evolution of groundwater resources in the middle-latitude desert zone of northern China.

**Acknowledgements**

This study was financially supported by the National Natural Science Foundation of China (41602196, 41771014 and 41602196) and the National Key Research and Development Program of China (2016YFA0601900). We thank the China Meteorological Data Sharing Service system for providing the weather data. Sincere thanks are also extended to Profs. Xiaoping Yang, Xunming Wang, Jule Xiao and other workmates, e.g., Qiuhong Li, Ziting Liu, Hongwei Li, and DeguoZhangfor their generous help in the research work.

[revised manuscript text omitted]

1380 **Fig. 2,** (a) Tectonic framework of the north China-Mongolian segment of the Central Asian Orogenic
1381 Belt (modified after Jahn, 2004). (b) Geological sketch map of the northern China-Mongolia tract
1382 (modified after Jian et al., 2010). The Solonker suture zone represents the tectonic boundary between
1383 the northern (Hutag Uul Block-Northern orogen) and the southern (southern orogen-Northern margin
1384 of North China craton) continental blocks. Note that the red line marks the early Permian
1385 paleobiogeographical boundary (Wang and Liu, 1986; Li, 2006), which coincides with the northern
1386 boundary of the suture zone.

[Figure]

1395 **Fig. 3.** The hydrogeological division map of the Otindag Desert.

[Figure]

1396

[Figure]

Legend:

● Groundwater  ── The Xar Moron Fault
+ Lake water  ── Section S1
■ River water  ┈┈ Paleao channel
◆ Spring water  ── River
▶ Precipitation  ⊙ City

2
3
4

Elevation (m)
High : 2042
Low : 597

**Fig. 42.** The locations of the water sampling sites in this study.

[Figure]

[Figure]

1433 **Fig. 53.** The  fingerprint diagram showing the variations of
1434 multiple ions' concentrations in the studied water samples in an equivalent unit. The $HCO_3+CO_3$
1435 concentration in the sample p1 was not shown, due to its value being lower than the detection limit.

[Figure]

1436

[Figure]

1448 **Fig. 6-4.** The Piper diagram (Piper, 1944) showing the relative abundances of major cations and anions
1449 in the studied water samples. Major water types are also shown in this diagram.

[Figure]

1450

[Figure]

1471 **Fig. 5. An Expanded Durov diagram (Durov, 1948; Lioyd and Heathcote, 1985; Al-Bassam et al.,**
1472 **1997; Chadha, 1999; Al-Bassam and Khalil, 2012) showing the linear dissolution or mixing**
1473 **process for groundwater and the ion-exchange process occurred in the groundwater and other**
1474 **waters in the study area.**

[Figure]

**Fig. 76.** The bivariate diagram of  δD and δ¹⁸O, i.e. the Craig diagram, for the natural water samples in this study. Different relationships between the groundwaters, lake waters, river waters, spring waters and the precipitation waters are  illustrated. AWMB, the annual weighted mean value at the Baotou station; AWMT, the annual weighted mean value at the Tianjin station; LWMB, thelong-term weighted means at the Baotou station; LWMT, the long-term weighted means at the Tianjin station; GMWL, the Global Meteoric Water Line; LMWL-B, the local meteoric water line calculated based on the data from the Baotou station; LWML-T, the local meteoric water line calculated based on the data from the Tianjin station; EL1, the evaporation line calculated based on the data of water samples collected in this study.

[Figure]

1485

[Figure]

**Fig. 87.** The seasonal mean distributions of (a) precipitation (a), (b) surface air temperature (b) and (c) water vapor pressure (c) from the Baotou and Tianjin weather stations (station sites seen in **Fig. 1a**) in the surrounding areas of the Otindag for the period in recent thirtyyears (1981-2010).

[Figure]

1504
1505 **Fig. 8.** The seasonal mean distributions of (d) δ¹⁸O (a) and (e) δD²H (b) values in precipitation from the
1506 Baotou and Tianjin weather stations in the surrounding areas of the Otindag for the period in recent
1507 sixteen years (1986-2001).

[Figure]

1508

[Figure]

**Fig. 9.** The bivariate diagram of δD$^2$ and δ$^{18}$O, i.e. the Craig diagram, for the natural water samples collected in the Otindag (this study) and the Dali Basin. Different relationships between the groundwaters, lake waters, river waters, spring waters and the precipitation waters are clearly illustrated. AWMB, AWMT, LWMB, LWMT, GMWL, LMWL-B, LMWL-T, and EL1 are the same as in  Fig. 7. EL2, the evaporation line calculated based on the data from the groundwater, lake water, river water and spring water samples collected from  the Otindag and  Dali Basin. The data for  the Dali were taken from studies(Chen et al. (2008) and  Zhen et al. (2014).

[Figure]

1527

[Figure]

1545 **Fig. 10.** (a) Sketch map showing the relationship between the groundwaters in the NPCSX and
1546 SPCSX areas, based on variations of (a) the chloride concentrations, (a)and (b) the TDS (b)
1547 concentrations, (c) the $\delta^{18}O$ values and (d) the $\delta D$ values of these water samples versus their distances
1548 away from the water sample g1 along the palaeo river channel (PCSX) from south to north.

[Figure]

1554 **Fig. 11.** The bivariate plot of Cl vs. δ²H in the groundwaters from the PCSX region, which showing that
1555 no significant evaporation process has been experienced by these groundwaters.

[Figure]

1556
1557 **Fig. 12.** Variations of δ¹⁸O (a) and δ²H(b) in the groundwaters versus their distances away from the
1558 groundwater sample g1 along the palaeo river channel (PCSX) from south to north. The dashed line in
1559 (c) and (d) represents the corresponding values of the spring water sample s2, and which is just well
1560 divided the samples into the NPCSX and the SPCSX parts.

[Figure]

[revised manuscript text omitted]

无下划线, 字体颜色: 自动设置

无下划线, 字体颜色: 自动设置

无下划线, 字体颜色: 自动设置

无下划线, 字体颜色: 自动设置

无下划线, 字体颜色: 自动设置

无下划线, 字体颜色: 自动设置

无下划线, 字体颜色: 自动设置

定义网格后不调整右缩进, 不对齐到网格

无下划线, 字体颜色: 自动设置

无下划线, 字体颜色: 自动设置

无下划线, 字体颜色: 自动设置

无下划线, 字体颜色: 自动设置

无下划线, 字体颜色: 自动设置

无下划线, 字体颜色: 自动设置

无下划线, 字体颜色: 自动设置

无下划线, 字体颜色: 自动设置

无下划线, 字体颜色: 自动设置

无下划线, 字体颜色: 自动设置

无下划线, 字体颜色: 自动设置

无下划线, 字体颜色: 自动设置

无下划线, 字体颜色: 自动设置

无下划线, 字体颜色: 自动设置

定义网格后不调整右缩进, 不对齐到网格

无下划线, 字体颜色: 自动设置

无下划线, 字体颜色: 自动设置

无下划线, 字体颜色: 自动设置

无下划线, 字体颜色: 自动设置

|---|---|---|

无下划线, 字体颜色: 自动设置

|---|---|---|

无下划线, 字体颜色: 自动设置

|---|---|---|

无下划线, 字体颜色: 自动设置

|---|---|---|

无下划线, 字体颜色: 自动设置

|---|---|---|

无下划线, 字体颜色: 自动设置

|---|---|---|

无下划线, 字体颜色: 自动设置

|---|---|---|

无下划线, 字体颜色: 自动设置

|---|---|---|

无下划线, 字体颜色: 自动设置

|---|---|---|

无下划线, 字体颜色: 自动设置

|---|---|---|

无下划线, 字体颜色: 自动设置

|---|---|---|

定义网格后不调整右缩进, 不对齐到网格

|---|---|---|

无下划线, 字体颜色: 自动设置

|---|---|---|

无下划线, 字体颜色: 自动设置

| --- | --- | --- |

无下划线, 字体颜色: 自动设置

| --- | --- | --- |

无下划线, 字体颜色: 自动设置

| --- | --- | --- |

无下划线, 字体颜色: 自动设置

| --- | --- | --- |

无下划线, 字体颜色: 自动设置

| --- | --- | --- |

无下划线, 字体颜色: 自动设置

| --- | --- | --- |

无下划线, 字体颜色: 自动设置

| --- | --- | --- |

无下划线, 字体颜色: 自动设置

| --- | --- | --- |

无下划线, 字体颜色: 自动设置

| --- | --- | --- |

无下划线, 字体颜色: 自动设置

| --- | --- | --- |

无下划线, 字体颜色: 自动设置

| --- | --- | --- |

无下划线, 字体颜色: 自动设置

| --- | --- | --- |

无下划线, 字体颜色: 自动设置

| --- | --- | --- |

定义网格后不调整右缩进, 不对齐到网格

| --- | --- | --- |

无下划线，字体颜色：自动设置

| | | |
|---|---|---|

无下划线，字体颜色：自动设置

| | | |
|---|---|---|

无下划线，字体颜色：自动设置

| | | |
|---|---|---|

无下划线，字体颜色：自动设置

| | | |
|---|---|---|

无下划线，字体颜色：自动设置

| | | |
|---|---|---|

无下划线，字体颜色：自动设置

| | | |
|---|---|---|

无下划线，字体颜色：自动设置

| | | |
|---|---|---|

无下划线，字体颜色：自动设置

| | | |
|---|---|---|

无下划线，字体颜色：自动设置

| | | |
|---|---|---|

无下划线，字体颜色：自动设置

| | | |
|---|---|---|

无下划线，字体颜色：自动设置

| | | |
|---|---|---|

无下划线，字体颜色：自动设置

| | | |
|---|---|---|

无下划线，字体颜色：自动设置

| | | |
|---|---|---|

无下划线，字体颜色：自动设置

定义网格后不调整右缩进, 不对齐到网格

无下划线, 字体颜色: 自动设置

无下划线, 字体颜色: 自动设置

无下划线, 字体颜色: 自动设置

无下划线, 字体颜色: 自动设置

无下划线, 字体颜色: 自动设置

无下划线, 字体颜色: 自动设置

无下划线, 字体颜色: 自动设置

无下划线, 字体颜色: 自动设置

无下划线, 字体颜色: 自动设置

无下划线, 字体颜色: 自动设置

无下划线, 字体颜色: 自动设置

无下划线, 字体颜色: 自动设置

无下划线, 字体颜色: 自动设置

无下划线, 字体颜色: 自动设置

定义网格后不调整右缩进, 不对齐到网格

无下划线, 字体颜色: 自动设置

无下划线, 字体颜色: 自动设置

无下划线, 字体颜色: 自动设置

无下划线, 字体颜色: 自动设置

无下划线, 字体颜色: 自动设置

无下划线, 字体颜色: 自动设置

无下划线, 字体颜色: 自动设置

无下划线, 字体颜色: 自动设置

无下划线, 字体颜色: 自动设置

无下划线, 字体颜色: 自动设置

无下划线，字体颜色：自动设置

|---|---|---|

无下划线，字体颜色：自动设置

|---|---|---|

无下划线，字体颜色：自动设置

|---|---|---|

无下划线，字体颜色：自动设置

|---|---|---|

定义网格后不调整右缩进，不对齐到网格

|---|---|---|

无下划线，字体颜色：自动设置

|---|---|---|

无下划线，字体颜色：自动设置

|---|---|---|

无下划线，字体颜色：自动设置

|---|---|---|

无下划线，字体颜色：自动设置

|---|---|---|

无下划线，字体颜色：自动设置

|---|---|---|

无下划线，字体颜色：自动设置

|---|---|---|

无下划线，字体颜色：自动设置

|---|---|---|

无下划线，字体颜色：自动设置

|---|---|---|

无下划线，字体颜色：自动设置

|---|---|---|

无下划线, 字体颜色: 自动设置

|---|---|---|

无下划线, 字体颜色: 自动设置

|---|---|---|

无下划线, 字体颜色: 自动设置

|---|---|---|

无下划线, 字体颜色: 自动设置

|---|---|---|

无下划线, 字体颜色: 自动设置

|---|---|---|

定义网格后不调整右缩进, 不对齐到网格

|---|---|---|

无下划线, 字体颜色: 自动设置

|---|---|---|

无下划线, 字体颜色: 自动设置

|---|---|---|

无下划线, 字体颜色: 自动设置

|---|---|---|

无下划线, 字体颜色: 自动设置

|---|---|---|

无下划线, 字体颜色: 自动设置

|---|---|---|

无下划线, 字体颜色: 自动设置

|---|---|---|

无下划线, 字体颜色: 自动设置

无下划线, 字体颜色: 自动设置

无下划线, 字体颜色: 自动设置

无下划线, 字体颜色: 自动设置

无下划线, 字体颜色: 自动设置

无下划线, 字体颜色: 自动设置

无下划线, 字体颜色: 自动设置

无下划线, 字体颜色: 自动设置

定义网格后不调整右缩进, 不对齐到网格

无下划线, 字体颜色: 自动设置

无下划线, 字体颜色: 自动设置

无下划线, 字体颜色: 自动设置

无下划线, 字体颜色: 自动设置

无下划线, 字体颜色: 自动设置

无下划线, 字体颜色: 自动设置

|---|---|---|

无下划线, 字体颜色: 自动设置

|---|---|---|

无下划线, 字体颜色: 自动设置

|---|---|---|

无下划线, 字体颜色: 自动设置

|---|---|---|

无下划线, 字体颜色: 自动设置

|---|---|---|

无下划线, 字体颜色: 自动设置

|---|---|---|

无下划线, 字体颜色: 自动设置

|---|---|---|

无下划线, 字体颜色: 自动设置

|---|---|---|

无下划线, 字体颜色: 自动设置

|---|---|---|

定义网格后不调整右缩进, 不对齐到网格

|---|---|---|

无下划线, 字体颜色: 自动设置

|---|---|---|

无下划线, 字体颜色: 自动设置

|---|---|---|

无下划线, 字体颜色: 自动设置

|---|---|---|

无下划线, 字体颜色: 自动设置

|---|---|---|

无下划线, 字体颜色: 自动设置

|---|---|---|

无下划线, 字体颜色: 自动设置

|---|---|---|

无下划线, 字体颜色: 自动设置

|---|---|---|

无下划线, 字体颜色: 自动设置

|---|---|---|

无下划线, 字体颜色: 自动设置

|---|---|---|

无下划线, 字体颜色: 自动设置

|---|---|---|

无下划线, 字体颜色: 自动设置

|---|---|---|

无下划线, 字体颜色: 自动设置

|---|---|---|

无下划线, 字体颜色: 自动设置

|---|---|---|

无下划线, 字体颜色: 自动设置

|---|---|---|

定义网格后不调整右缩进, 不对齐到网格

|---|---|---|

无下划线, 字体颜色: 自动设置

|---|---|---|

无下划线, 字体颜色: 自动设置

|---|---|---|

无下划线, 字体颜色: 自动设置

|---|---|---|

无下划线, 字体颜色: 自动设置

|---|---|---|

无下划线, 字体颜色: 自动设置

|---|---|---|

无下划线, 字体颜色: 自动设置

|---|---|---|

无下划线, 字体颜色: 自动设置

|---|---|---|

无下划线, 字体颜色: 自动设置

|---|---|---|

无下划线, 字体颜色: 自动设置

|---|---|---|

无下划线, 字体颜色: 自动设置

|---|---|---|

无下划线, 字体颜色: 自动设置

|---|---|---|

无下划线, 字体颜色: 自动设置

|---|---|---|

无下划线, 字体颜色: 自动设置

|---|---|---|

无下划线, 字体颜色: 自动设置

|---|---|---|

定义网格后不调整右缩进, 不对齐到网格

|---|---|---|

无下划线，字体颜色：自动设置

|---|---|---|

无下划线，字体颜色：自动设置

|---|---|---|

无下划线，字体颜色：自动设置

|---|---|---|

无下划线，字体颜色：自动设置

|---|---|---|

无下划线，字体颜色：自动设置

|---|---|---|

无下划线，字体颜色：自动设置

|---|---|---|

无下划线，字体颜色：自动设置

|---|---|---|

无下划线，字体颜色：自动设置

|---|---|---|

无下划线，字体颜色：自动设置

|---|---|---|

无下划线，字体颜色：自动设置

|---|---|---|

无下划线，字体颜色：自动设置

|---|---|---|

无下划线，字体颜色：自动设置

|---|---|---|

无下划线，字体颜色：自动设置

|---|---|---|

无下划线，字体颜色：自动设置

|---|---|---|

定义网格后不调整右缩进，不对齐到网格

|---|---|---|

无下划线，字体颜色：自动设置

|---|---|---|

无下划线，字体颜色：自动设置

|---|---|---|

无下划线，字体颜色：自动设置

|---|---|---|

无下划线，字体颜色：自动设置

|---|---|---|

无下划线，字体颜色：自动设置

|---|---|---|

无下划线，字体颜色：自动设置

|---|---|---|

无下划线，字体颜色：自动设置

|---|---|---|

无下划线，字体颜色：自动设置

|---|---|---|

无下划线，字体颜色：自动设置

|---|---|---|

无下划线，字体颜色：自动设置

|---|---|---|

无下划线，字体颜色：自动设置

|---|---|---|

无下划线，字体颜色：自动设置

无下划线, 字体颜色: 自动设置

无下划线, 字体颜色: 自动设置

定义网格后不调整右缩进, 不对齐到网格

无下划线, 字体颜色: 自动设置

无下划线, 字体颜色: 自动设置

无下划线, 字体颜色: 自动设置

无下划线, 字体颜色: 自动设置

无下划线, 字体颜色: 自动设置

无下划线, 字体颜色: 自动设置

无下划线, 字体颜色: 自动设置

无下划线, 字体颜色: 自动设置

无下划线, 字体颜色: 自动设置

无下划线, 字体颜色: 自动设置

无下划线，字体颜色: 自动设置

无下划线，字体颜色: 自动设置

无下划线，字体颜色: 自动设置

无下划线，字体颜色: 自动设置

定义网格后不调整右缩进，不对齐到网格

无下划线，字体颜色: 自动设置

无下划线，字体颜色: 自动设置

无下划线，字体颜色: 自动设置

无下划线，字体颜色: 自动设置

无下划线，字体颜色: 自动设置

无下划线，字体颜色: 自动设置

无下划线，字体颜色: 自动设置

无下划线，字体颜色: 自动设置

无下划线，字体颜色: 自动设置

|---|---|---|

无下划线, 字体颜色: 自动设置

|---|---|---|

无下划线, 字体颜色: 自动设置

|---|---|---|

无下划线, 字体颜色: 自动设置

|---|---|---|

无下划线, 字体颜色: 自动设置

|---|---|---|

无下划线, 字体颜色: 自动设置

|---|---|---|

定义网格后不调整右缩进, 不对齐到网格

|---|---|---|

无下划线, 字体颜色: 自动设置

|---|---|---|

无下划线, 字体颜色: 自动设置

|---|---|---|

无下划线, 字体颜色: 自动设置

|---|---|---|

无下划线, 字体颜色: 自动设置

|---|---|---|

无下划线, 字体颜色: 自动设置

|---|---|---|

无下划线, 字体颜色: 自动设置

|---|---|---|

无下划线, 字体颜色: 自动设置

|---|---|---|

无下划线, 字体颜色: 自动设置

|---|---|---|

无下划线, 字体颜色: 自动设置

|---|---|---|

无下划线, 字体颜色: 自动设置

|---|---|---|

无下划线, 字体颜色: 自动设置

|---|---|---|

无下划线, 字体颜色: 自动设置

|---|---|---|

无下划线, 字体颜色: 自动设置

|---|---|---|

无下划线, 字体颜色: 自动设置

|---|---|---|

定义网格后不调整右缩进, 不对齐到网格

|---|---|---|

无下划线, 字体颜色: 自动设置

|---|---|---|

无下划线, 字体颜色: 自动设置

|---|---|---|

无下划线, 字体颜色: 自动设置

|---|---|---|

无下划线, 字体颜色: 自动设置

|---|---|---|

无下划线, 字体颜色: 自动设置

|---|---|---|

无下划线, 字体颜色: 自动设置

|---|---|---|

无下划线, 字体颜色: 自动设置

|---|---|---|

无下划线, 字体颜色: 自动设置

|---|---|---|

无下划线, 字体颜色: 自动设置

|---|---|---|

无下划线, 字体颜色: 自动设置

|---|---|---|

无下划线, 字体颜色: 自动设置

|---|---|---|

无下划线, 字体颜色: 自动设置

|---|---|---|

无下划线, 字体颜色: 自动设置

|---|---|---|

无下划线, 字体颜色: 自动设置

|---|---|---|

定义网格后不调整右缩进, 不对齐到网格

|---|---|---|

无下划线, 字体颜色: 自动设置

|---|---|---|

无下划线, 字体颜色: 自动设置

|---|---|---|

无下划线, 字体颜色: 自动设置

|---|---|---|

无下划线, 字体颜色: 自动设置

|---|---|---|

无下划线, 字体颜色: 自动设置

|---|---|---|

无下划线, 字体颜色: 自动设置

|---|---|---|

无下划线, 字体颜色: 自动设置

|---|---|---|

无下划线, 字体颜色: 自动设置

|---|---|---|

无下划线, 字体颜色: 自动设置

|---|---|---|

无下划线, 字体颜色: 自动设置

|---|---|---|

无下划线, 字体颜色: 自动设置

|---|---|---|

无下划线, 字体颜色: 自动设置

|---|---|---|

无下划线, 字体颜色: 自动设置

|---|---|---|

无下划线, 字体颜色: 自动设置

|---|---|---|

定义网格后不调整右缩进, 不对齐到网格

|---|---|---|

无下划线, 字体颜色: 自动设置

|---|---|---|

无下划线, 字体颜色: 自动设置

无下划线, 字体颜色: 自动设置

无下划线, 字体颜色: 自动设置

无下划线, 字体颜色: 自动设置

无下划线, 字体颜色: 自动设置

无下划线, 字体颜色: 自动设置

无下划线, 字体颜色: 自动设置

无下划线, 字体颜色: 自动设置

无下划线, 字体颜色: 自动设置

无下划线, 字体颜色: 自动设置

无下划线, 字体颜色: 自动设置

无下划线, 字体颜色: 自动设置

无下划线, 字体颜色: 自动设置

定义网格后不调整右缩进, 不对齐到网格

无下划线，字体颜色: 自动设置

|---|---|---|

无下划线，字体颜色: 自动设置

|---|---|---|

无下划线，字体颜色: 自动设置

|---|---|---|

无下划线，字体颜色: 自动设置

|---|---|---|

无下划线，字体颜色: 自动设置

|---|---|---|

无下划线，字体颜色: 自动设置

|---|---|---|

无下划线，字体颜色: 自动设置

|---|---|---|

无下划线，字体颜色: 自动设置

|---|---|---|

无下划线，字体颜色: 自动设置

|---|---|---|

无下划线，字体颜色: 自动设置

|---|---|---|

无下划线，字体颜色: 自动设置

|---|---|---|

无下划线，字体颜色: 自动设置

|---|---|---|

无下划线，字体颜色: 自动设置

|---|---|---|

无下划线，字体颜色: 自动设置

|---|---|---|

定义网格后不调整右缩进, 不对齐到网格

|---|---|---|

无下划线, 字体颜色: 自动设置

[remaining 94,659 characters of this post omitted]